# The critical dynamics of hippocampal seizures

Gregory Lepeu [1], Ellen van Maren [1], Kristina Slabeva[1], Cecilia Friedrichs-Maeder[1], Markus Fuchs[1], Werner J. Z'Graggen [2], Claudio Pollo[2], Kaspar A. Schindler [1], Antoine Adamantidis [1,4], Timothée Proix [3,4] & Maxime O. Baud [1,4] ✉

Epilepsy is defined by the abrupt emergence of harmful seizures, but the nature of these regime shifts remains enigmatic. From the perspective of dynamical systems theory, such *critical transitions* occur upon inconspicuous perturbations in highly interconnected systems and can be modeled as mathematical bifurcations between alternative regimes. The predictability of critical transitions represents a major challenge, but the theory predicts the appearance of subtle dynamical signatures on the verge of instability. Whether such dynamical signatures can be measured before impending seizures remains uncertain. Here, we verified that predictions on bifurcations applied to the onset of hippocampal seizures, providing concordant results from in silico modeling, optogenetics experiments in male mice and intracranial EEG recordings in human patients with epilepsy. Leveraging pharmacological control over neural excitability, we showed that the boundary between physiological excitability and seizures can be inferred from dynamical signatures passively recorded or actively probed in hippocampal circuits. Of importance for the design of future neurotechnologies, active probing surpassed passive recording to decode underlying levels of neural excitability, notably when assessed from a network of propagating neural responses. Our findings provide a promising approach for predicting and preventing seizures, based on a sound understanding of their dynamics.

Any brain can seize: from flies to fishes, and from mice to humans, epileptic seizures can strike without warning. In addition to the 70 million people with epilepsy worldwide who suffer from recurrent unprovoked seizures[1], up to 10% of the human population[2] will experience the danger of an occasional seizure (typically provoked) despite having a healthy brain. Thus, a fundamental question is how do seizures and physiological brain oscillations alternatively emerge from the same neural machinery?

From the standpoint of dynamical systems theory, seizures result from critical transitions in a bistable system (the brain) composed of alternating 'non-ictal' and 'ictal' (i.e., seizure) regimes[3–7]. In other words, seizures represent one possible regime of the brain, putatively a byproduct of its vital excitability. As the site of most human seizures[8], the hippocampus in particular may operate on the brink of instability[9,10]. Therein, positive neuronal feedbacks can amplify weak yet relevant inputs, but consequently increase the vulnerability to seizures[11]. At such tipping or 'critical' points, although the observable

[1]Center for experimental neurology, Sleep-wake epilepsy center, NeuroTec, Department of Neurology, Inselspital, University Hospital Bern, University of Bern, Bern, Switzerland. [2]Department of Neurosurgery, Inselspital, University Hospital Bern, University of Bern, Bern, Switzerland. [3]Department of Fundamental Neuroscience, University of Geneva, Geneva, Switzerland. [4]These authors jointly supervised this work: Antoine Adamantidis, Timothée Proix, Maxime O. Baud. ✉e-mail: maxime.baud.neuro@gmail.com

state of the system appears stable, inconspicuous variations that are difficult to foresee and oppose can precipitate critical transitions[6,12,13]. Developing practical methods to estimate neural *resilience* (i.e., the amount of perturbation sustainable without causing a seizure[14,15]) could help mitigate the risk of seizures.

Mathematically, critical transitions can only follow a limited number of defined bifurcation types, which capture the generic dynamical signatures of a system's possible trajectories under different conditions[7,12,16,17]. For example, in excitable circuits with *resonator* bifurcation properties, imposing stimulations at resonance frequencies may lead to dangerous neural amplification of weak inputs[16,18]. In contrast, for circuits with *integrator* bifurcation properties, no such risk exists, as long as the stimulation remains below a certain intensity threshold[16]. Which of these dynamics apply to specific neural circuits is being investigated[17,18], but a previously published model, known as the 'Epileptor', posited that integration of perturbations may represent a common scenario to provoke an ictal transition (i.e., seizure onset)[7]. This model depicts a *fold* bifurcation (visualized as an S-shaped diagram in Fig. 1C, D), where the brain is incrementally driven to a critical point as neural excitability rises. Consequently, *resilience* decreases[7,12], increasing the likelihood of an ictal transition, and *recovery* from subthreshold perturbations becomes slower[13,19]. Of practical importance, the resulting dynamical signature known as *critical slowing*[12,13,20,21] may be reflected in the changing statistics of longitudinal recordings, or in the growing impact of probing perturbations imposed on the system.

The decades-long search for such *warning signs* heralding ictal transitions in EEG of patients with epilepsy has challenged the best machine-learning algorithms[22,23] and frustratingly led to contradictory results in regards to the validity of the critical slowing hypothesis[15,24–27]. However, these studies were correlational by design and did not directly control, nor probe neural excitability, for example, with repeated small perturbations[28–31]. To date, experimental validation of these theoretical predictions was mostly obtained in vitro, from seizure-like events in brain slices kept in an artificial milieu[7,15,32], and rarely from the intact brain in vivo[32].

To fill this gap, we systematically tested the predictions of the Epileptor model in vivo. We characterized the critical neural dynamics in hippocampal circuits using targeted optogenetic stimulations in freely moving non-epileptic mice and electrical stimulations in hospitalized patients with epilepsy. Our findings highlight the superiority of actively probing brain networks as opposed to relying on passive dynamic signatures to assess underlying levels of neural excitability.

## Results

Phenomenologically, unprovoked and provoked seizures in the hippocampus can similarly produce a patient's stereotypical symptoms (Fig. 1A, B). On intracranial EEG (iEEG), these seizures share invariant features beyond their abrupt focal onset such as propagation and slowing discharges before an abrupt offset[7,33] (Fig. 1B). To generate testable hypotheses, we implemented the previously published Epileptor model[7] (Fig. 1C, D) that reproduces these invariant features of seizures in silico (Fig. 1E1) and provided detailed predictions on the dynamics of seizure onsets beyond those previously tested. First, we verified the nature of the bifurcation in vivo, probing hippocampal circuits with neurostimulation in non-epileptic freely-moving wildtype mice ($N = 17$, Fig. 1E–G2) and in patients with epilepsy undergoing a pre-surgical evaluation with iEEG for clinical reasons ($N = 10$, Fig. 1E–G3, Supplementary Table 1). This allowed us to establish robust, interpretable and translational means of measuring resilience to seizure (Fig. 1E) and neural excitability (Fig. 1F, G) and to verify their dynamical meaning in the model. Second, we assessed the correlation between resilience (Fig. 2), recovery rate from perturbations (Fig. 3) and passive dynamical signatures (Fig. 4) obtained from network recordings (Figs. 5 and 6) under pharmacologically controlled

conditions of excitability. Last, we compared the predictive value of active versus passive dynamical signatures to decode underlying neural excitability (Fig. 6) and herald ictal transitions in mice (Fig. 7).

## Dynamical model

The Epileptor model is formulated as a set of five differential equations that capture the dynamics of the interictal and ictal sequences in epilepsy into three interconnected subsystems ($\{x_1, y_1\}$, $\{x_2, y_2\}$, $\{z\}$, see methods)[7]. In this model, slow changes in excitability determine faster neural firing and its possible degeneration into a seizure upon crossing a critical point in a fold bifurcation. The resulting dynamics are summarized in a S-shaped bifurcation diagram (Fig. 1C, D and Supplementary Fig. 1). Therein, two basins of attraction (the two regimes) are separated by a divergent flow acting as a barrier[7,12]. The height of this seizure threshold determines a given level of resilience, here the distance from the trough to the crest (Fig. 1C, D). At the critical point (resilience→0, empty red dot in Fig. 1D), the threshold disappears and self-excitation due to positive feedback suffices to shift the system into the ictal regime consituting an unprovoked seizure. Alternatively, an external stimulation can push the system above the threshold into the ictal regime, constituting a provoked seizure (blue arrows in Fig. 1D). By design, the Epileptor bifurcation for seizure onset behaves as an integrator: it has no resonance frequency and seizures cannot occur upon inhibitory inputs[16].

## Metrics and their dynamical meaning

Like others[34,35], we probed resilience to ictal transitions using the time-to-seizure, that is the number of pulses or total duration of stimulation necessary to force an ictal transition (Fig. 1E). In the model, we found that the time-to-seizure tightly corresponds to the path-length traveled to reach the threshold in Epileptor's state-space for given stimulation parameters (double arrow in Fig. 1C), a distance also reflected in the deflections of the iEEG signal (Supplementary Fig. 2C–G). Furthermore, we inferred recovery rates from response magnitudes to single-pulse perturbations (measured as the line length of evoked iEEG potentials, Fig. 1F, G). In the model, we found that the measured line length reflects the length of the path traveled on an excursion within the 'non-seizure' basin of attraction (curved arrows in Fig. 1D, Supplementary Fig. 2A, B).

## Provoking seizures in mice

Confirming the existence of a latent seizure threshold in the healthy mouse brain ($N = 17$), we were able, like others[34,36,37], to induce seizures on the first trial of stimulation of hippocampal circuits (no need for kindling). We specifically targeted pyramidal neurons forming a positive feedback loop in the entorhinal–CA1 hippocampal circuit (temporo-ammonic branch)[38–40], using simple or intersectional transfections with adeno-associated viruses (AAV) carrying Channelrhodopsin 2 (ChR2, methods, Supplementary Fig. 3). Two months after the transfection, we triggered hippocampal seizures 'on demand' with trains of opto-pulses (pulse-width 3 ms, wavelength 473 nm, 20 Hz) over a median [range] duration of 5.0 s [1.5–25] (tested from 0.25 to 30 sec, Fig. 1E2, Supplementary Movie 1). This result confirms that synchronously driving pyramidal neurons in different nodes of hippocampal circuits (here tested in CA1 and layer III entorhinal cortex) can entrain a healthy brain into a seizure[34,36] (Supplementary Fig. 4).

## Provoking seizures in human participants

In a pre-surgical context, electrical stimulations are routinely applied directly to different cortical areas of patients with epilepsy undergoing iEEG to provoke seizures and study them[41]. Similar to results in healthy mice, we found that seizures were inducible in the human hippocampus with train stimulations (bipolar, biphasic pulse-width 1 ms, 1–5 mA, 60 Hz) over a median [range] duration of 3.0 s [1.0–4.0] ($N = 7$, Fig. 1A, B, Supplementary Table 1). In a subset of participants tested

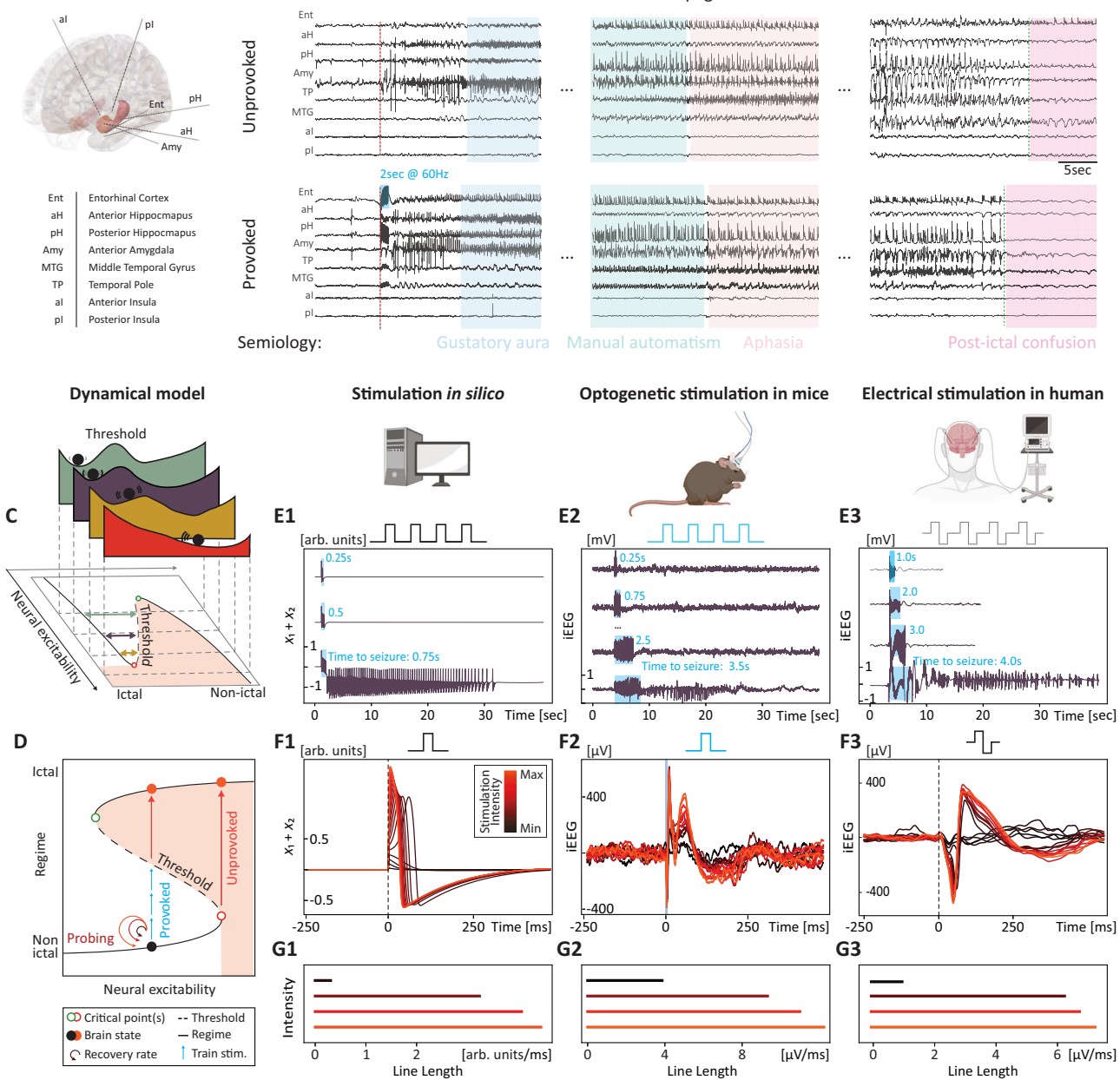

**Fig. 1 | Probing seizure resilience and neural excitability. A** Intracranial EEG (iEEG) electrodes implanted in a patient with epilepsy undergoing pre-surgical evaluation. The hippocampus is highlighted in red, the amygdala in orange. **B** Corresponding examples of iEEG signals from one unprovoked (spontaneous) and one provoked seizure (electrical stimulation in the entorhinal cortex, 2 s at 60 Hz), recorded for clinical localization of the seizure onset zone. The provoked seizure recapitulates most of the electrographical and semiological characteristics of the unprovoked seizure. **C** The 'Epileptor' models the brain as a bistable dynamical system. Therein, the brain can be represented as stability landscapes (top, green to red), where the ictal (light red shading in the lower pannel) and non-ictal regimes (white background) form basins of attraction, separated by a threshold (dashed line). Bidirectional changes in neural excitability (the system's control parameter) modulate the stability landscape with lower (yellow) or heightened resilience (green double arrow). When excitability reaches the critical point (empty red dot), the non-ictal regime disappears and the system is forced to transition into the ictal regime. **D** For a given level of neural excitability (black ball), resilience to seizure can be measured as the amount of external perturbation (here, stimulations, blue arrows) necessary to cross the threshold and transition to the seizure regime (provoked seizure). When the system is close to the critical point (empty red dot, resilience tends to zero), ictal transitions can occur in the absence of

external excitation (unprovoked seizure). **E1–3** Experimentally, trains of stimulation of increasing duration were used to measure seizure resilience. Example in silico using the Epileptor model (**E1**, 20 Hz, 3 ms pulse-width), in mice Using optogenetic stimulation on the entorhinal cortex in mice (**E2**, 20 Hz, 3 ms pulse-width on Channelrhodopsin-transfected pyramidal neurons) and in humans using electrical stimulation on the entorhinal cortex (**E3**, 60 Hz, 1 ms pulse-width). The time of train stimulation necessary to provoke a seizure is indicative of the distance to the seizure threshold. **F1–3** Smaller perturbations (here, single pulses) can be used to probe neural excitability without inducing a seizure. Dynamically, they correspond to an excursion contained within the non-ictal regime (circle arrows in **D**, see Supplementary Fig. 2). Experimentally, to probe the dynamic range of physiological neural excitability, we used a range of single-pulse stimulations of varying intensity (dark to bright red) leading to increasing responses in silico (**F1**, 3 ms pulse), as well as in vivo in mice (**F2**, iEEG response to single 3 ms laser pulses between -0–50 mW) and human patients (**F3**, iEEG response to single 1 ms electrical pulses between 0.2–10 mA). **G1–3** For a given perturbation, the induced response can be quantified as the line length of the iEEG signal over a 250 ms window, which intuitively reflects the excursion length in state-space (Supplementary Fig. 2). **E1–3** created with BioRender.com released under a Creative Commons Attribution-NonCommercial-NoDerivs 4.0 International license https://creativecommons.org/licenses/by-nc-nd/4.0/deed.en.

more than once in the same hippocampus, the time-to-seizure remained identical, suggesting the presence of a fixed seizure threshold (participants 11 and 15 in Supplementary Table 1).

## Integrative dynamics in the hippocampus

By varying stimulation parameters, we were able to confirm the integrative dynamics of the bifurcation in hippocampal circuits. Experimentally in mice, the time-to-seizure decreased as a function of increasing stimulation frequency onto excitatory pyramidal neurons, from a median of 27 sec at 4 Hz to a median of 8 sec at 40 Hz ($N = 8$, Supplementary Fig. 5C2). Also, arhythmic opto-stimulations at an average rate of 20 per second were as efficient as rhythmic stimulation at exactly 20 Hz in inducing hippocampal seizures ($N = 9$, Supplementary Fig. 5C4, F). Further, in vitro studies have indicated that inhibitory interneurons may play a role in seizure initiation[42–45]. In a supplementary experiment, we transfected transgenic mice (PV_Cre), with an AAV virus to specifically express a fast Channelrhodopsin (ChETA)[46] in PV-interneurons of the CA1 dorsal hippocampus ($N = 4$, Supplementary Fig. 3G, H). In contrast to our manipulations on pyramidal neurons, stimulation of local PV-interneurons in the same circuit entrained brain oscillations at the same and higher frequencies but did not result in seizures (Supplementary Fig. 5C3, D3). Taken together these results suggest the lack of critical sensitivity to a resonance frequency in the hippocampus. Rather, hippocampal circuits integrate successive excitatory perturbations until a threshold is met (i.e., resilience is overcome), consistent with the physiological presence of a fold bifurcation[16] consistutive of the healthybrain.

## Probing variable neural resilience

In a fold bifurcation, resilience is directly proportional to the distance to the critical point and can be measured as the minimal perturbation sufficient to provoke an ictal transition (Fig. 1C, D, Fig. 2A). For experimental measurement of varying resilience in mice and humans, we used pharmacological manipulations of GABA-A receptor-mediated synaptic inhibition which are known to modulate neural

excitability[47,48]. These included agonists such as benzodiazepines (BZD) given intraperitoneally in mice (diazepam 5 mg/kg) or intravenously in humans (clonazepam 0.5–1 mg) or the antagonist Pentylenetetrazole (PTZ) given at subconvulsive doses intraperitoneally in mice (10–20 mg/kg). Similarly to simulations (Fig. 2B, Supplementary Fig. 2G), the time-to-seizure significantly increased when excitability decreased (BZD, percent change compared to control injection with NaCl i.p. [bootstrapped 95% CI]: +97% [+72, +140], 40 sessions among $N = 17$ mice, Fig. 2C) and decreased when excitability increased (PTZ, −17% [−8, −28], 23 sessions among $N = 9$ mice, Fig. 2C). These effects were dose-dependent (Fig. 2C3) but independent of the stimulation frequency (Supplementary Fig. 6D). In the human cohort, PTZ was not administered (no clinical value), but one participant had a seizure provoked before and immediately after the administration of BZD, allowing for a direct comparison of resilience. In line with the results obtained in mice, the time to seizure increased from 2 s at baseline to 4 s with BZD (+100%, $N = 1$, Fig. 2D). Confirming the model's prediction, this set of results emphasizes the rapidity (seconds) and spatio-temporal precision with which varying seizure thresholds can be measured using neurostimulation in vivo.

## Probing variable neural excitability

Theoretically, resilience and recovery covary with the topology of the basin of attraction. A deeper and steeper basin increases both resilience (c.f. height of the hill in Fig. 3A2) and recovery rates (c.f. steepness of the slope in Fig. 3A2)[12]. Conversely, in a flat basin with vanishing borders, slower recovery (i.e. critical slowing) is expected[12]. By measuring the line length[15,49] of the iEEG response evoked upon perturbation, we found that recovery rates were indeed inversely related to pharmacologically controlled higher or lower levels of neural excitability (Fig. 3B1–D1 and Fig. 3C3). To capture the dynamic range of responses (line length), an input-output curve (IOC) was generated for single-pulse stimulations of increasing intensity (Fig. 3B2–D2). The area under this curve (thereafter simply IOC) captures in one value (from zero: no response at any intensity, to one: maximal response at minimal intensity) a given

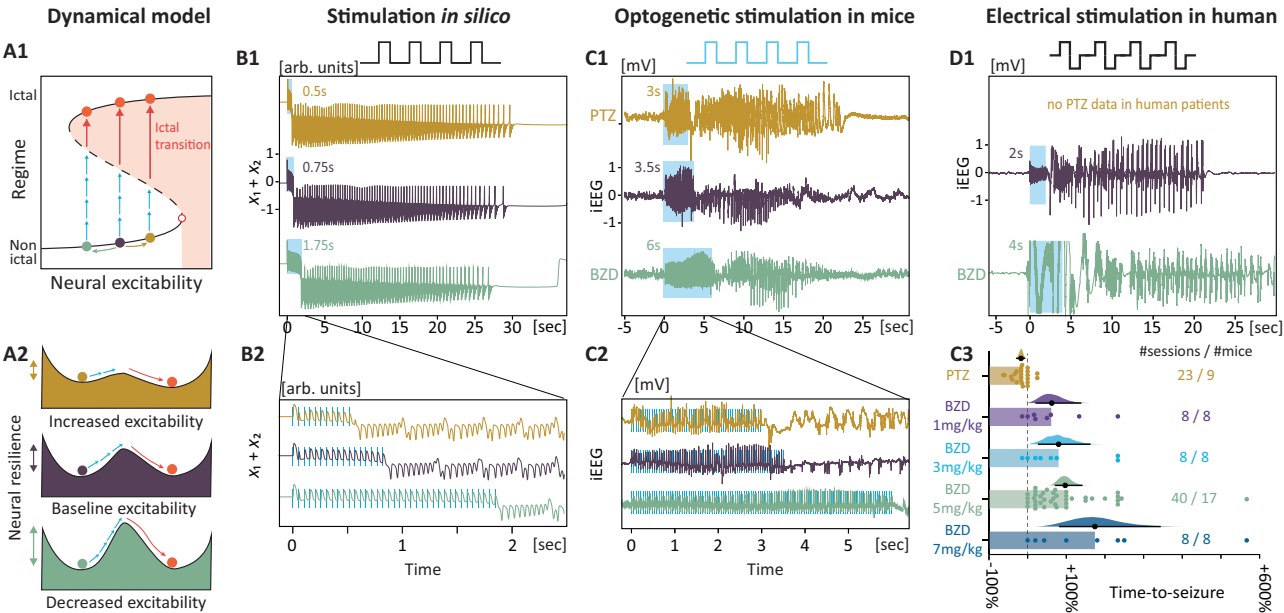

**Fig. 2 | Probing variable neural resilience.** Using the time-to-seizure, neural resilience can be probed bidirectionally in conditions of higher and lower levels of neural excitability. **A1–A2** Prediction that bidirectional changes in neural excitability lead to measurable changes in resilience to seizure (number of pulses, blue arrows). **B1** The prediction was modelled in silico by measuring the time-to-seizure upon changing the control parameter ($x_0$) in the Epileptor model. **C1–D1** The

prediction was tested in vivo, by measuring the time-to-seizure in the presence of an agonist (benzodiazepine, BZD) or an antagonist (Pentylenetetrazol, PTZ) of the GABA-A receptor. **B2–C2** Zoom in on the stimulation train leading to the onset of a seizure. **C3** Average changes in time-to-seizure (mean ± bootstrapped 95% CI difference to the NaCl control session) with increasing doses of BZD and a subconvulsive dose of PTZ across mice and sessions.

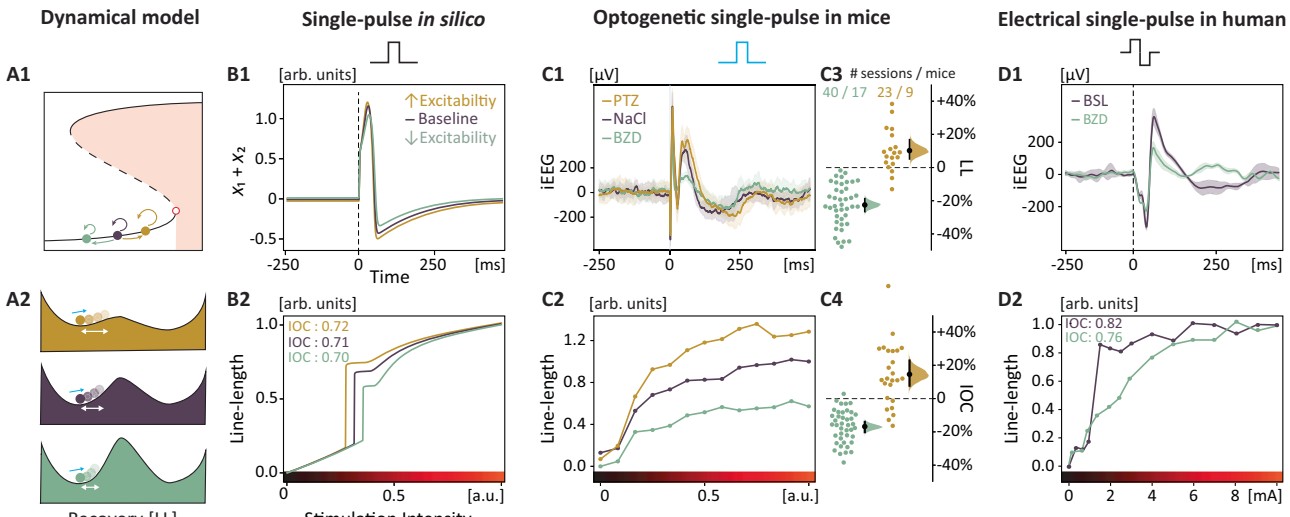

**Fig. 3 | Probing variable neural excitability.** Using the line length of evoked responses, recovery can be probed bidirectionally in conditions of higher and lower levels of neural excitability. **A1–A2** Prediction that bidirectional changes in neural excitability lead to different recovery rates (line length, circle arrows in **A1**, white double arrow in **A2**) upon the same perturbation (single-pulse, blue arrow in A2). **B1–D1** Modulation of excitability (same as in Fig. 2) results in detectable changes in iEEG responses and their recovery to single-pulse stimulation in silico (**B1**, response at half-maximum intensity), in mice (**C1**, iEEG response at maximum intensity), and humans (**D1**, iEEG response at 3 mA). Thick lines are the mean across sessions, shading the standard deviation. **B2–D2** Measured iEEG responses (line length,

'output') to a range of single-pulse stimulations of varying intensity ('input') resulting in an 'input-output curve' under variable excitability in silico (**B2**), in mice (**C2**, $N = 9$–17) and in one participant (**D2**, group result in Fig. 5). **C3** Magnitude of the iEEG responses across mice, quantified as the line length of the signal over a 250 ms window and normalized to the baseline (NaCl). Half-violin plot shows mean differences with bootstrapped 95% CI across respectively 23 (PTZ) or 40 (BZD) sessions among 9 or 17 mice. **C4** Quantification of the area under the input-output curve (IOC) as a function of changes in excitability. Half-violin plot shows mean differences with bootstrapped 95% CI across respectively 23 (PTZ) or 40 (BZD) sessions among 9 or 17 mice.

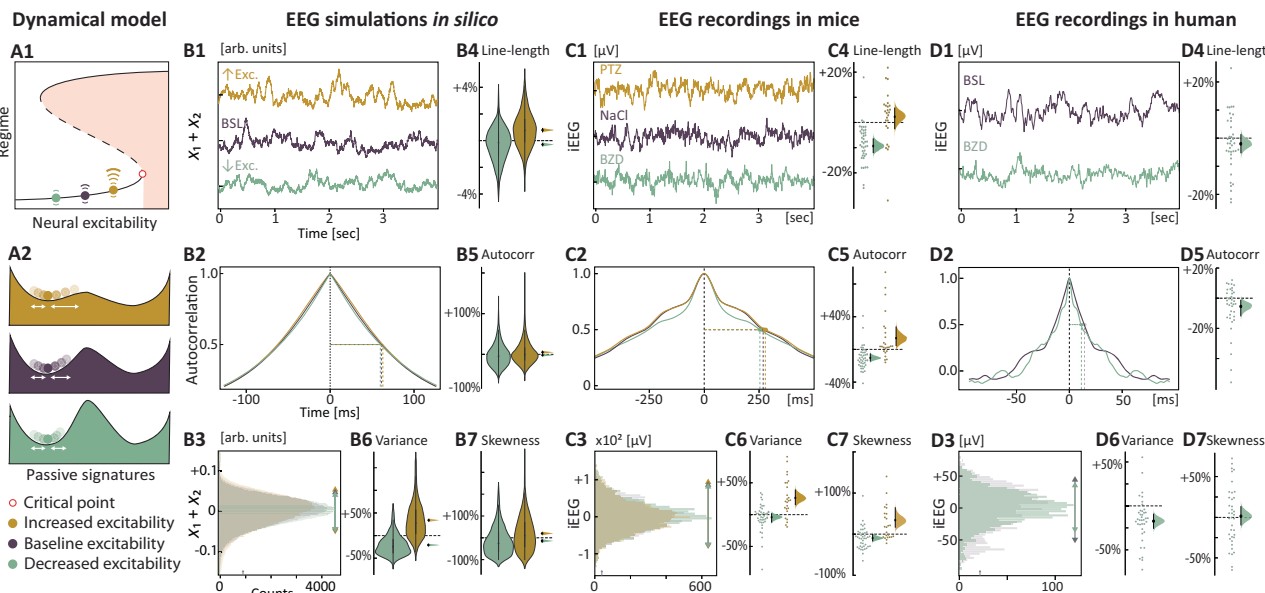

**Fig. 4 | Passive signatures of neural excitability.** Passive dynamical signatures are found in bidirectional changes in univariate statistics (line length, autocorrelation, variance, and skewness) in conditions of higher and lower levels of neural excitability. **A1–A2** Prediction that endogenous stochastic perturbations result in trajectories of increasing length and asymmetry with increasing excitability. **B1–D1** Example of recorded signals (4 s) in the absence of stimulation at different levels of excitability in silico, (**B1**, $x_1 + x_2$ in the Epileptor in presence of stochastic noise) as well as in vivo from iEEG in the hippocampus in mice (**C1**) and participants (**D1**). **B2–D2** Corresponding autocorrelation function of the signal with calculation of the

lag value at half-maximum. **B3–D3** Corresponding histogram of the signal's values across conditions. **B4–B7** For each passive signature, quantification of the changes when excitability is varied, expressed in difference to baseline condition and calculated on 790 simulated iEEG samples. Half-violin plot on the right shows mean differences with bootstrapped 95% CI. **C4–C7** Mean (±bootstrapped 95% CI) change in passive signatures in presence of BZD ($N = 17$) or PTZ ($N = 9$) across 103 sessions in mice, normalized to the control session (NaCl). **D4–D7** Mean (±bootstrapped 95% CI) change in passive makers in presence of BZD across 36 hippocampal electrodes in 6 participants.

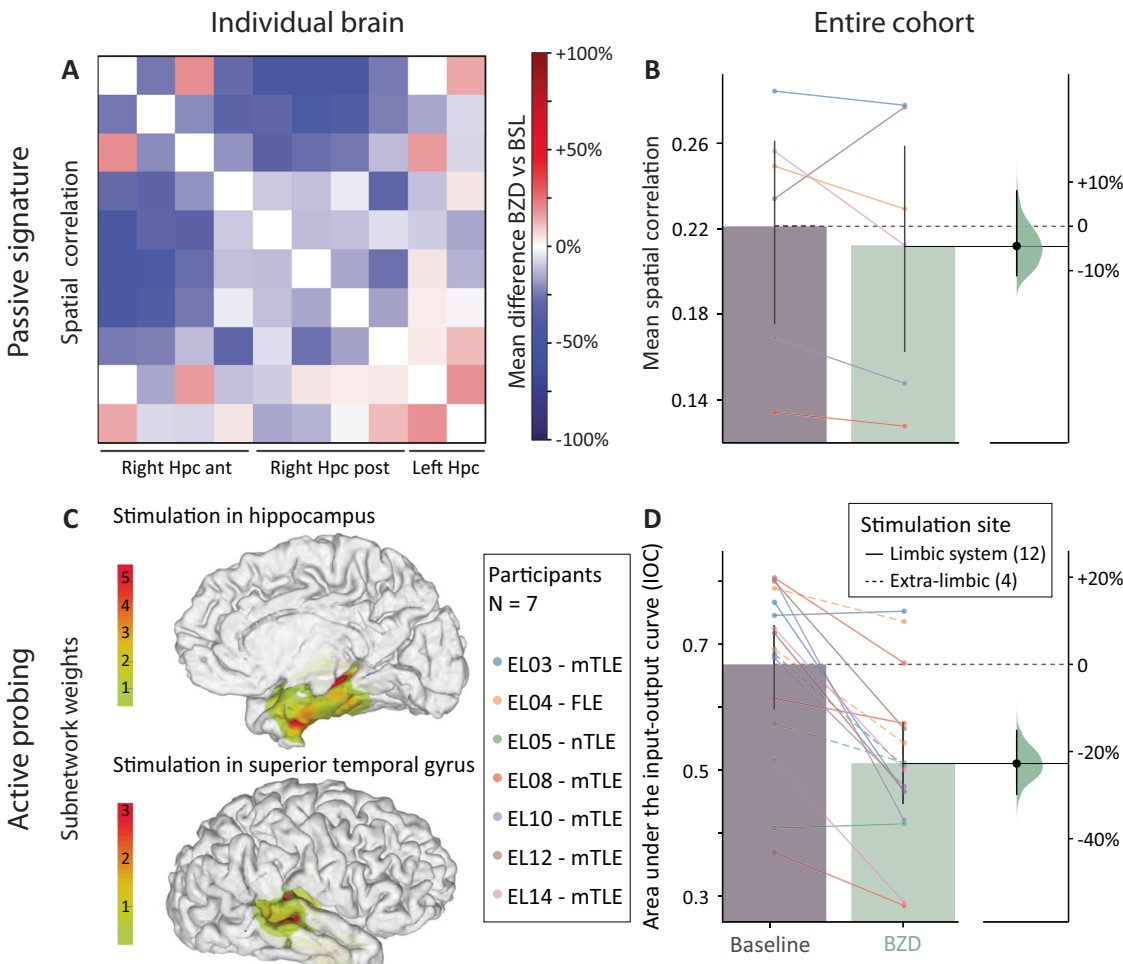

**Fig. 5 | Network dynamics in humans.** Probing network responses to single-pulse stimulations uncovers changes in network dynamics. **A** For a representative participant (EL014), matrix of mean difference in Pearson correlation between all pairs of hippocampal electrodes compared to the control condition (baseline pre-BZD i.v.). **B** Mean (± bootstrapped 95% CI) spatial correlation across participants (*n* = 6). **C** In a representative participant, two examples of sub-networks responding to stimulation, respectively in the right hippocampus and the right superior temporal gyrus, identified by NMF and projected to the cortical surface. **D** Mean (± bootstrapped 95% CI) decrease in excitability with BZD across 16 sub-networks identified among 7 participants (colored dots). Full lines are for stimulation sites in the hippocampal circuits (limbic), dashed lines for stimulation in the extra-limbic neocortex. Supplementary Table 1 shows further clinical characteristics of the participants. mTLE and nTLE mesio and neocortical temporal lobe epilepsy, FLE frontal lobe epilepsy.

excitability level, irrespective of changes in slope (e.g., Fig. 3D2) or height (e.g. Fig. 3C2). Similarly to simulations (Fig. 3B), evoked iEEG responses decreased with decreased excitability in mice (BZD, IOC −17% 95% CI [−14, −20], 40 sessions among *N* = 17 mice, Fig. 3C4) as well as in participants (BZD, −7%, Fig. 3D2, *N* = 1, group analysis below), and increased with increased excitability in mice (PTZ, +15% [+8, +23], 23 sessions among *N* = 9 mice, Fig. 3C4). For BZD, these effects were dose-dependent and already present at low doses (Supplementary Fig. 7). In contrast to other studies probing excitability[28,31,32], we obtained these results in controlled pharmacological conditions, confirming that variable degrees of underlying neural excitability can be assessed without provoking a seizure. As opposed to the method developed by Klorig et al.[34], we did not rely on the response probability to estimate excitability, which drastically reduced the number of probing pulses needed.

**Passive signatures of neural excitability**
We further found that changes in the system's recovery rates were also reflected in simple passive statistics of longitudinal iEEG recordings, including variance, skewness, line length, and autocorrelation of the signal, in line with prior publications[15,26,27]. Theoretically, a complex system is never completely steady but rather orbits with varying

excursion lengths within a basin of attraction, being constantly subjected to internally generated stochastic perturbations (i.e., 'noise', Fig. 4A). Thus, when resilience decreases, dynamical signatures may reflect slower recovery from these endogenous perturbations[12,13]. When adding stochastic noise to the Epileptor at high and low excitability levels, we found in silico the predicted changes in statistics with variable effect size (Fig. 4B). Decreased excitability resulted in decreased line length (−0.3% [−0.4, −0.1]), variance (−21% [−23, −19]), skewness (−14% [−20, −8]) and autocorrelation (−2% [−5, +1]). Conversely, increased excitability resulted in increased line length (+0.8% [+0.7, +0.9]), variance (+34% [+31, +37]), skewness (+20% [+14, +27]), and autocorrelation (+5% [+2, +8]), which are known indirect signs of critical slowing[15,26,27]. In the mouse hippocampus, iEEG line length (−9% [−12, −6]) and autocorrelation (−10% [−14, −5]) significantly decreased with BZD (40 sessions among *N* = 17 mice), whereas variance (+26% [+16, +39]), autocorrelation (+14% [+5, +27]) and skewness (+34% [+17, +64]) significantly increased with PTZ (23 sessions among *N* = 9 mice, Fig. 4C). In participants, i.v. BZD administration decreased variance (−16% [−8,−25]) and autocorrelation (−6% [−12, −1]) in the hippocampal iEEG but other signatures did not show any significant changes (*N* = 6 participants, Fig. 4D). Thus, passive dynamical signatures observed

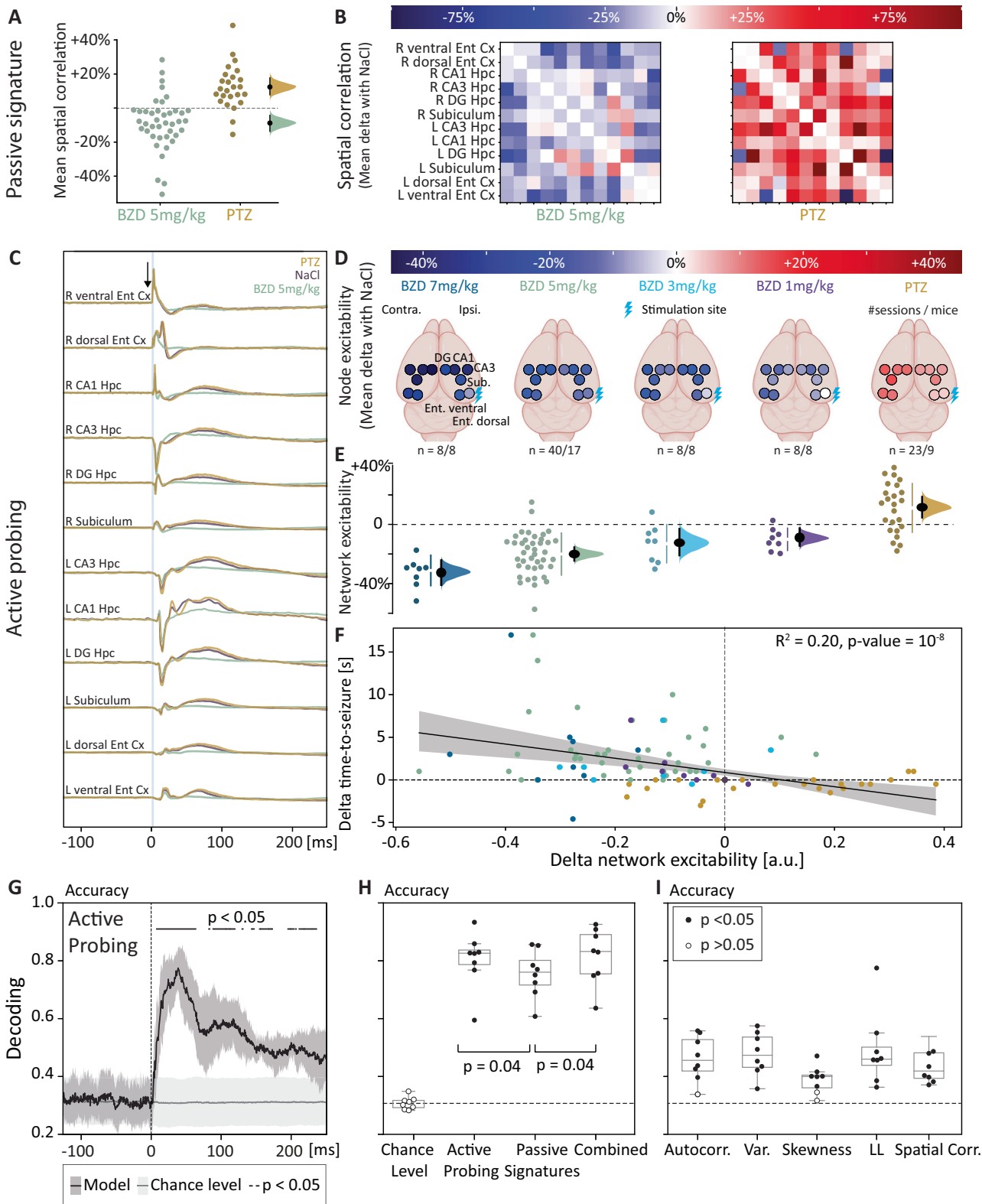

in vivo under controlled pharmacological conditions were always in the expected direction but not always significant, highlighting their equivocal value.

## Network dynamics in humans

Theoretically, changes in resilience and recovery rate can also become apparent from reverberations across connected nodes of a network, another *multivariate* (versus univariate, Figs. 2–4) dynamical signature[12,13]. Yet, we found no significant change in the average spatial correlation in passive recordings from hippocampal electrodes across connections and participants receiving a BZD injection ($N = 7$, Fig. 5A, B, mean difference with baseline and bootstrapped 95% CI −4% [−11, +1]). We thus asked whether active probing could uncover dynamical changes in hippocampal circuits. We used the unsupervised clustering algorithm non-negative matrix factorization[50,51] (NMF, Fig. 5C, D, Supplementary Fig. 8) to capture iEEG responses shared in a

**Fig. 6 | Decoding network dynamics in mice.** Probing network responses to single-pulse stimulations highlights changes in network dynamics and enables better decoding of the underlying level of neural excitability. **A** Mean (±bootstrapped 95% CI) difference in Pearson correlation between all pairs of electrodes compared to the control condition (NaCl) across 103 sessions among 17 mice. **B** Representative example in one mouse of the GABAergic modulation of the average normalized spatial correlation between all recording electrodes (no probing). **C** Representative example in one mouse of the GABAergic modulation of the average iEEG response to optogenetic single-pulse stimulation (active probing, down-pointing arrow in the right entorhinal cortex) and ensuing propagation across hippocampal circuits. **D** Mean GABAergic modulation of response to single-pulse stimulation at maximal intensity in each recording channel (dots) across mice ($N = 17$), compared to the NaCl condition after bootstrapping (values within 95% confidence intervals were left blank). **E** Changes in IOC (±bootstrapped 95% CI) compared to the control condition (NaCl) for conjoint network responses to stimulation computed across electrodes, stimulation intensities, and pharmacological conditions using NMF (see Supplementary Fig. 8). Half-violin plot shows mean differences with bootstrapped 95% CI, $N$ are reported in the figure. **F** Differences in resilience inversely correlates with differences in single-pulse network responses in the same session. The thick line shows the linear regression and shading the 95% CI. R2 is Pearson's correlation coefficient and $p$ the two-sided $p$ value. **G–I** Average (±SD, $N = 8$) accuracy (unseen test data) and comparison of different single-trial multilabel classifiers (three balanced excitability levels: low, normal, high) based on the raw iEEG response to single pulses (active probing, 0–0.25 s, **G**, **H**) or multisite iEEG passive recordings (multiple (**H**) or single (**I**) passive signatures, 4 s as in Fig. 4) or the combination of these features (combined, **H**). Mean ± SD chance-level shown in gray (100 label one-sided permutations test, see methods) with significant timepoints as horizontal black bars. Each dot ($N = 8$) corresponds to one mouse that received both BZD and PTZ in different sessions, filled if significant ($p < 0.05$, one-sided permutations test, see methods). * shows significant differences between classifiers ($p < 0.05$, two-sided paired Wilcoxon rank-test). **D** created with BioRender.com, released under a Creative Commons Attribution-NonCommercial-NoDerivs 4.0 International license https://creativecommons.org/licenses/by-nc-nd/4.0/deed.en.

sub-network connected to an electrode undergoing single-pulse probing over a range of intensities (input-output curves as in Fig. 3). Across seven participants and 16 stimulation sites (15 intensities per site, up to 80 recording channels per participant), probed sub-network responses showed a significant decrease after BZD (IOC −23% [−15, −30], $N = 7$, Fig. 5D). This result suggests that active probing can uncover changes in brain network dynamics not visible in passive recordings.

## Network dynamics in mice

In mice, spatial correlation measured in iEEG passively sampled from nodes of the hippocampal circuits (total of 12 electrodes bilaterally in the hippocampus, subiculum and entorhinal cortex) slightly decreased with BZD (−8% [−3,−13] compared to NaCl, 40 sessions among $N = 17$ mice) and slightly increased with PTZ (+12% [+7, +15], 23 sessions among $N = 9$ mice, Fig. 6A, B). As above, we used active probing and NMF to capture more sensitively shared dynamics across the network at different stimulation intensities (Fig. 6C–E, Supplementary Fig. 8). We found a decrease in the network IOC with BZD (7 mg/kg: −31% [−39, −25], BZD 5 mg/kg: −18% [−22,−14], BZD 3 mg/kg: −13% [−21,−4], BZD 1 mg/kg: −9% [−13,−3]) and an increase with PTZ (+11% [+5, +18], Fig. 6E). Importantly, bidirectional and dose-dependent differences in seizure resilience were negatively correlated with network responses probed in the same session (Pearson correlation: $R^2 = 0.20$, $p < 10^{-6}$, 87 sessions among $N = 17$ mice, Fig. 6F, Supplementary Fig. 9A). This result links recovery rates to resilience at the network-level and over a broad range of underlying excitabilities, confirming their dynamical relationship in a fold bifurcation and supporting the idea of probing the brain to assess its resilience.

## Decoding network dynamics in mice

Next, we formally assessed the superiority of a probing strategy in decoding momentary states of excitability over the poorer predictive value of partially correlated passive dynamical signatures ($R^2$ ranging from 0.09 to 0.29, Supplementary Fig. 9B). To do so, we adopted a machine-learning approach to formally assess and compare the single-trial predictive value of short segments of probed (0.25 s) and passively recorded iEEG (4 s). We trained a multilabel logistic regression to classify states of low, normal, and high excitability corresponding to the three balanced pharmacological conditions (one-third each: BZD vs. NaCl vs. PTZ). To evaluate and compare the performance of each classifier, we calculated the accuracy of this three-label classification (see methods), which can be directly interpreted as the percentage of single trials that were correctly classified (chance-level accuracy -0.33, obtained for each mouse by training on shuffled labels). An *active probing* classifier trained on the multichannel iEEG response to single pulses over 0.25 s without feature extraction performed well above chance level (median accuracy = 0.83, $p < 0.01$, $N = 8$, Fig. 6G, H), suggesting that important information could be extracted from the voltage waveform recorded across 12 channels (Fig. 6C). Multiple classifiers trained post-hoc on single timepoints confirmed the absence of predictability from the raw baseline iEEG (Fig. 6G, recorded voltage from −100 to 0 ms), peak predictability between 20–50 ms and sustained above-chance-level for at least 250 ms after stimulation. In the absence of stimulation, single-feature classifiers performed poorly, when trained on individual passive signatures drawn from 4 s baseline segments for each iEEG channel (Accuracy -0.5, Fig. 6I). However, a multi-feature *passive signatures* classifier combining these individual features had good performance (Accuracy = 0.76, $p < 0.01$, $N = 8$, Fig. 6H), suggesting that no single passive signature fully reflects neural excitability. It was nevertheless inferior to active probing and a combined classifier taking all inputs ($p = 0.04$, two-sided paired Wilcoxon rank-test, $N = 8$, Fig. 6H) that had similar performance among them. These machine-learning results highlight the decoding value of active probing over that of partially correlated passive dynamical signatures.

## Warning signs of ictal transitions

In a final mouse experiment, we showed how active probing could anticipate a PTZ-induced ictal transition, as done by Graham et al. for 4-aminopyridine-induced seizures[32]. All previous experiments characterized the dynamics of the hippocampus at some distance from the critical point, where an external provoking perturbation is necessary to cross the seizure threshold (green, grey, and yellow landscape in Fig. 1C). However, for even higher levels of excitability, the system approaches and crosses the critical point beyond which a seizure is inevitable even in the absence of provoking perturbations[7,12,15] (Fig. 7A). Such an ictal transition can be modeled with Epileptor, as in its original use, by imposing higher values on the epileptogenicity parameter $x_0$[7]. With rising excitability, serial probing leads to ever-larger responses, the expected dynamical signature of approaching a critical point, now representing a true 'warning sign'[13,20] (Fig. 7B, C and Supplementary Fig. 10A1). Such gradual loss of resilience has been thoroughly characterized by imposing artificial ionic concentrations onto brain slices, which results in repeating 'ictogenic ramps' that invariably lead to seizure-like events[15,32]. To verify these dynamical predictions in vivo, we injected 8 mice with convulsive doses of PTZ (25–35 mg/kg) and probed excitability with repeated single pulses every 8–12 s until a seizure occurred (Fig. 7D). For a fixed probing pulse, we found that iEEG responses slightly increased up to 10 min before the ictal transition (18 sessions among $N = 8$ mice, Fig. 7E–G and Supplementary Fig. 10B1), similarly to our experiments with subconvulsive PTZ (Fig. 6). In the last four minutes before the ictal transition though, single pulses now triggered large epileptiform responses across the

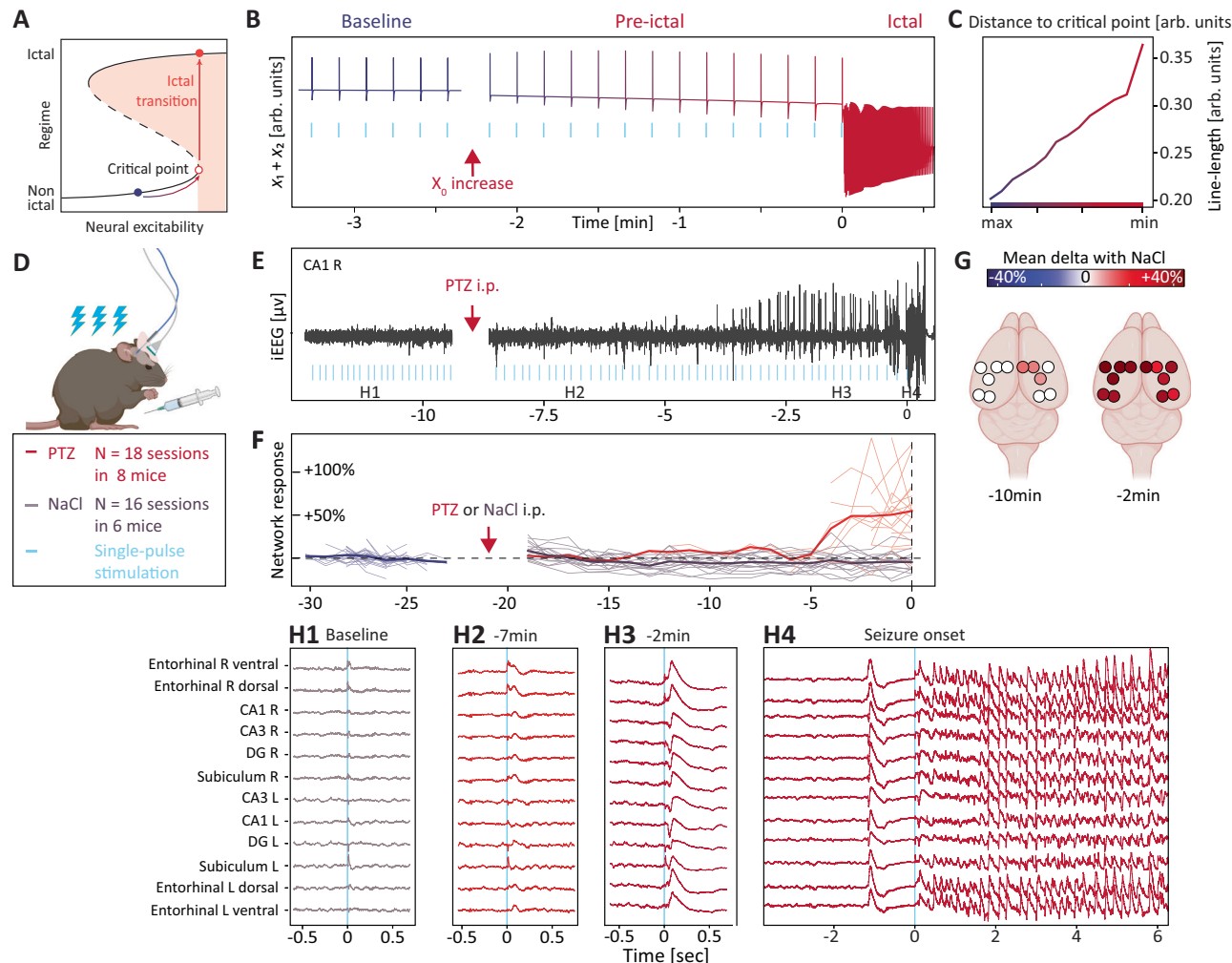

**Fig. 7 | Warning signs of ictal transitions.** The system's response gradually increases for the same probing stimulus at the approach of the critical point. **A** Prediction that a progressive increase in neural excitability results in a loss of resilience close to the critical point, where minimal perturbations can tip the system into an ictal transition. **B** In silico probing with single-pulse stimulations during an increase in excitability up to the critical point. In chronological order, early stimulation pulses (cyan tick) yield habitual responses (dark blue), whereas later pulses yield exaggerated responses (redder colors), until the last pulse provokes a seizure. **C** Corresponding responses to single-pulse stimulation, quantified as line length as a function of the distance to the critical point, measured in state space. **D** In vivo ($N = 8$) single-pulse stimulations during increasing excitability up to the critical point over the 5–20 min following the injection of a single convulsive dose of PTZ

(25–35 mg/kg). **E** Example of single-pulse stimulation (cyan ticks, every 8–12 s) and recording in the right CA1 hippocampus at baseline and after injection of PTZ. **F** Individual (thin lines) and average (thick line) NMF coefficients (see methods) of the network response to single pulses with increasing excitability (PTZ, red) and in control conditions (NaCl, gray). **G** Mean differences in single-pulse responses (line length, left blank if non-significant after bootstrapping) compared to baseline for each electrode 10 and 2 min before seizure, across 18 sessions among $N = 8$ mice. **H** Zoom in on the network response to single-pulse at baseline (**H1**) and increasing levels of excitability (**H2** to **H4**). In this example, the last pulse induces a seizure but seizures could also start between simulations. **D** and **G** created with BioRender.com released under a Creative Commons Attribution-NonCommercial-NoDerivs 4.0 International license https://creativecommons.org/licenses/by-nc-nd/4.0/deed.en.

network (+54% [+36, +70], Fig. 7H3), mimicking spontaneous epileptic spikes that also appeared in the recording (Fig. 7H4, Supplementary Fig. 10B6) and reminiscent of results in Graham et al.[32]. At this point, seizures could be provoked by a single-pulse (6 out of 18 seizures, Fig. 7H4) or start spontaneously between stimulations (12 out of 18 seizures), confirming that resilience had vanished.

## Discussion

In this study, we verified how fundamental mathematical predictions on fold bifurcations apply to hippocampal seizures in mice and humans in vivo, powerfully expanding prior experimental evidence. Beyond previous work on passive dynamical signatures[15,24–27], our study also provides a robust experimental and clinical framework to actively gauge *excitability* and *seizure thresholds* in brain networks adding mathematical formalism[16,21] to terms that are sometimes ambiguous in epileptology. The importance of such measurements is

to be found in their ability to reflect the risk of upcoming seizures. By actively probing hippocampal circuits, we uncovered dynamical signatures of critical slowing that can serve as warnings about imminent ictal transitions. The key contributions of this study are further specified below.

First, in contrast to previous experiments mostly in vitro[7], we characterized a circuit-specific bifurcation in vivo, using precise probing tools, namely optogenetics[34,36] in freely-moving mice and targeted electrical stimulation in hospitalized participants[18,41]. These direct brain stimulations applied over a range of pharmacologically controlled excitability levels, invariably led to self-sustained and self-terminating seizures, revealing the presence of a latent ictal regime[5,7,11]. More specifically, the presence of integrative dynamics, where lower or higher frequency inputs cumulate at different rates up to a threshold, suggests that hippocampal circuits are intrinsically organized around an integrator type of bifurcation[16] (here a fold

bifurcation), possibly explaining how hippocampal circuits can easily generate seizures. Beyond passive observations, our experimentation with resilience and recovery in vivo yielded unambiguous evidence in support of the pre-existing Epileptor model and highlighted how provoking seizures can help characterize a latent bifurcation. Importantly, knowing the bifurcation type helps define safe versus potentially seizure-provoking stimulation parameters[16,18].

Second, we found and measured varying resilience (i.e., the distance to the seizure threshold) when controlling excitability through bidirectional GABA-A receptor pharmacology, well established to either prevent or promote seizures[47,48]. Agonists (BZD) showed dose-dependent increases in resilience, whereas antagonists (subconvulsive PTZ) decreased resilience at the approach of the critical point. The rapidity and spatio-temporal precision with which neurostimulation can measure seizure thresholds within individuals offers a refined approach to drug screening in epilepsy that contrasts with commonly used but often criticized rudimentary approaches such as maximal electroshocks or PTZ seizure tests[52,53].

Third, we found that the speed of recovery from subthreshold perturbations reflected varying degrees of underlying excitability and correlated with resilience. For example, at the lowest excitability level (BZD in non-epileptic mice), recovery was fastest (probed iEEG responses were shorter) and resilience highest. Intuitively, as the distance to the threshold increases, so too does the slope of the basin of attraction, linking high resilience with fast recovery and narrower variations in signals through the geometry of the fold bifurcation[12,13] (Fig. 1C). Conversely, at the highest levels of excitability, on the verge of seizing (convulsive PTZ), recovery from perturbations sharply slows down. Such pre-ictal 'step-change' in recovery rates was also recorded by others in brain slices exposed to penicillin[54], high potassium[15], low magnesium[32] or 4-aminopyridine[32] as well as in vivo, upon cortical injection of 4-aminopyridine[32]. In the minutes preceding the onset of seizures in slices and in vivo, Graham et al. observed the appearance of prolonged potentials in dendrites, reflecting increased calcium entry upon optogenetic probing[32]. These dendritic 'plateau potentials' were associated with increased neuronal firing rates and prolonged recovery in the evoked cortical response. Thus, different modes of inducing seizures in vitro and in vivo share the same pre-ictal dynamical signatures - critical slowing - a possibly universal phenomenon[12,13,19-21] that may relate to specific neuronal (or dendritic) mechanisms in epilepsy. The interest and relevance of dynamical signatures lie in their phenomenological nature, reflecting governing rules of a system's dynamics that do not depend upon a detailed understanding of biophysical mechanisms and can likely be measured at a range of spatial scales from dendrites to circuits[27,33].

Fourth, our machine-learning results showed that snap-shots of actively probed signals (250 ms) more reliably uncovered underlying levels of excitability with ~80% accuracy, over passive signatures of critical slowing (increased line length, variance, skewness, autocorrelation and spatial correlation) which had here ambiguous predictive value, and have yielded conflicting results in the literature[15,24-27]. Unlike others using signal averaging over longer 10-min recordings in sleep and wake[55], we found weak correlations between passive and active dynamical signatures at a shorter timescale in the awake brain, a practical result in line with theoretical predictions[29,30,56]. Despite the growing use of head-implanted devices, actively probing the brain to assess recovery dynamics has so far not reached practice[26,53]. Yet, coupled with ever more sophisticated machine-learning, active probing could help assess momentary neural resilience, enabling real-time forecasts of seizure risk or timely therapeutic adjustments in patients with epilepsy[28,31]. More broadly, such a principled dynamical approach may be core to the development of next-generation implantable neurostimulators to treat psychiatric and neurological brain disorders[57-59]. Indeed, current empirical stimulation protocols are ignorant of the specific dynamics of targeted neural circuits, blind to potential risks, and likely do not fully leverage potential therapeutic opportunities[58-62]. Thus, an AI-assisted device that could probe and control the excitability of specific brain circuits may find broad applicability.

While thorough, our study is nevertheless limited. First, our manipulation of neural excitability relied solely on the pharmacology of the GABA-A receptor. In our opinion, however, neural excitability is best conceptualized as a latent parameter that integrates the effects of a large number of variables, including, for example, endogenous cyclical fluctuations[5,63], brain states[64], pathological changes[35], genetic mutations[65], ion concentrations[66], the excitation-inhibition balance[67], and pharmacological modulation[68]. As such, our results provide one of potentially many crucial links with a tangible biological mechanism, while establishing a quantitative framework to assess others. Second, our ability to study seizure thresholds in humans was limited as seizures were always induced for clinical reasons (whereas single-pulse probing was done for research). In the future, more substantial clinical datasets will allow for a more systematic probing of different seizure dynamotypes[17]. Third, our optogenetic model of seizures on-demand enabled circuit and cell-type specificity but likely does not represent the sole neuronal mechanism of ictal transition.

Taken together, our translational study provides a foundation for an approach in which ever-varying neural dynamics are gauged using minute perturbations, reducing to practice the general idea that resilience and risk in complex systems can be probed[12,13,19-21]. In future neuro-engineering efforts, incorporation of real-time brain probing[28,31] will likely be necessary for dynamically targeted neurostimulation[58-62]. Overall, a dynamical systems approach to neurostimulation will help physicians, mathematicians, physicists, and engineers align on core concepts and formalize shared vocabulary for impactful advances in the field.

## Methods
### Epileptor model
The Epileptor is a previously published five-dimensional neural mass model of seizure activity[7]. Conceptually, this model is divided into three interconnected subsystems: a fast subsystem (variables $x_1$ and $y_1$) models fast ictal discharges, a slower subsystem (variables $x_2$ and $y_2$) models slower spike-wave events, and a very slow permittivity variable $z$ governs the switching between ictal and interictal states.

$$\dot{x}_1 = y_1 - f_1(x_1, x_2) - z + I_1 \tag{1}$$

$$\dot{y}_1 = y_0 - 5x_1^2 - y_1 \tag{2}$$

$$\dot{z} = \frac{1}{\tau_0}\left(4(x_1 - x_0) - z\right) \tag{3}$$

$$\dot{x}_2 = -y_2 + x_2 - x_2^3 + I_2 + 0.002g(x_1) - 0.3(z - 3.5) \tag{4}$$

$$\dot{y}_2 = \frac{1}{\tau_2}\left(-y_2 + f_2(x_2)\right) \tag{5}$$

Where:

$$g(x_1) = \int_{\tau_0}^{1} e^{-\gamma(t-\tau)} x_1(\tau) d\tau \tag{6}$$

$$f_1(x_1, x_2) = \begin{cases} x_1^3 - 3x_1^2, & \text{if } x_1 < 0 \\ (x_2 - 0.6(z-4)^2)x_1, & \text{if } x_1 \geq 0 \end{cases} \tag{7}$$

$$f_2(x_2) = \begin{cases} 0, & if\, x_2 < -0.25 \\ 6(x_2 + 0.25), & if\, x_2 \geq -0.25 \end{cases} \qquad (8)$$

with $\tau_0 = 20000$, $\tau_2 = 10$, $I_1 = 3.1 + Istim_1$, $I_2 = 0.45 + Istim_2$, and $\gamma = 0.01$. Note that compared to the original Epileptor parameters, $\tau_0$ was chosen with a larger value, to obtain longer seizure and interictal period durations. To approximate the conditions of our experiments in vivo, we chose the excitability (or epileptogenicity) parameter $x_0$ such that no seizure occurs spontaneously, i.e., $x_0 = -2.25$ for the NaCl condition, $x_0 = -2.20$ for the PTZ condition, and $x_0 = -2.30$ for the BZD condition. When the Epileptor was set in an epileptogenic state (Fig. 7), we used $x_0 = -2.0$. We modeled electrical stimulations as additional inputs $Istim_1$ and $Istim_2$ included in $I_1$ and $I_2$, respectively. The stimulation amplitude $Istim_1$ and $Istim_2$ were set to 2 and 5, respectively, unless stated otherwise. Initial conditions were chosen for each excitability condition such that the system was lying on the fixed point at the beginning of the simulation.

The time correspondence between simulations and experiments was chosen such that one-time step in the simulation corresponds to 10 ms of real-time. Temporal stimulation parameters were then chosen in the experimental setting (frequency: 20 Hz, stimulation duration: 3 ms). The system was simulated using a fourth-order Runge–Kutta method. For stochastic simulations we used additive white Gaussian noise on both the fast and slow subsystem, more specifically on variables $x_1$, $x_2$, and $y_2$, with mean 0 and variance 0.005, 0.0001, and 0.0001, respectively. The stochastic system was integrated using a modified Runge–Kutta scheme for stochastic differential equations[69].

## Human experiments and data
**Human participants.** Human data were collected from 10 patients (Supplementary Table 1) with intractable epilepsy undergoing invasive pre-surgical evaluation with stereo-EEG at Inselspital, Bern, Switzerland. Electrodes were implanted as clinically necessary for seizure localization and without relationship to the present research study. These intracranial EEG electrodes enable direct brain stimulations with pulses of electrical current to probe neural excitability in the form of cortico-cortical evoked potentials (CCEPs), hereafter termed iEEG responses for simplicity. Seizures were triggered for clinical reasons, but participants provided informed consent for receiving additional single-pulse stimulation and analysis of the iEEG data. This study was approved by the ethics committee of the Canton Bern (ID 2018-01387).

**Data acquisition.** Each iEEG electrode (DIXI medical, Microdeep®, France) consists of 8–18 platinum channels with a diameter of 0.8 mm and a length of 2 mm with varying spacing. The MRI and postsurgical CT were co-registered using the Lead-DBS software (www.lead-dbs.org) to determine the exact location of each electrode contact. A neurologist (MOB) labeled the channels based on their anatomical locations. The iEEG recording was amplified using a 128-channel Neuralynx ATLAS system (Neuralynx Inc., USA), with a sampling frequency of 2KHz, a voltage range of ± 2000 μV along with a digital trigger signal to identify stimulation onsets.

**Cortical electrical stimulations.** A neurostimulator (ISIS Stimulator, Inomed Medizintechnik GmbH, Germany) was used to deliver a single or a train of bipolar (neighboring contacts) stimulations at varying intensity and with a square-biphasic pulse of a total width of 1 ms. The same stimulation protocol was repeated before and after the intravenous administration of clonazepam 0.5–1 mg, a GABA-A receptor agonist of the benzodiazepine class (BZD), given for medical reasons (end of clinical work-up). The single-pulse protocol (SP) consisted of varying intensities ranging from 0.2 – 10 mA, each pulse repeated three times and randomly delivered with an inter-stimulation-interval (ISI) of

at least 4 sec. Seizures were provoked in the human participants using 60 Hz bipolar stimulations at 1–3 mA over a few seconds (1–6 s).

**Data pre-processing.** The human iEEG signals were preprocessed in Matlab (The MathWorks, Inc., Natick, Massachusetts, United States) in the following steps: (1) calculating bipolar derivations by subtracting monopolar recordings from two neighboring channels on the same electrode lead. (2) removing remaining stimulation artifacts by interpolation of a 12 ms window ([−2, 10]ms from trigger onset). A kriging technique is applied where a linear fit with random noise (gaussian distribution of standard deviation of 50 ms preceding data) connects the beginning and the end of the interpolation window. (3) bandpass 0.5–200 Hz and 50 Hz (and harmonics) notch filtering followed by resampling to a frequency of 500 Hz. Human and mouse iEEG features were extracted in a common processing pipeline (see below).

Lead-DBS (www.lead-dbs.org) and the Freesurfer image analysis suite (http://surfer.nmr.mgh.harvard.edu/) were used to produce 3d brain visualizations.

## Mouse experiments and data
**Mice.** All experiments on mice were conducted in accordance with protocols approved by the veterinary office of the Canton of Bern, Switzerland (license no. BE 19/18 and BE 51/2022). A total of 34 C57BL/6JRj and 4 PV_ires_Cre male mice aged between 2 and 4 months old were used. Mice were housed in ventilated cages, with food and water ad libitum under controlled conditions (12:12 h light-dark cycle, constant temperature 22 °C, and humidity 30–50%).

**Virus transfection.** Mice were anesthetized with Isoflurane (5% in ambient air for induction and 1.5–2% for maintenance, Abbvie, Switzerland). They were then placed in a digital stereotaxic frame (David Kopf Instrument, USA), and body temperature was kept at 37 °C using a rectal probe and closed-loop heating system (Harvard Apparatus, USA). Eyes were protected with ointment (Bepanthen, Bayer, Germany), and analgesia was given as subcutaneous injection of Meloxicam 2 mg/kg (Boehringer Ingelheim, Switzerland). Scalp fur was removed using a depilatory cream (Weleda, Switzerland), and the scalp was disinfected with Betadine (Mundipharma, Switzerland). After skin opening, the periosteum was roughened, and burr holes were drilled at the targeted coordinates.

In the main preparation ($N = 28$), we used an intersectional strategy to express Channelrhodopsin (Ch2R) specifically in pyramidal cells projecting from the medial entorhinal cortex (MEC) to the CA1 region of the hippocampus (PN$_{MEC->CA1}$). Two recombinant AAV were injected in two different target brain regions known to be connected, such that only neurons transfected with both viruses would express ChR2 (Supplementary Fig. 3D–F): (1) 450 nl of a retrograde virus containing the opsin on inverted cassette (AAVretro_EIFa_DIO_Ch2R(H134R)_eYFP, UNC Vector Core, USA) were injected into the right CA1 dorsal hippocampus (coordinates: antero-posterior (AP) −2.0 mm from Bregma, medio-lateral +1.3 mm from Bregma and dorso-ventral −1.6 mm from the skull level). (2) 450 nl of an anterograde virus containing the Cre recombinase under CamKII promoter to target pyramidal cells (AAV1_CamKII_Cre_SV40, Addgene, USA) were injected into the right MEC (+3.2 mm laterally from Lambda along the lambdoid suture and DV −2.5 mm from skull level). Viruses were loaded on a 500 nl Hamilton syringe (Model 7000.5, Hamilton Company, USA) and injected using a micro-infusion pump (Pump 11 Elite Nanomite, Harvard Apparatus, USA) at the rate of 50 nl/min, with a 10 min pause before syringe retraction. The skin incision was then sutured, and mice were placed back in their home-cage for recovery. They were monitored and received analgesia (Meloxicam 2 mg/kg) for 3 days.

Three other virus constructs were used for different control experiments (Supplementary Fig. 3 and 5): 1) To verify the circuit-specificity of our intersectional viral strategy and control that

Channelrhodopsin was necessary to induce seizure, two mice received the same two viruses at the same coordinates, but without the Channelrhodopsin (AAV1_CamKII_Cre_SV40 and AAVretro_EIFa_DIO_eYFP, Supplementary Fig. D1). 2) To control that seizure could also be induced by direct stimulation of CA1 pyramidal cells, four mice received an anterograde virus in CA1 (450 nl of AAV2_CamKII_Ch2R(H134R)_eYFP, Supplementary Fig. 3A, B and Supplementary Fig. D2). 3) To assess whether seizures could also be induced by stimulation of inhibitory cells, four PV_ires_Cre mice received a cre-dependent anterograde virus in the CA1 dorsal hippocampus (AAVdj_EIFa_DIO_ChETA-eYFP, Supplementary Fig. 3G, H and Supplementary Fig. 5D3). The ChETA ospin was chosen to allow stimulation at higher frequencies (>40 Hz) in fast-spiking interneurons[38](Supplementary 5C3).

**Electrodes implantation.** Three weeks after viral injection, mice were implanted with intracranial electrodes for multisite iEEG recordings. Implantation surgery was carried out in the stereotactic frame through the same incision. For longer surgeries of 3–4 h, a reversible mix (10 µl/g) was used for anesthesia, with the following composition: 10% Midazolam 5 mg/ml (Sintetica, Switzerland), 2% Medetomidine 1 mg/ml (Graeub AG, Switzerland), 10% Fentanyl 0.05 mg/ml (Sintetica, Switzerland) and 78% NaCl 0.9%. Bilateral frontal skull EEG screws (Ø1.9 mm, Paul Korth GmbH, Switzerland) were soldered to a stainless-steel cable (W3 wire, USA) and inserted at coordinates −1.0AP, ±2.0 ML. Reference and ground EEG screws were inserted above the cerebellum and the olfactory bulb, respectively. Twelve intracerebral depth electrodes, made of tungsten wires (Ø76.2 µm, model 796000, A-M System, USA), were pinned in an 18-EIB board (Neuralynx, USA) and inserted one by one and glued in place at the following coordinates: MEC on the lambdoid suture, ±3.2 ML, − 2.5DV; CA1 −2.0AP, ±1.3 ML, −1.6DV; DG −2.0AP, ±1.3 ML, −2.3DV; CA3 −2.0AP, ±2.2 ML, −2.2DV and Subiculum −3.2AP, ±1.6 ML and −1.8 DV (All DV coordinates are calculate from skull level). The right entorhinal electrode was glued to a homemade optical fiber implant (Ø200 µm, 0.39 NA Core Multimode Optical Fiber, FT200EMT inserted and glued into CFLC128 ceramic ferrules, Thorlabs, USA, Supplementary Fig. 3C). Each electrode was cemented to the skull and to neighboring electrodes to provide better stability.

Upon completion of the implantation, anesthesia was reversed with a mix for reversing anesthesia (10 µl/g), composed of 5% Atipamezole 5 mg/ml (Graeub AG, Switzerland), 2% Naloxone 4 mg/ml (OrPha Swiss GmbH, Switzerland), 50% Flumazenil 0.1 mg/ml (Anexate, Roche, Switzerland) and 43% NaCl. After the surgery, mice were monitored in their home-cage for a week and received analgesia (Meloxicam 2 mg/kg) for three days. During a brief isoflurane anesthesia, mice were connected to the iEEG recording system, and habituated for another week to move freely with the recording cable.

**iEEG data acquisition.** The implanted EIB board was connected to either a HS-16-CNR-MDR50 Neuralynx or a RHD 16-channel Intan (Intan Technologies, USA) headstage, and the optic fiber to a homemade optical patch-cord (optic fiber FT200EMT glue in a FC/PC connector, 30230G3, Thorlabs, USA). iEEG signals were amplified and digitized at 2000 Hz using either the Digital Lynx SX data acquisition system (Neuralynx, USA) or the Intan RHD USB interface board (Intan, USA). The mice iEEG signals were preprocessed in Python (Python Software Foundation, https://www.python.org/) with a 0.5–800 Hz bandpass and a 50 Hz (and harmonics) notch filter.

**Optogenetic stimulation.** For opto-stimulation, a patch-cord was connected to a 473 nm blue laser (Cobolt 06-MLD, HÜBNER Photonics GmbH, Germany) controlled by a Matlab (Mathworks, USA) script through a pulse train generator (PulsePal 2, Sanworks, USA). The digital trigger signal was recorded along with the electrophysiology

data. The analog modulation mode of the lasers was used to stimulate different light intensities by employing varying input voltages. Maximum intensity was calculated to be around 30 mW at the tip of the optic fiber. The reliability of the laser outputs and modulation was ensured previously by recording laser power for each of the stimulation protocols with a photodiode (PM100A, Thorlabs) connected to the Digital Lynx with a Universal Signal Mouse board (Neuralynx Inc., USA).

**Behavioral assessment.** Video of the seizures were recorded using zenithal webcams (HD Pro C920, Logitech, Switzerland) and the OBS Studio software (https://obsproject.com/). Video recordings of the induced seizures were scored offline and blinded to pharmacological condition, using a modified Racine scale as follow: 0: no visible change, 1: behavioral arrest, 2: clonus without rearing, 3: clonus with rearing, 4: clonus and falling on side, 5: wild jumping, 6: death. Three mice (18 sessions) were not filmed and couldn't be included in the analysis.

**GABA-A receptor agonists and antagonists.** A benzodiazepine (BZD, Diazepam 10 mg/2 ml, Roche, Switzerland) and Pentylenetetrazole (PTZ, P6500, Sigma-Aldrich, USA) were diluted in NaCl 0.9% such as to inject a constant volume intraperitoneally across concentrations (2 µl/g i.p.). Diazepam dose was set at 5 mg/kg if not specified otherwise. subconvulsive and convulsive doses of PTZ were initially given at 20 mg/kg and 30–40 mg/kg, respectively, but occasionally had to be adjusted on a per-animal basis. For example, if a subconvulsive dose nevertheless led to a seizure, data from this session were discarded and the next dose for this mouse was reduced to 75% of the previous dose.

**Optogenetically-induced seizures.** In the main experiments (Figs. 2–4, and 6), each mouse (N = 17) underwent one to six experimental blocks. An experimental block included three 1.5-h sessions on different afternoons (second half of the light phase), each with one of three different pharmacological conditions: BZD, subconvulsive PTZ or control NaCl i.p. Sessions within an experimental block were organized in random order at intervals of 48–72 h to allow for drug elimination and minimize kindling (i.e., the tendency for seizures to become more severe over time). Each session started with a 10-min baseline recording, followed by three different optogenetic stimulation protocols preceded by an i.p. injection of the same drug (BZD, PTZ, or NaCl) 3 min before (repeated at 50% for PTZ, fast elimination[34,47] and 10% for BZD, slower elimination[43,44]). To avoid changes in neural excitability related to vigilance stages, mice were kept awake by gentle handling. The optogenetic protocols were, in a fixed order: (1) 270 paired pulses (PP) at varying inter-pulse intervals (6–2000 ms) every 8–12 s over 45 min, with three different intensities (3 ms at 1/3 max intensity, 2/3 max intensity or max intensity) for the first pulse and a fixed intensity (3 ms at 2/3 max) for the second pulse. (2) 60 single pulses (SP, 3 ms) at 12 different intensities, linearly distributed in the range of the laser analog modulation (0.45 V to 1 V), over 45 min. Ten additional low intensities (analog modulation 0.45V-0.55 V) were used to determine the minimal intensity necessary to obtain a detectable response (rheobase) in the iEEG. (3) Train stimulations (20 Hz) of increasing duration (0.25 to 30 sec, presented at one-minute intervals) for seizure induction until a seizure occurred (Fig. 1C1). Seizures were visually detected by a trained experimenter as sustained (>10 s) ictal activity continuing after the end of the stimulation. In a supplementary experiment involving 8 mice, seizures were also provoked at different stimulation frequencies (4, 7, 10, 20, 40 Hz, Supplementary Fig. 2). Each frequency was tested within one of five experimental blocks that each included three sessions with different pharmacological conditions, as above. In another supplementary experiment, a subset of mice (N = 8, Fig. 6) underwent one additional experimental block that included five sessions, each randomized to

one of four different doses of Diazepam (1, 3, 5, and 7 mg/kg) or the control condition (NaCl).

**PTZ-induced seizures.** For this experiment (Fig. 7), 8 mice underwent 1–3 recording sessions including 10 min baseline recording, followed by a first protocol of stimulation with only single-pulse (20 pulses at maximum intensities, every 8–12 s) to probe excitability at baseline. After convulsive PTZ injection single pulses were repeated every 8–12 s until the recording was stopped after the occurrence of a seizure (as above). Two control experiments were included: (1) The same stimulation protocol was carried out in the same mice before and after injection of NaCl i.p.; (2) To verify that single-pulse stimulations were not necessary to induce seizures, another five control mice received supra-threshold PTZ in absence of optogenetic stimulation and also developed seizures (5 out of 5).

**Mice histology.** Mice were euthanized after the end of the recording blocks. They were anesthetized with 250 mg/kg Pentobarbital (Esconarkon 1:20, Streuli Pharma AG, Switzerland) and transcardially perfused first with cold NaCl 0.9% and then with 4% formaldehyde for 5 min each. Extracted brains were post-fixed in 4% formaldehyde (Grogg Chemie, Switzerland) for 24 h, then transferred in sucrose for 48 h before being flash-frozen with −80° methylbutane. Brains were then sliced along the either sagittal, coronal or axial axes (40 μm slices) on a cryostat (Hyrax C 25, Zeiss, Germany), and collected in PBS. For immunostaining against the GFP, slices were first incubated for 1 h in a blocking solution composed of PBST and 4% bovine serum albumin. Free-floating slices were then incubated for 48 h at 4° with anti-GFP primary antibody (1:5000, Ref. A10262, Invitrogen, USA). They were then rinsed 3 × 10 min with PBS containing 0.1% Triton and incubated 1 h at room temperature with a secondary antibody AlexaFluor 488 (1:500 Abacam, ab96947). Finally, slices were washed again for 3 × 10 min and mounted on glass slides. Images were obtained using an epifluorescence microscope (Olympus BX 51, Olympus Corporation, JP) at different magnifications (4–10×).

### Common data processing pipeline

After pre-processing, all the remaining signal analysis was carried out using custom Python scripts and following identical steps for mouse and human data.

**Single channel-level iEEG response analysis.** The evoked response to a stimulation pulse was measured as the line length (LL)[15,49] per millisecond (ms) of the iEEG signal as follows:

$$LL = \frac{\sum_{i=1}^{N} |(x_i - x_{i-1})|}{N} * sf/1000 \quad (9)$$

Where $N$ is the number of datapoints over which the $LL$ is calculated, $sf$ is the sampling frequency and $x$ the iEEG voltage measured at each datapoint. For single and paired-pulse responses, the LL was calculated over the first 250 ms (Supplementary Fig. 3H, I). This window includes both negative peaks of a typical CCEP described in human[66]. For LL calculated during 20 Hz train stimulation (Supplementary Fig. 2J) the 50 ms inter-pulse window was taken. For each session and each intensity, the pulses with the higher LL were visually checked to ensure that there was no artifact and removed otherwise. This process was done blind to the session condition.

**Network-level iEEG response analysis.** To measure differences in single-pulse iEEG responses across multiple electrodes and stimulation intensities and summarize them in a single value, we used non-negative matrix factorization (NMF)[50,51]. For each individual (mice or human participants), responses to each single stimulation across sessions and conditions were measured in each electrode using the LL as described

above and stacked into an input matrix V of dimension N_electrodes x N_stimulations. The NMF algorithm (sklearn implementation) then decomposes (*factorizes*) in a non-supervised manner the input matrix V into two smaller matrices W (size N_electrodes x Rank) and H (N_stimulations x Rank) (see Supplementary Fig. 8):

$$V \sim R = W \times H \quad (10)$$

where the algorithm seeks to minimize the difference between the reconstructed matrix $R$ and the original matrix $V$ using the multiplicative updates algorithm[50]. The resulting $W$ matrix represents the weights assigned by the algorithm to each recording electrode and H the activation coefficient of these weights for each stimulation. Thus, one stimulation site may give rise to a sub-network of channels that tend to respond together and are grouped in a common basis function (Wᵢ). The rank corresponds to the number of sub-networks in which $V$ can be decomposed. In the mice optogenetic experiment, only one stimulation site was used (MEC) and all iEEG electrodes, implanted in limbic areas, showed robust evoked responses. Therefore, a rank of 1 was always selected (see Supplementary Fig. 8). In humans, for each participant two different stimulation sites were used and only part of the recording electrodes showed evoked responses. Consequently, we performed a stability NMF analysis[70] resulting in an optimal rank between 2 and 8. For each stimulation site, sub-networks that show increased response to increasing stimulation intensity were then selected as the responsive sub-networks and kept for analysis, effectively discarding background noise from non-responsive electrodes (Fig. 5C). Finally, for each sub-network and each pharmacological condition, we compute an input-output response curve by computing the average H coefficient in response to each stimulation intensity. The area under this curve (IOC) was used to measure the overall network responses across intensities and electrodes.

**Passive signatures of critical transition.** In addition to active probing, several passive dynamical signatures have also been used in the past to evaluate critical transitions. To measure them, we used iEEG signal at distance (4 s) from pulse stimulations. Five classical dynamical signatures of critical slowing in the time- and frequency-domain were computed: variance, skewness, line length, autocorrelation, and spatial correlation.

They were respectively calculated for each channel on 4 s iEEG epochs, bandpass between 0.5–100 Hz, respectively with the Numpy, Scipy, Statsmodels and Numpy functions as follows:

$$Variance = \frac{\sum_{i=1}^{N}(x_i - \bar{x})(x_i - \bar{x})}{N} \quad (11)$$

$$Skewness = \frac{\sqrt{N(N-1)}}{N-2} \frac{\sum_{i=1}^{N}(x_i - \bar{x})^3/N}{SD^3} \quad (12)$$

$$Autocorr = \frac{\sum_{i=k+1}^{N}(x_i - \bar{x})(x_{i-k} - \bar{x})}{\sum_{i=1}^{N}(x_i - \bar{x})(x_i - \bar{x})} \quad (13)$$

$$Spatial\ corr. = \frac{\sum_{i=1}^{N}(x_i - \bar{x})(y_i - \bar{y})}{\sqrt{\sum_{i=1}^{N}(x_i - \bar{x})^2} * \sqrt{\sum_{i=1}^{N}(y_i - \bar{y})^2}} \quad (14)$$

Where $N$ is the number of datapoints, $x$ the measured iEEG voltage at each datapoint, $\bar{x}$ the mean iEEG voltage, and SD the standard deviation, i.e., the square root of the variance. For the autocorrelation, the measure taken was the width of the autocorrelation function at half-maximum value. For the spatial correlation, the Pearson correlation coefficients were calculated between each electrode pair ($x$ and $y$) and

then averaged to obtain a mean spatial correlation. The line-length calculation defined above to measure iEEG responses to probing pulses was applied on passive recordings with an integration window of 4 s.

**Classifiers.** To investigate the consistency of our effects at the single-trial level, we trained logistic regression classifiers to decode the excitability level (i.e., the pharmacological condition) based on the iEEG signal. We built three different classifiers: (1) an *active probing* classifier that takes as input the iEEG signals from all the channels over a 250 ms window following a single-pulse stimulation; (2) a *passive signatures* classifier that takes as input the five passive signatures of critical transition described above, calculated on a 4 s iEEG epochs between two pulses for each channel; (3) a combined classifier that takes as inputs all the multi-channel passive signatures and response to the single-pulse. The classifiers were built using the sklearn multiclass implementation of the logistic regression with L2 regularization to avoid overfitting. The classifier was then trained to attribute to a given vector $X_i$ a probability $p_k$ to belong to a given class k as follows:

$$p_k(X_i) = \frac{e^{X_i W_k + W_{o,k}}}{\sum_{l=0}^{K-1} e^{X_i W_l + W_{o,l}}} \qquad (15)$$

Where $K$ is the total number of classes and W the coefficient matrix. As a classification problem, the objective for the optimization is then:

$$\min W\, C_i = 1^n \sum_{k=0}^{K-1} [y_i = k] \log(p_k(X_i)) + l2 \qquad (16)$$

With $y_i$ being the label of the observation $X_i$, $n$ the total number of observations and l2 the penalty term. For each mouse, one classifier was trained to discriminate between three classes (NaCl, subconvulsive PTZ and BZD) using a fivefolds cross-validation strategy. Performance was assessed with the accuracy as follows:

$$\text{Accuracy} = \frac{TP + TN}{TP + TN + FP + FN} \qquad (17)$$

Where TP, TN, FP and FN are respectively the numbers of true positives, true negatives, false positives and false negatives. To determine the chance-level, classification scores were compared with the ones obtained from surrogate data in which the class labels were permuted 100×. $P$ values were then obtained using this formula:

$$p - value = \frac{C + 1}{n_{perm} + 1} \qquad (18)$$

Where $C$ is the number of permutations whose score is higher than the true score, and $n_{perm}$ the total number of permutations.

**Statistics.** If not specified otherwise, statistical testing was performed using bootstrapped estimation of the group average and graphical representations[68]. Differences between conditions were calculated and reported as the mean difference (i.e., the effect size) and its 95% confidence interval (95% CI), obtained by performing bootstrap resampling 5000 times. For comparison across conditions, data were always either paired or normalized by block to control condition (NaCl in mice, baseline in participants).

### Reporting summary

Further information on research design is available in the Nature Portfolio Reporting Summary linked to this article.

## Data availability

Sample and process data necessary for running the code and reproducing the analysis and results presented in the paper are available at: https://doi.org/10.6084/m9.figshare.25305238. Please note that this repository is not meant to be a comprehensive data repository. If you require access to the full, unprocessed dataset used in the research, please contact the authors directly. Source data are provided with this paper.

## Code availability

The code used to generate the analysis and figures presented in the paper is available at: https://doi.org/10.6084/m9.figshare.25305238.

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

## Acknowledgements
This work was funded by a Swiss National Science Foundation grant (Ambizione 179929) and by a Velux Stiftung grant (1232) to Dr. Baud. We thank Dr. Richard J Burman, Dr. Forrest Webler, and Dr. Vikram Rao for their proofreading of the article and suggestions. We thank Maria Essers and Regina Reissmann for their help with the histological slices. We thank Dr. Mignardot for the visual rendering of brain reconstructions. Some of the visuals were created using BioRender.com.

## Author contributions
G.L., M.O.B., K.A.S., and A.A. designed the study. G.L. and K.S. developed the optogenetic model. G.L. collected the mouse data. M.F., W.Z., and C.P. supervised the implantation and the clinical care of the patients. E.V.M., M.O.B., and C.F.M. collected the human data. T.P. ran the in silico simulations. G.L. analyzed the data with contributions from K.S., E.V.M., and T.P. G.L., T.P., and M.O.B. wrote the first manuscript. All authors contributed to the final version of the manuscript.

## Competing interests
M.O.B. holds shares with Epios, Ltd., a medical device company based in Geneva. All other authors report no conflict of interest.
