## [Peer Review File · Nature Communications]

The Critical dynamics of hippocampal seizuresREVIEWER COMMENTS

Reviewer #1 (Remarks to the Author):

Review of Lepeu et al

This paper sets out to provide experimental evidence, from both mice and human recordings, regarding whether transitions into an epileptic seizure might be characterized by stimulating the relevant brain networks. The importance of this issue, clinically, is about developing a means of predicting whether a seizure is imminent. They follow up recent work by other groups, in which this question is framed in terms of the “resilience” of normal brain function to making this transition. Brain states that are far from the transition point are “resilient” to perturbations, whereas those that are very close may be tipped over the transition by small perturbations, or even noise in the system. The key point is that in this latter state, the sensitivity to noise means that the transitions are inherently unpredictable (that is, noise is an important determinant). Mapping this distance from the transition point – what the authors term “resilience” – could therefore be of great value for predicting seizures clinically.

The topic is, therefore an important one, in my opinion, and the novelty of the paper is the attempt to demonstrate the utility of “active probing” for mapping of normal brain states. More importantly, active probing may yield information that is not readily available from analyses of passive recordings of brain function, and so could be a very useful tool clinically. A major problem with the paper, however, is that it was not structured appropriately, and consequently is rather difficult to read. It was extremely dense, with many complex figures, but the descriptions within the text of the results were often rather cursory. There were also long blocks of expository text - e.g. about the Epileptor model. This model is described in detail in other publications, and while the descriptions here were quite readable, and may be very helpful to unfamiliar readers, they would be better relegated to the supplementary information (the same could be said for Figure 1), or alternatively, as a stand alone review, and not in the results section. The inclusion of extended descriptions of this model has an unfortunate consequence, because far less space is given to the novel elements of the study. For instance, the first result occurs after a whole page of exposition in the results section, at the end of the section “Provoking seizures in mice”, but is given in very vague terms as a high level synopsis, in a single sentence, but without any real detail, referencing only “Extended figure 3”. The fact that this figure is in the supplementary info is indicative of my point; it illustrates a key methodological step of the analysis of the real experimental data, and as such is more important to include than the “review-like” figure 1, which is in any case replicated schematically in panel A of figures 2-4.

This pattern (giving the novel result as a synopsis, after rather long descriptions of the motivation behind the study and the model) is repeated throughout the results. If the results section focused more upon the experimental data, far more detail could be provided, and I think the paper would be improved greatly. Without this detail, however, I personally found it very difficult to get to the nub of what the novel results show. For instance, in Fig 6H, it appears that active probing improves upon passive probing alone, but when both are used, if anything it seems slightly worse than active probing alone (although the stats for that comparison was not provided). I'd have thought that using more information should improve matters. This detail was not mentioned at all in the results text.

For many of the listings of specific results, it wasn't clear what was the sample size. Was this always "21 mice" as indicated early in the results? Also, the authors state that only 1 human subject received BZD treatment, but for all the human data presentation, the implication was that there were multiple biological replicates. Please just state the sample size for each statistic presented.

I was interested that the description of prior work behind this study focused mainly upon high level theoretic models of critical transitions, which are agnostic to the detail of how these arise in cortical networks. In contrast, two other studies which used very similar techniques to those used here (optogenetic stimulation prior to the onset of epileptic activity) and which are thus highly relevant to this manuscript, namely Klorig et al, and Graham et al, were referenced, but virtually in passing, and without any proper discussion. Both studies used in vivo models, although the referencing implies that the latter was only of in vitro studies (cf. lines 304-5). The latter of these, in particular, presents a biological model of the altered optogenetic response in terms of a change in dendritic excitability (see also PMID: 36923333), and thus grounds theoretical models such as the Epileptor model in the actual biology of cortical networks. I am interested to know whether the authors consider their findings to support this biological model, or if not, how their data diverges from that study.

Another piece of biological data in the manuscript that deserves more attention is the failure to trigger seizures by optogenetic stimulation of PV interneurons. There have been a raft of publications showing that optogenetic stimulation of (subsets of) interneurons can trigger seizure-like events, although notably these have almost all been done in the presence of 4-aminopyridine (e.g. PMIDs 26134638, 25505119, 29024712, 25546300). I was interested therefore that this failed to occur in normal mice. This should be discussed with reference to these prior papers.

The readability of the manuscript could be improved by being substantially shortened. At times, the writing is too grandiose for my taste: the first line of the abstract, for instance "volcanoes erupt, markets crash". This paper has relevance to neither particularly, since no one is talking about stimulating mountains to test their resilience. I also do not like the repeated use of parenthetical statements which just repeat what is in the text, but using different wording (e.g. line 212

“Intuitively, as the recovery rate increases (a speed)...”). In other places parentheses were used to give the exact opposite meaning (lines 184f “decrease (respectively increase)...”). In both cases, I found this style quite distracting, and unhelpful. The topic of the paper is difficult enough as it is, and so the reader would be helped by making the writing as simple as possible.

The equations in Fig1 are provided in the methods (and are more readable there), so should not be in the figure.

There are lots of syntactical and grammatical errors throughout the manuscript – the most common are many errors regarding mismatched nouns and verbs (singular and plural mismatches, e.g. line 199, “In one included human subjects...” and multiple other examples; also “included” is not really necessary), missing pronouns, and the occasional really poor sentence (“We transfected these each group of neurons using simple of intersectional transfections...”). Also figure 2 legend – “makers” should be “markers”. A proper proof-reading would help.

Reviewer #2 (Remarks to the Author):

This study aims to measure the resilience (i.e. seizure threshold?) using a variety of passive and active methods in a computer model, rodent model, and human epilepsy. The idea of measuring seizure threshold has very important implications, and this work has some interesting controlled experiments that show they can measure the threshold. However, the presentation really clouds this fact. The article tries to claim too much: the title is too broad, the figures spend too much time on examples or obvious findings, much effort is spent trying to prove a model prediction about resonance and inhibition that really is not important to the overall message, and even the phrase “limbic seizures” is an overreach. While there is some very important information here, and I really appreciate the ability to measure seizure threshold in a controlled fashion in animals and humans, the salient results appear to be limited, or at least they are not presented in a rigorous, summarized fashion. Most of my scientific concerns are focused on issues when the authors try to extrapolate from the model to mechanisms (e.g. lack of inhibitory drive because no resonance)—I do not think those discussions are necessary for this work to be useful and interesting; conversely I think it would require many more experiments to prove those things. I recommend focusing on the methods (which look very useful) and how they seem successful under controlled conditions rather than making statements about mechanisms (which cannot be proven well in these studies).

Many examples here in test resilience over time. But repeated tests can change resilience. Did they randomize the order, or somehow account for this?

Throughout the paper there is a lot of talk about feedback loops. But Epileptor does not inherently have a feedback loop. And did they actually test the loop in the experiments? It is not apparent to me if they did

L 65 The following sentence is written in very confusing fashion “Therein, finetuned positive neuronal feedbacks ensure sensitivity to weak yet relevant inputs, at the cost of low resilience to perturbations”. Specifically, “finetuned”, “ensure sensitivity”, “relevant”, “cost” and “low resilience” are all very loaded words that could be interpreted in a variety of ways.

L 97 The following sentence implies that seizure prediction algorithms have failed, which is not true (there are many successful algorithms...the problem is a lack of prospective data). “has defeated many machine-learning algorithms”

Why does this article focus on the “limbic system” rather than ‘temporal lobe epilepsy’? The latter is by far the more common terminology.

L 120 why a “fold bifurcation”. Isn’t the Epileptor a saddle node bifurcation? Fold is jargon here.

“resilience” needs to be defined in the introduction. It is used throughout the paper.

L 127. The issue with provoked seizures is that you cannot be certain it is the same seizure as a spontaneous one. This line reveals the problem—it can be ‘another path over the threshold’. Since “any brain can seize”, this becomes a serious issue. How can you be sure the provoked seizures are the same?

L 132 The model of provoked seizures is highly nonphysiological. What is the goal of this section? It needs to be clear that this is a limited model and describe why it is being used

L145+. Combining time to seizure, resilience, path length, and line length is unproven and seems unjustified. (see comments for Ext Fig 3)

From the video this seizure does not look like an Epileptor seizure. It starts not with a saddle node bifurcation (like Epileptor), but instead looks like a SNIC. The offset likewise does not have the slowing down of a homoclinic bifurcation, but has arbitrary firing like a Fold Limit Cycle.

L 154 It is unclear what conclusion we can make about human seizures starting with 3 seconds of stimulation. Even assuming the system is integrating, it is total area under the curve, i.e. pulse width x pulses x amplitude. Amplitude in the human stim cannot be compared with the mice. The electrodes, amps, volumes of tissue are all completely different. This is not a valid comparison with the mice. Furthermore, the text implies that there is a fixed seizure threshold, but in fact half (2/4) of the subjects tested twice had DIFFERENT thresholds. This whole section is very weak.

Fig. 1. The numbers in E describe some very complex details that really have no explanation in the text: resilience, recall flow, stability landscape, etc. This is probably too much detail unless there is going to be an explanation in the appendix.

What do you mean “no need for kindling”? That these mice would seize with the first-ever exposure to pulse trains? That is what you say

Ext Fig. 1, E&F Why is ER bright only in E and CA1 bright only in F?

Ext Fig 2: Where does the statement come from that integrators don't have inhibition induced transitions but resonators do? Applying that statement to real seizures/physiology seems a major and unsubstantiated reach, and it is frankly unclear why it even matters. Are the authors trying to use a dynamics paper to prove that interneurons cannot produce seizures? And this issue becomes a major roadblock in the rest of this paper, because the authors try so hard to prove it, but ultimately do not.

What is an “inhibitory stimulation” in Epileptor? a negative pulse? What is it in vivo? Do those two situations really agree?

Showing similar frequency characteristics between Epileptor and physiology is not valuable, since Epileptor has arbitrary temporal values.

I do not understand what the point is about the statement with 100 Hz stim into ChETA (lines 950-952). What was the result?

Thus there is a lot of data in this figure but much of it seems arbitrary and it does not really prove what it claims to prove (that Epileptor successfully models these seizures?). In addition, the title of this figure is NOT proven by the data (“lack of resonance in the limbic system”). That title should be changed. It seems like the authors are making several unsubstantiated leaps (the model predicts only bifurcations with resonance can start with inhibitory stimuli, then they want to say the physiology has no resonance, meaning it cannot start with inhibition) to make strong claims about

mechanisms. This is unsubstantiated by their data and requires several leaps of faith between the model and physiology. But I don't see why it's important to show this at all. Nevertheless, what this figure does show is that it appears to be the integration of positive input that leads to a seizure, at least when that input arrives fast enough not to dissipate. This really seems to be the crux, but it is not explained anywhere. Proving that resonance is not present will require much more meticulous testing, even in the Epileptor (all you have done is show integration is present, not that resonance is absent). Proving this would require a vast range of frequencies to disprove resonance, then highly meticulous control of interneurons over a vast range of potential conditions to disprove inhibitory drive (and note that several people have PROVEN inhibitory drive! (e.g. Marco de la Curtis, Ben Ari)). And proving anything occurs "in the limbic system" will require many more in vivo experiments (simultaneous recordings in ANT, cingulate, amygdala, hippocampus, parahippocampal gyrus, etc). The paper really does not do this, it's all just temporal lobe and should be presented as such.

Ext fig 3: C - F do not make sense to me. Are they swapped? Even then I am confused.

In G-H, the idea that "cumulative LL" corresponds to seizure threshold is brought up. This would be very helpful, but I do not see any systematic proof that this works, only a hand-waving figure here. Of note, I am quite sure it will not work for all time scales, i.e. a very slow, long pulse train might reach the same LL as a short, fast one, but only the fast one will cause a seizure. Simply "calling" this 'distance to seizure' is not enough—it needs to be proven better.

L 167 "From a dynamical standpoint, the limbic system is thus bistable in healthy mice and human subjects." The study did not test healthy human subjects (which is implied by the phrasing). The study did not test "the limbic system" but instead just a small region of hippocampus.

L 173-180: The small experiment with ChETA tested whether fast PV spiking can cause seizures. It did not. But this did not prove "lack of resonance" nor that inhibition cannot cause seizures. It tested only a single frequency, and as stated before the statement about inhibition has questionable application to real physiology. Overall, I do not see the purpose of this line of experimentation: why do you need to disprove resonance and inhibition (which is VERY hard)? The authors do not even have a model that has resonance that would allow them to test it in silico...why are they looking in vivo? Why not just focus on the positives and the model they already have? This section is weak.

L 183-202: These experiments show that the length of stimulation equates to seizure threshold, and that PTZ lowers/BZD raises the seizure threshold. These experiments are straightforward and lend credence to the stimulation duration as a surrogate of seizure threshold. However, the results are exactly as expected. What I do not understand is whether this really "probes neural resilience". Although the neurophysiological literature has been quite lax about what "resilience" means, the authors need to be specific here. These experiments simply seem to be measuring seizure

threshold—is that what they mean by “resilience”? More importantly, Fig. 2 has 9 panels, but 8 of them are just examples. The only “data” are in C4. Is that the entire summary of the experiment? I cannot read that tiny figure, which has a mean difference to NaCl and bootstrapping. What is the difference from “normal”? This figure should be focused on the summary results, not simply rehashing the concepts from prior figures.

L204-222: This section and Fig 3 assess how the systems respond to single pulse stimuli, essentially measuring the evoked potential with line length. The results are that PTZ makes the LL longer, and Benzos make it shorter. These are straightforward and unsurprising results. I do not understand the use of AUC in 3C3 and 3D3. AUC is usually from an ROC curve—which curve? I also do not understand the scatterplot in 3C3.

L 224-273 show that the PTZ/BZD changes the background activity and connectivity, which are not surprising.

L 274-292: This section now applies all the prior methods, which were not very surprising or informative, to make a classifier of dynamics. This is the first section that has intriguing data. As in most prior figures (which is a weakness), Fig 6 is mostly examples and the actual results are relegated to a corner and hard to read. Though it is hard to decipher, I believe what they have shown here is that these quantitative methods are able to discern between animals with BZD, PTZ, or nothing. That is really important and needs to be much clearer. And the results are extremely hard to interpret. Just how good is the classifier? What do these performance measures mean? Were there held out data or is this overtrained? Why is the multivariate “combined” worse than just active probing (6H)?

L 294-314:

These are very interesting experiments, made to assess whether they can measure seizure threshold in a controlled fashion. These results are quite intriguing. However, once again the summary result (the most important) is relegated to small parts of the figure. I recommend making it much more clear how these measures are functioning as a biomarker of seizure risk. One important thing to discuss is how to account for if one of the SPES triggered the seizure (which is certainly likely in these PTZ animals). Would that SPES be removed from analysis? How would you know if it was the trigger?

Discussion:

The statement “fundamental mathematical predictions that are domain agnostic” is vague and should be rewritten.

L. 330: This study did not prove lack of resonance. It is also unclear why you need to prove it was an integrator.

L 338. This study did not confirm that increased excitability decreases [i.e. 'causes' a decrease in] resilience [which is never defined]. It confirmed that some types of increased excitability are correlated with seizure risk in these limited models. This statement needs to be heavily qualified

L. 345. The study did not show the measures had precision. Perhaps they did—but that was not demonstrated by the figures. Is there a dose response relationship between the measures and the drug? That would be tremendously important.

Reviewer #3 (Remarks to the Author):

1. Start of results. Is epileptor a neural mass model or a phenomenological model of the EEG?
2. Figure 1. It would be helpful to know what the control parameter/neural excitability represents in the context of the epileptor variables.
3. There seems to be a grammar error in this sentence: We transfected these each group of neurons using simple of intersectional transfections with adeno-associated viruses (AAV) carrying Channelrhodopsin 2
4. What evidence do you have that epileptor is a good model of limbic seizures?
5. Figure 2, what is the link between neural excitability in the epileptor model and the duration of optogenetic/electrical stimulation? Is neural excitability a variable in epileptor? and is that variable somehow related to duration of optogenetic/electrical stimulation? i.e. without considering every single detail what is the intuitive biophysical relationship between the two? Neural excitability is a property of the model. Optogenetic stimulation is not part of the model. Is the control parameter x_0 a measure of neural excitability? Why? The manuscript should make these points clearer.
6. Figure 2, "Quantification of changes in resilience (time-to-seizure) across mice (a) and sessions (s)". Not sure what "(a)" and "(s)" is referring to in the figure.
7. Extended figure 2. In the epileptor model, what does it mean to provide excitatory stimulation vs inhibitory stimulation? In B4 why stimulate epileptor for 0.75s while in C4 you stimulate for 3.5s? In the caption you say validate the epileptor prediction but hasn't epileptor been designed to produce/mimic this data? Normally a model would be fit to some data that it could explain and

then it gets applied to something new where if it explains something new, that can be considered a prediction.

8. Extended figure 3. The column titles '...in silico' and '...in vivo' seem to clash with the caption descriptions of A to K. For instance the caption description in C refers to in vivo but it is in the 'in silico' column and G seems to refer to the model but then is in the 'in vivo' column. The caption needs to better clarify what is in silico and what is not.

9. typo...within on average a median [range] of 3.0 seconds

10. You say "From a dynamical standpoint, the limbic system is thus bistable in healthy mice and human subjects." This seems overly confident. What do you mean by this? Up to this point you mainly showed how time to seizure properties are similar in silico and in vivo. Why does this imply bistability? Why not multistability or something else? Also, doesn't electrical/optogenetic stimulation drive fast neural activity rather than a change say in slowly varying physiological parameters like synaptic strength, so are we really moving from one fixed point to another via a parameter change?

11. Figure 3 B2 what does CE stand for? Cortical excitability or something else? The model and the data have similar properties, what can be done to better fit the model to the data? i.e. B3 vs C3. In Figure 2 the physiological 'control parameter' is time to seizure but then in figure 3 the physiological 'control parameter' is the drug being used, so what is the actual physiological control parameter and how does it relate to the model's control parameter?

12. Figure 6, F, what is actually represented on the x-axis? What is 'network excitability'?

13. For decoding network dynamics, does time of day confound the ability to classify BZD, NaCl and PTZ states? i.e. was BZD, NaCl and PTZ each always delivered at the same time of day?

14. "we found that iEEG responses slightly increased up to 10 minutes before the ictal transition" increased in frequency, amplitude, duration or what?

15. Figure 7, thinking of PTZ concentration as the physiological control parameter, is there any way to model the increase in concentration in the tissue as it is being absorbed? and plotting the profile of this modelled concentration in the lead up to the first seizure, for instance in E and F.

Reviewer #4 (Remarks to the Author):

"The Critical dynamics of limbic seizures" by Lepeu et al. detail a series of experiments in computer simulations, mice and humans that investigate if epileptic seizures can be predicted in a using passive observations or active probing. I enjoyed reading the paper and in my view, the noteworthy results are some level of consistency across computer simulations, mice and humans, particularly in the active probing paradigm. These sets of results contribute towards a growing literature on

markers of increased excitability, proximity to bifurcation points, or impending transitions to seizures. The results presented largely agree with existing work [Meisel 2015, Chang 2018, Maturana 2020, Graham 2023 just to name a few], therefore providing further support to multiple strands of ongoing investigations.

The results also offer some potentially novel insights, particularly the comparison of passive signal markers vs active probing. It appears that active probing is superior or “more sensitive than passive signature” in humans. However, these results are also somewhat in disagreement with Meisel 2015 (PNAS) showing that passive markers of network synchronisation are at least correlated with active probing results with an $R^2 > 0.5$. The authors explain that “The brain operates over multiple spatial and temporal scales and statistical properties may differ at different scales not studied here”, which may serve as an explanation. However, as an additional analysis, I suggest the authors directly compare their passive and active signatures to enable comparison between their models, as well as to previous literature.

The authors further highlight another potential novelty in the nature of the bifurcation (fold) in limbic seizures. Please can the authors clarify if these results are limited to the computer model and mouse model (and only partially verified in one human patient)? If so, the conclusion/discussion paragraph should highlight that we cannot draw this conclusion on humans yet. Indeed, multiple papers in the past have hypothesised that a variety of bifurcation types might underpin limbic seizures in humans [Saggio 2020].

Overall, the level of evidence for human epilepsy is comparatively low and bordering on anecdotal. Nevertheless, I think the authors should be applauded for their effort in attempting to establish this link. Some more nuanced discussion on the translational perspective would better highlight this effort and do it justice in terms of extrapolating from the animal and computer model.

To assess the statistical validity of this work, more details are needed in the description, and in the code, which should be made available for review. For example, in Fig 5 D, I would either expect to see 16, or 7 data points in each condition, but the figure appears to show 12? I would strongly suggest more transparency in this regard in the next revision. It also appears that multilevel statistics might be appropriate in some instances, e.g. to account for multiple observations from individuals or conditions.

Review of Lepeu et al

Reviewer #1 (Remarks to the Author):

This paper sets out to provide experimental evidence, from both mice and human recordings, regarding whether transitions into an epileptic seizure might be characterized by stimulating the relevant brain networks. The importance of this issue, clinically, is about developing a means of predicting whether a seizure is imminent. They follow up recent work by other groups, in which this question is framed in terms of the “resilience” of normal brain function to making this transition. Brain states that are far from the transition point are “resilient” to perturbations, whereas those that are very close may be tipped over the transition by small perturbations, or even noise in the system. The key point is that in this latter state, the sensitivity to noise means that the transitions are inherently unpredictable (that is, noise is an important determinant). Mapping this distance from the transition point – what the authors term “resilience” – could therefore be of great value for predicting seizures clinically.

The topic is, therefore an important one, in my opinion, and the novelty of the paper is the attempt to demonstrate the utility of “active probing” for mapping of normal brain states. More importantly, active probing may yield information that is not readily available from analyses of passive recordings of brain function, and so could be a very useful tool clinically. A major problem with the paper, however, is that it was not structured appropriately, and consequently is rather difficult to read. It was extremely dense, with many complex figures, but the descriptions within the text of the results were often rather cursory. There were also long blocks of expository text - e.g. about the Epileptor model. This model is described in detail in other publications, and while the descriptions here were quite readable, and may be very helpful to unfamiliar readers, they would be better relegated to the supplementary information (the same could be said for Figure 1), or alternatively, as a stand alone review, and not in the results section. The inclusion of Supplementary descriptions of this model has an unfortunate consequence, because far less space is given to the novel elements of the study.

R: We thank the Reviewer for pointing out the density of the text and the imbalance between explanatory paragraphs, and novel results. We have now more prominently emphasized the novelty of the results in the result section. We also shortened the paragraph on the description of the model, and emphasized how we generated hypotheses for the rest of the result section. In our opinion, it is important to keep this central piece of methodology/theory in the result section, and not relegate the entire paragraph to the methods, as the journal’s style is to go straight from the introduction to the results. Many readers may not read the methods, and will still benefit from a brief explanation on the model underlying all analyses. Moreover, we explore predictions from Epileptor that were not thoroughly tested in prior publications, and these reasoning clarify what hypothesis is tested with each experiment. We now include one summary sentence at the end of each paragraph of the result section to emphasize the novelty or confirmatory value of each result.

For instance, the first result occurs after a whole page of exposition in the results section, at the end of the section “Provoking seizures in mice”, but is given in very vague terms as a high level synopsis, in a single sentence, but without any real detail, referencing only “Supplementary figure 3”. The fact that this figure is in the supplementary info is indicative of my point; it illustrates a key methodological step of the analysis of the real experimental data, and as such is more important to include than the “review-like” figure 1, which is in any case replicated schematically in panel A of figures 2-4.

R: We agree that line-length calculation and its in vivo application is a key methodological step in our study. We now include important methodological tools, such as the time-to-seizure and line-length earlier in the Result section and in Figure 1.

We agree that our study involved many methods, including computational, optogenetics in mice and electrical stimulations in humans. As a result, the first actual result appears at the end of the second paragraph in the result section. We have now shortened this paragraph and made the first result more prominent.

This pattern (giving the novel result as a synopsis, after rather long descriptions of the motivation behind the study and the model) is repeated throughout the results. If the results section focused more upon the experimental data, far more detail could be provided, and I think the paper would be improved greatly. Without this detail, however, I personally found it very difficult to get to the nub of what the novel results show.

R: We now added some detailed methodological explanations and numerical values to the results throughout the result section and emphasized the novelty of the results in one closing sentence at the end of each paragraph. We maintained the motivation drawn from the model, because many represent novel predictions that have not previously been tested. Describing the model's prediction at the beginning of each paragraph formalizes the tested hypothesis. This is now clearly stated in the manuscript (line 92).

For instance, in Fig 6H, it appears that active probing improves upon passive probing alone, but when both are used, if anything it seems slightly worse than active probing alone (although the stats for that comparison was not provided). I'd have thought that using more information should improve matters. This detail was not mentioned at all in the results text.

R: We agree that adding passive signatures to active probing does not improve predictability, most likely because all information can already be extracted with active probing alone. The 'active probing' model is based on the raw iEEG traces whereas the third one (the previous "combined") was trained on just the line-length values of the responses, not the raw traces. We redid the analysis with a new "combined" model which used both LFP traces and passive markers and found that a few animals improved when passive markers were added (Fig. 6H). Nevertheless, on average there was no significant improvement compared to the active probing classifier, showing that passive metrics don't add valuable information for most animals. We now use this model combining raw iEEG and passive signatures, instead of the prior model combining a single line-length value with passive metrics.

For many of the listings of specific results, it wasn't clear what was the sample size. Was this always "21 mice" as indicated early in the results?

R: We do want to be absolutely clear with the sample size for each tested prediction and apologize in case the sample size was not clearly stated for one or the other test. In our original submission, we did provide the number of mice (m) and sessions (s) in the text along with each statistical test. We now verified systematically that this is always the case. Most animal results (figures 2,3,4 and 6A-F) are based on 17 animals that all received BZD and from which 9 also received PTZ. The classifiers (6G-I) were trained on the 8 animals that received both BZD and PTZ in a randomized manner.

The "21 mice" referred to all animals where optogenetically seizure induction was attempted with the main viral construct ($PN_{MEC} \rightarrow CA1$). The 4 "extra" mice are animals used in Ext. Fig. 5 (previously 2) to test different stimulation parameters. They did not receive any GABAergic drugs therefore are not used in the main figures.

However, to improve the clarity regarding this point, we now removed this reference to “21 mice” in the main text and discussed only results for the 17 animals that receive PTZ/BZD (line 146). The mice used in a supplementary experiment are discussed in the supplementary data only.

Also, the authors state that only 1 human subject received BZD treatment, but for all the human data presentation, the implication was that there were multiple biological replicates. Please just state the sample size for each statistic presented.

R: This is a slight misunderstanding, we had 10 human patients, 7 of which had provoked seizures and another subset of 7 patients received single pulses before and after a benzodiazepine. This information appeared in the text of the results (line 157) and in Supplementary Table 1 of the original submission and in figures 3, 4 and 5. However, only one human subject had a seizure induction both with and without BZD (figure 2 D2). This is due to the fact that seizure inductions itself can be uncomfortable to patients and were induced only for clinical reasons, not for the direct purpose of this study.

I was interested that the description of prior work behind this study focused mainly upon high level theoretic models of critical transitions, which are agnostic to the detail of how these arise in cortical networks. In contrast, two other studies which used very similar techniques to those used here (optogenetic stimulation prior to the onset of epileptic activity) and which are thus highly relevant to this manuscript, namely Klorig et al, and Graham et al, were referenced, but virtually in passing, and without any proper discussion.

R: We agree that both these studies are important for our work as they both used response to optogenetic single pulses to measure cortical excitability. We now emphasize these results even more, throughout our article. Specifically, we include a discussion of possible mechanisms of pre-ictal step-changes in the fourth paragraph of the discussion (lines 355-9).

Both studies used *in vivo* models, although the referencing implies that the latter was only of *in vitro* studies (cf. lines 304-5).

R: We now revised our referencing of these studies to emphasize the fact that they were done *in-vivo*. In our original submission, we explicitly referred to the study by Graham and colleagues as also an *in vivo* study already in the introduction (previously line 101, now 96) and in the discussion (previously line 358, now 355) of the original manuscript. We now list the study by Graham et al., only as an *in-vivo* study in the introduction. We also changed the wording at line 304-5 to avoid the possible perception of it being solely an *in-vitro* study: *“Such gradual loss of resilience has been thoroughly characterized by imposing artificial ionic concentrations onto brain slices, which results in repeating ‘ictogenic ramps’ that invariably lead to seizure-like events.”*

The latter of these, in particular, presents a biological model of the altered optogenetic response in terms of a change in dendritic excitability (see also PMID: 36923333), and thus grounds theoretical models such as the Epileptor model in the actual biology of cortical networks. I am interested to know whether the authors consider their findings to support this biological model, or if not, how their data diverges from that study

R: In their paper, Graham and colleagues showed a pre-ictal increase in response to single pulses in two different seizure models, the zero-magnesium slice and focal injection of 4-aminopyridine (4-AP) both in slice and *in vivo*. Such findings resemble what we observed in our supra-threshold PTZ injections experiment (figure 7), and these concordant results across experimental models seem to indicate that such findings are not preparation specific, which

attests to the robustness of the effect. Although the work by Graham et al. was already explicitly cited in the discussion, we now emphasize it even more. We added a sentence in the discussion mentioning their biological findings: “*In the latter experiment, Graham et al. observed pre-ictal prolonged plateau potentials in dendrites, reflecting a calcium entry upon optogenetic probing*²⁹” (lines 355-6).

We believe our study and theirs are highly complementary. Although we don't have the fine spatial resolution provided by the calcium imaging used in Graham et al, we were able to observe changes in response across many nodes of the limbic network, both in mice and humans. This suggests changes in neuronal response at the cellular level are also reflected at the network level, which is easier to assess in a translational approach.

Another piece of biological data in the manuscript that deserves more attention is the failure to trigger seizures by optogenetic stimulation of PV interneurons. There have been a raft of publications showing that optogenetic stimulation of (subsets of) interneurons can trigger seizure-like events, although notably these have almost all been done in the presence of 4-aminopyridine (e.g. PMIDs 26134638, 25505119, 29024712, 25546300). I was interested therefore that this failed to occur in normal mice. This should be discussed with reference to these prior papers.

R: We now included the references suggested by the Reviewer in our manuscript (lines 171-2). We agree with the Reviewer that these results on inhibitory interneurons are interesting, but as stressed by reviewer 2, definitively proving that inhibition cannot induce seizure in non-epileptic animals would require other experiments and is beyond the scope of this paper. For a more in-depth discussion of these findings, please refer to the answer to reviewer 2 below.

The readability of the manuscript could be improved by being substantially shortened. At times, the writing is too grandiose for my taste: the first line of the abstract, for instance “volcanoes erupt, markets crash”. This paper has relevance to neither particularly, since no one is talking about stimulating mountains to test their resilience.

R: Although we don't study any of these phenomena, we feel that such sentences convey the idea that critical transitions are a dynamical phenomenon present in many natural complex systems. The statistical methods used in this paper as signatures of critical transitions (line length, variance, autocorrelation, skewness) have been developed in other fields, and their use is justified in our study partly by their successful application in other domains. Even though we are not the first ones to apply them to epilepsy, we believe it is interesting to stress the seemingly universal value of these straight-forward statistics. For these reasons, we kept the opening sentence of our paper as is.

I also do not like the repeated use of parenthetical statements which just repeat what is in the text, but using different wording (e.g. line 212 “Intuitively, as the recovery rate increases (a speed)...”). In other places parentheses were used to give the exact opposite meaning (lines 184f “decrease (respectively increase)...”). In both cases, I found this style quite distracting, and unhelpful. The topic of the paper is difficult enough as it is, and so the reader would be helped by making the writing as simple as possible.

R: We do aim at making the paper as readable as possible. We are aware of the fact that we discuss many concepts, and these parenthetical statements aimed at clarifying our point. The Reviewer suggests that they may actually add confusion and be contra-productive, we thus removed most of such parenthetical statements.

The equations in Fig1 are provided in the methods (and are more readable there), so should not be in the figure.

R: We now removed the equations from Fig. 1.

There are lots of syntactical and grammatical errors throughout the manuscript – the most common are many errors regarding mismatched nouns and verbs (singular and plural mismatches, e.g. line 199, “In one included human subjects...” and multiple other examples; also “included” is not really necessary), missing pronouns, and the occasional really poor sentence (“We transfected these each group of neurons using simple of intersectional transfections...”). Also figure 2 legend – “makers” should be “markers”. A proper proof-reading would help.

R: We apologize for the syntactical and grammatical errors. We believe that all errors were now found after proof-reading by native English speakers.

Reviewer #2 (Remarks to the Author):

This study aims to measure the resilience (i.e. seizure threshold?) using a variety of passive and active methods in a computer model, rodent model, and human epilepsy.

R: In our original submission, we did partially define neural resilience in Fig.1 and the result section: “The height of this seizure threshold determines a given level of resilience, here the distance from the trough to the crest (Fig. 1D-E).” We thank the reviewer for pointing out that a formal definition of the concept in the introduction was lacking. We now define neural excitability and resilience as follow in the second paragraph of the introduction: “*the amount of perturbation sustainable without causing a seizure*”(line 72).

The idea of measuring seizure threshold has very important implications, and this work has some interesting controlled experiments that show they can measure the threshold. However, the presentation really clouds this fact. The article tries to claim too much: the title is too broad, the figures spend too much time on examples or obvious findings, much effort is spent trying to prove a model prediction about resonance and inhibition that really is not important to the overall message, and even the phrase “limbic seizures” is an overreach.

R: Throughout the manuscript, we changed the phrase ‘limbic seizures’, a phrase often used in the animal epilepsy literature, to stick to a more accurate description of ‘hippocampal seizures’. We have also changed our title accordingly. We also agree on the Reviewer’s point on the importance of measuring seizure thresholds (see next point).

While there is some very important information here, and I really appreciate the ability to measure seizure threshold in a controlled fashion in animals and humans, the salient results appear to be limited, or at least they are not presented in a rigorous, summarized fashion.

R: We agree with the Reviewer that the ability to measure seizure thresholds precisely as a time-to-seizure is an important advance in our study. This represents the body of Fig.2, and the importance of this result has now been emphasized. In the paragraph *Probing variable neural resilience* in the result section, we now state: “*This set of results confirmed the model’s prediction on hippocampal resilience and emphasize the precision with which seizure thresholds can be measured in experimental and clinical settings.*”

However the study goes far beyond this finding. It shows that it is not necessary to induce a seizure to estimate the distance to the seizure threshold. Measuring seizure thresholds in mice and humans is likely relevant to develop interventions to raise it. Estimating the distance to the threshold with minimal perturbations opens the possibility of assessing the momentary state of the brain, and the risk of an ictal transition. It is a more translatable finding.

Most of my scientific concerns are focused on issues when the authors try to extrapolate from the model to mechanisms (e.g. lack of inhibitory drive because no resonance)—I do not think those discussions are necessary for this work to be useful and interesting; conversely I think it would require many more experiments to prove those things.

R: We agree with Reviewer 2 that evaluating the physiological mechanisms of ictogenesis is not the aim of the paper. Rather, we focused on the dynamics of the ictal transition and on developing tools for quantification of these dynamics. We now emphasize this in the first paragraph of the results. For this reason, we kept the experiment involving interneurons as part of the supplementary data in our original submission and in the current revision. As suggested by the Reviewer, we changed the subtitle “lack of resonance” to “integrative dynamics” and focused this figure on the ability of the hippocampus to integrate cumulative perturbations up to the seizure threshold. Indeed, proof for negative findings is notoriously difficult and not our goal here.

I recommend focusing on the methods (which look very useful) and how they seem successful under controlled conditions rather than making statements about mechanisms (which cannot be proven well in these studies).

R: We agree with the Reviewer that the emphasis here should be placed on the methods rather than the mechanisms. We explored many and focused on line-length and time-to-seizure after realizing that 1) they are applicable to experiments and robust, 2) they reflect our dynamical understanding of the brain responses. To emphasize the importance of these methods, we now present them in Fig. 1 and the first paragraph of the result section.

Many examples here in test resilience over time. But repeated tests can change resilience. Did they randomize the order, or somehow account for this?

R: Out of concern that repeated tests of the resilience could change resilience, we did randomize the order of the pharmacological conditions as explained in the methods section of our original submission (line 747).

Throughout the paper there is a lot of talk about feedback loops. But Epileptor does not inherently have a feedback loop.

R: Positive feedback are a key component of systems that present ‘catastrophic bifurcation’ (Scheffer 2009). The Epileptor itself is composed of subsystems (i.e. the equations) that depend on each other (i.e. the variables link the equations), creating feedback loops and the emergence of runaway dynamics.

And did they actually test the loop in the experiments? It is not apparent to me if they did.

R: The high interconnection of the entorhinal-hippocampal excitatory neurons is well established. Feedback loops between CA1 and the MEC have been thoroughly studied both in epileptic and non-epileptic animals before (Ang, Carlson, and Coulter 2006; Lu et al. 2016; Wozny et al. 2005). A long-standing hypothesis is that this organization of the connections leads to the possibility of recurrent excitatory loops with the development of runaway excitation across the circuit (McCormick and Contreras 2001). We now include these references. In our experiments, we provide histological data that attests to the fact that we transfected these well-characterized connections (Supp. Figure 3).

L 65 The following sentence is written in very confusing fashion “Therein, finetuned positive neuronal feedbacks ensure sensitivity to weak yet relevant inputs, at the cost of low resilience

to perturbations”. Specifically, “finetuned”, “ensure sensitivity”, “relevant”, “cost” and “low resilience” are all very loaded words that could be interpreted in a variety of ways.

R: We thank the Reviewer for pointing this out, and now reformulated this sentence to “*Therein, positive neuronal feedbacks can amplify weak yet relevant inputs, but consequently increase the vulnerability to seizures*” lines 68-69.

L 97 The following sentence implies that seizure prediction algorithms have failed, which is not true (there are many successful algorithms...the problem is a lack of prospective data). “has defeated many machine-learning algorithms”

R: Our lab being active in the field of seizure forecasting, we agree with the Reviewer that seizure prediction algorithms have not failed. We however want to emphasize that finding precursor signatures has been a challenge. We now changed the sentence to: “*The decades-long search for such warning signs heralding ictal transitions in EEG of patients with epilepsy has challenged the best machine-learning algorithms*” (lines 90-1).

Why does this article focus on the “limbic system” rather than ‘temporal lobe epilepsy’? The latter is by far the more common terminology.

R: The reviewer is right that several structures classically incorporated in the limbic system (thalamus, amygdala, cingular cortex, retrosplenial cortex, pyriform cortex, etc.) are not recorded or probed in this study. As the temporal lobe could also refer to some extra-limbic structures, we opted instead for the terminology ‘hippocampus’ instead of ‘limbic system’ and ‘hippocampal seizures’ instead of ‘limbic seizures’.

L 120 why a “fold bifurcation”. Isn’t the Epileptor a saddle node bifurcation? Fold is jargon here.

R: Technically, a saddle-node bifurcation is a fold bifurcation (Izhikevich 2006). We opted for the appellation “fold bifurcation” because it is the one typically used in the critical transition literature (Scheffer 2009; Scheffer et al. 2009, 2012) including in some epilepsy articles (W. C. Chang et al. 2018; Meisel and Kuehn 2012), We believed that both terminologies would sound jargon for an unfamiliar reader, but the “fold” might be more intuitive as it is visible in the “S shaped” bifurcation diagram.

“resilience” needs to be defined in the introduction. It is used throughout the paper.

R: We thank the Reviewer for highlighting this point. A proper definition of resilience was indeed lacking in the original introduction, and only present later in the result section. It is now included in the introduction as follows: “*the amount of perturbation sustainable without causing a seizure*” (line 72). We used this definition after Scheffer and Holling’s work (Holling 1973; Scheffer 2009). It is also the one previously applied to the context of epilepsy by Chang and colleagues (Chang et al., 2018).

L 127. The issue with provoked seizures is that you cannot be certain it is the same seizure as a spontaneous one. This line reveals the problem—it can be ‘another path over the threshold’. Since “any brain can seize”, this becomes a serious issue. How can you be sure the provoked seizures are the same?

R: We agree this is an important point and we added an additional panel to figure 1 and a new supplementary figure (Supplementary Fig. 4) to address it. Both in our animal experiments and human study, the electrographic and clinical manifestations of provoked and unprovoked seizures were very similar. Electrographically, provoked seizures do contain all the classical EEG hallmarks of seizures: spike and waves, high frequency fast activity or post-ictal depression. An example of a provoked and an unprovoked seizure in a human patient is now

provided in Fig.1. In humans, it is clinically relevant to determine whether the provoked seizure is typical or atypical, as only the former has diagnostic value (Cuello Oderiz et al. 2019). This information was provided in the original submission in the Supplementary table 1. We also specified this point in the result section: “Similar to results in healthy mice, we found that seizures were inducible in the human hippocampus with train-stimulations (bipolar, biphasic pulse-width 1s, 1-5mA, 60Hz) over a median [range] duration of 3.0 seconds [1.0-4.0] (N=7, Fig. 1A-B, Supp. Table 1)”(lines 154-7).

We now also provide a new supplementary figure (Supp. Fig. 4) to show the spectral overlap between seizures starting after optogenetic stimulation (over the threshold) and after PTZ injection (through the critical point, around the threshold). The seizure semiology in mice was the one classically described in other animal models of temporal lobe epilepsy and could be scored using a Racine scale ranging from no behavior changes or freezing to “wild jumping”.

L 132 The model of provoked seizures is highly non physiological. What is the goal of this section? It needs to be clear that this is a limited model and describe why it is being used
R: Unprovoked seizures and provoked seizures are non-physiological manifestations of brain dynamics. Provoking seizures with optogenetic perturbations does not happen in real life, but provoking seizures in humans for diagnostic reasons (epileptology) or a side-effect of certain experiments (e.g. TMS) or therapies (e.g. cortical stimulation) does occur. The optogenetic-provoked seizures here represent our testable model, used to measure resilience to seizure *in vivo*. The goal of this section is to present the model and describe briefly the methodology used. It explains how we measured resilience which is a crucial part of our study. We believe that optogenetically-provoked seizure is a precise and good model, with some limitations, just like all models of seizures have limitations. We now have included the following sentence to the limitation section of the discussion: “*Third, our optogenetic model of seizures-on-demand enabled circuit and cell-type specificity, but likely does not represent the sole neuronal mechanism of ictal transition*” (lines 387-9).

To better characterize provoked seizures and compare them with unprovoked ones, we now added the new Supp. figure 4.

L145+. Combining time to seizure, resilience, path length, and line length is unproven and seems unjustified. (see comments for Ext Fig 3)

R: We address these important points in the response concerning the Supp. Fig. 5 (previously 3, see below)

From the video this seizure does not look like an Epileptor seizure. It starts not with a saddle node bifurcation (like Epileptor), but instead looks like a SNIC. The offset likewise does not have the slowing down of a homoclinic bifurcation, but has arbitrary firing like a Fold Limit Cycle

R: To better characterize the seizure shown in the video, we looked at the inter-spike intervals as previously done by Saggio and colleagues (Saggio et al. 2020) (see figure below). Although the first interval is indeed higher than others, we didn't find a consistent scaling of the spike frequency at the seizure onset as would be expected in a SNIC. More generally across our recordings, we observed the absence of a scaling law at the seizure onset both in our optogenetic and PTZ seizures (see Supp. Fig. 4). Regarding the onset baseline jump, our recording settings contain a high-pass at 0.5Hz making the observation of DC shift difficult, therefore we could not use this feature to distinguish saddle-nodes.

Concerning the offset, we typically observed a slowing down in the frequency (but not amplitude) of the spikes toward the end of the seizure (see Supp.Fig. 4), suggesting a homoclinic bifurcation (Saggio et al. 2020). We nevertheless stress that our work here focuses on the seizure onsets and not on the offset bifurcation.

L 154 It is unclear what conclusion we can make about human seizures starting with 3 seconds of stimulation. Even assuming the system is integrating, it is total area under the curve, i.e. pulse width x pulses x amplitude. Amplitude in the human stim cannot be compared with the mice. The electrodes, amps, volumes of tissue are all completely different. This is not a valid comparison with the mice.

R: We acknowledge that our data on seizure induction in humans are constrained by how seizures are induced in clinical practice. In our original submission, we included a sentence in the discussion to put the emphasis on such limitation. We do not try to directly compare the seizure resilience between animals and humans, but observe that in humans too seizures can be induced by a short train of perturbations. This opens the possibility of quantifying seizure threshold similarly.

In general we are not claiming to have a perfect comparability between the human, animal, and simulation experiments. We merely show that beyond the technical differences (e.g. electrical versus optogenetic stimulation versus inputs in differential equations), some statistics in the signal and relationships between these statistics (e.g. line-length and time-to-seizure) are similar.

Furthermore, the text implies that there is a fixed seizure threshold, but in fact half (2/4) of the subjects tested twice had DIFFERENT thresholds. This whole section is very weak.

R: Only two human subjects (EL011 and EL015) underwent multiple seizure induction protocols on the same electrodes and in the same pharmacological condition, and both showed fixed seizure thresholds. We agree that such a low number of observations cannot be used to draw general conclusions, but we merely refer to them as “*suggesting*” (line 158) the presence of a fixed seizure threshold.

For the two others, we believe that the Reviewer is making reference to EL008 and EL017. For EL008, the two sessions are made on different electrodes, respectively in the left and right hippocampus, as indicated in Supp. Table 1. We do believe that seizure thresholds are

brain region-dependent and can be different due to technical reasons (e.g. exact positioning of the electrodes). We are therefore careful to only compare repeated stimulation in the same region and the same electrodes.

EL017 is the patient where a seizure induction protocol was also administered in the BZD condition, therefore the seizure threshold change is in line with the idea that GABAergic drugs affect the seizure threshold. These are the seizures shown in Figure 2 D1.

Fig. 1. The numbers in E describe some very complex details that really have no explanation in the text: resilience, recall flow, stability landscape, etc. This is probably too much detail unless there is going to be an explanation in the appendix.

R: As suggested by the reviewers, we now moved this panel in the supplementary materials and replaced it by a simpler version.

What do you mean “no need for kindling”? That these mice would seize with the first-ever exposure to pulse trains? That is what you say

R: Yes, this is what we meant. No previous exposure to train stimulations were necessary to induce seizure. We thought this sentence might be important as experimentalists often used kindling protocol including with optogenetics (see Cela et al., 2019 for example). This was not necessary in our case.

Ext Fig. 1, E&F Why is ER bright only in E and CA1 bright only in F?

R: These pictures are taken from the same mouse’s brain, at different axial coordinates. When sliced in the axial axis, the CA1 field of the dorsal hippocampus and the medial entorhinal cortex are not at the same depth (z-coordinate), respectively ~ -1.40mm and ~ -2.96mm from Bregma (Franklin and Paxinos 2008).

Ext Fig 2: Where does the statement come from that integrators don’t have inhibition induced transitions but resonators do?

R: These statements come from Izhikevich, “Dynamical Systems in Neuroscience: The Geometry of Excitability and Bursting” 2006, and in particular the chapters 1.2.4 called “Neurocomputational properties” and 7.2 called “Integrators vs. Resonators”(Izhikevich 2006).

Here is a table taken from the latter, from which our table in Ext. Fig. 5 (previously Ext. Fig. 2) is inspired.

properties	integrators		resonators	
bifurcation	saddle-node on invariant circle	saddle-node	subcritical Andronov-Hopf	supercritical Andronov-Hopf
excitability	class 1	class 2	class 2	class 2
oscillatory potentials	no		yes	
frequency preference	no		yes	
I-V relation at rest	non-monotone		monotone	
spike latency	large		small	
threshold and rheobase	well-defined		may not be defined	
all-or-none action potentials	yes		no	
co-existence of resting and spiking	no	yes	yes	no
post-inhibitory spike or facilitation (brief stimuli)	no		yes	
inhibition-induced spiking	no		possible	

Figure 7.15: Summary of neurocomputational properties.

The features that we actively tested in Ext. Fig. 5 are highlighted in yellow.

Applying that statement to real seizures/physiology seems a major and unsubstantiated reach, and it is frankly unclear why it even matters. Are the authors trying to use a dynamics paper to prove that interneurons cannot produce seizures? And this issue becomes a major roadblock in the rest of this paper, because the authors try so hard to prove it, but ultimately do not.

R: We apologize if the goal of the paper came as aiming at proving that interneurons cannot produce seizures. This was not our intent and one reason to have the interneuron experiment as supplementary data. We believe that both pyramidal neurons and interneurons play an important role in seizures, as they form a densely interconnected network. Interneurons stimulations is only one out of a series of three experiments (frequency effects, irregular stimulations and interneurons stimulations) designed to confirm that we have integrative dynamics in the hippocampus. We do agree with the Reviewer that we cannot rule out resonance entirely and have revised the text accordingly (see also reply to comment above).

What is an “inhibitory stimulation” in Epileptor? a negative pulse? What is it in vivo? Do those two situations really agree?

R: Inhibitory stimulations in the Epileptor are indeed trains of negative pulses on variables x_1 and x_2 , as was done for positive pulses. *In vivo*, we used trains of optogenetic stimulations on PV interneurons. Other biological options could have been chosen as the Epileptor doesn't contain explicit models of different neuron types (Jirsa et al. 2014).

Showing similar frequency characteristics between Epileptor and physiology is not valuable, since Epileptor has arbitrary temporal values.

R: Indeed, the Reviewer is right that Epileptor has arbitrary temporal values. Yet what matters is the respective order of magnitude of the time scales involved in the model, as they influence how bifurcations are crossed and constrain our modeling choices. To prove that the hierarchy of time scales set by the Epileptor model corresponds to our *in vivo* findings, we decided to match a few general features of our *in vivo* model such as:

- 1) A length of perturbation induced by each pulse in the order of hundreds of ms (200-300ms, see Fig2 B1-D1).
- 2) The ability to start seizure after a few seconds of 20Hz train stimulation.
- 3) Seizure that last around 30s

I do not understand what the point is about the statement with 100 Hz stim into ChETA (lines 950-952). What was the result?

R: In the interneuron experiment, we have used a modified channelrhodopsin (ChETA) that can fire at high frequencies (e.g. 100Hz). Classic channelrhodopsin such as Ch2R saturate for frequency higher than 40-50Hz (see Gunaydin et al., 2010). Since PV CA1 interneurons can fire at high frequencies, we believe it was important to also entrain this neuronal population at faster frequency.

Thus there is a lot of data in this figure but much of it seems arbitrary and it does not really prove what it claims to prove (that Epileptor successfully models these seizures?). In addition, the title of this figure is NOT proven by the data ("lack of resonance in the limbic system"). That title should be changed.

R: We now changed the title as suggested by the Reviewer to *Integrative dynamics in the hippocampus*.

It seems like the authors are making several unsubstantiated leaps (the model predicts only bifurcations with resonance can start with inhibitory stimuli, then they want to say the physiology has no resonance, meaning it cannot start with inhibition) to make strong claims about mechanisms. This is unsubstantiated by their data and requires several leaps of faith between the model and physiology. But I don't see why it's important to show this at all.

R: The goal of our study is to characterize the dynamical properties of the mouse hippocampus, while disclaiming that we cannot directly translate this to a unique biological mechanism. In fact in our original submission, we have clearly stated in our discussion that "*While thorough, our study is nevertheless limited. First, our manipulation of neural excitability relied solely on the use of modulating GABA-A receptor function. However, in our opinion, excitability is best conceptualized as a latent parameter that integrates the effects of a large number of variables, including endogenous cyclical fluctuations, pathological changes⁴⁵, and pharmacological modulation⁵⁷. As such, our results provide one of potentially many crucial links between tangible biological mechanisms and neural excitability while establishing a quantitative framework to assess others*" (line 378-83). We have now added another sentence to show the limitation about cellular mechanisms: "*Third,*

our optogenetic model of seizures-on-demand enabled circuit and cell-type specificity, but likely does not represent the sole neuronal mechanism of ictal transition” (lines 387-89). In addition to these limitation statements, we have followed the Reviewer’s advice and we have de-emphasized the fact that we could not induce seizures by stimulating interneurons.

Nevertheless, what this figure does show is that it appears to be the integration of positive input that leads to a seizure, at least when that input arrives fast enough not to dissipate. This really seems to be the crux, but it is not explained anywhere.

R: As stated in another response above, we followed the Reviewer’s advice and reframed this figure and the corresponding text as evidence for integration of cumulative perturbations instead of lack of resonance.

Proving that resonance is not present will require much more meticulous testing, even in the Epileptor (all you have done is show integration is present, not that resonance is absent).

R: The absence of resonance in the Epileptor is a formal consequence of the nature of the onset bifurcation (fold). Because of this, it can be analytically shown that the eigenvalues of the fixed point near the bifurcations are real, which means that there is no resonance effect. Indeed a saddle-node bifurcation corresponds to an integrator, not a resonator. This is a well-known result (Izhikevich 2006).

Proving this would require a vast range of frequencies to disprove resonance, then highly meticulous control of interneurons over a vast range of potential conditions to disprove inhibitory drive (and note that several people have PROVEN inhibitory drive! (e.g. Marco de la Curtis, Ben Ari)).

R: Although we agree that other authors have greatly emphasized the role of interneurons at the onset of seizures (M. Chang et al. 2018; Sessolo et al. 2015; Shiri et al. 2014; Yekhlef et al. 2015), they studied the early involvement of interneurons in seizures in highly unstable cortex, after application of the chemo-convulsant 4-AP. In contrast, we looked at the possibility to start seizure with interneurons in non-epileptic brains where cortical physiology is stable. To our knowledge, such data was not reported by these groups, and these key experimental differences could explain the different results.

Please see here the result of a literature review about optogenetic seizure induction in non-epileptic animals. No paper inducing seizures with interneurons in non-epileptic animals was found.

FIGURE 1.4: Optogenetic induction of seizures

Review of in vivo optogenetic seizure induction. PubMed research using the search: (optogenetics) AND (seizure). From the 230 results, 15 studies were then manually selected based on abstract and paper inspection. Inclusion criteria were: 1) in vivo studies 2) with data in non-epileptic animals and 3) in absence of pro-convulsive drugs. Seizure criteria were the same as in our study: sustained (>10s) electrographic ictal activity after the end of stimulation. Color-code indicated cell type stimulation, italic font indicate study where kindling protocols were necessary prior to the first induce seizure. With these criteria, no study showing seizure induction by interneurons stimulation has been found.

We now explicitly specify that our findings are in non-epileptic mice and only *suggest* an absence of inhibitory drive (line 176).

And proving anything occurs “in the limbic system” will require many more in vivo experiments (simultaneous recordings in ANT, cingulate, amygdala, hippocampus, parahippocampal gyrus, etc). The paper really does not do this, it’s all just temporal lobe and should be presented as such.

R: We agree with the reviewer's point on the “limbic system” nomenclature and change it to ‘hippocampal’ in all the sentences referring directly to our results.

Ext fig 3: C - F do not make sense to me. Are they swapped? Even then I am confused.

R: We thank the Reviewer for pointing this out. There were some mistakes in the lettering of the different plots and their corresponding legends. They are now corrected and we apologize for the disagreement.

In G-H, the idea that “cumulative LL” corresponds to seizure threshold is brought up. This would be very helpful, but I do not see any systematic proof that this works, only a hand-waving figure here.

R: Using simulation in the Epileptor, we originally showed that the “cumulative line-length to seizure” extracted from the time series correlated with the actual “cumulative path-length to seizure” in the phase space (previously Supplementary Fig. 3F, now Supplementary Fig. 2E). The cumulative path-length is defined as the total distance traveled in the model’s phase space (5 dimensions) up to the seizure threshold.

Additionally, we are now showing in a new graph (Supp. Fig. 2G) that the “cumulative line-length” metric is related to the x_0 control parameter of the Epileptor model, which represents the epileptogenicity, setting the distance to the seizure threshold.

Of note, I am quite sure it will not work for all time scales, i.e. a very slow, long pulse train might reach the same LL as a short, fast one, but only the fast one will cause a seizure. Simply “calling” this ‘distance to seizure’ is not enough—it needs to be proven better.

R: We thank the Reviewer for the interesting point made on the relation between the accumulation of perturbation and their dissipation. We address it by looking at how our “cumulative line-length” metric changes for different stimulation frequencies in Epileptor (Supplementary Fig. 2F). As expected by the Reviewer, for a given level of excitability (x_0), the cumulative line-length to the seizure was frequency dependent. Nevertheless, for a given frequency, the relationship between the cumulative line-length and excitability was linear, showing that the relationship holds for a given frequency. Specifically, for low frequency (<10Hz), seizures couldn’t be triggered due to dissipation of the perturbation between two pulses. For frequency between 10 and 20Hz, the ability to induce seizure was dependent on the underlying level of excitability (x_0 , represented by different shapes in Supplementary Fig. 2F). Interestingly, such results correspond to what we also observed in mice (Supplementary Fig. 6E)

Based on these findings, we now state clearly that both “cumulative line-length” and “time-to-seizure” metrics are frequency dependent, and therefore the same stimulation frequency should be used to compare seizure thresholds across different experiments. For this reason, we also stick to the “cumulative line-length” terminology instead of ‘distance to seizure’.

L 167 “From a dynamical standpoint, the limbic system is thus bistable in healthy mice and human subjects.” The study did not test healthy human subjects (which is implied by the phrasing). The study did not test “the limbic system” but instead just a small region of hippocampus.

R: Indeed the phrasing was ambiguous. For clarity and simplicity, we now changed it to :
“Thus, from a dynamical systems perspective, the hippocampus exhibits the characteristics of a bifurcation.”

L 173-180: The small experiment with ChETA tested whether fast PV spiking can cause seizures. It did not. But this did not prove “lack of resonance” nor that inhibition cannot cause

seizures. It tested only a single frequency, and as stated before the statement about inhibition has questionable application to real physiology. Overall, I do not see the purpose of this line of experimentation: why do you need to disprove resonance and inhibition (which is VERY hard)? The authors do not even have a model that has resonance that would allow them to test it in silico...why are they looking in vivo? Why not just focus on the positives and the model they already have? This section is weak.

R: On both these issues, please see the response made above. In the ChETA experiment, we didn't test a single frequency but seven different frequencies (4, 7, 10, 20, 40, 70, 100Hz).

L 183-202: These experiments show that the length of stimulation equates to seizure threshold, and that PTZ lowers/BZD raises the seizure threshold. These experiments are straightforward and lend credence to the stimulation duration as a surrogate of seizure threshold. However, the results are exactly as expected. What I do not understand is whether this really “probes neural resilience”. Although the neurophysiological literature has been quite lax about what “resilience” means, the authors need to be specific here. These experiments simply seem to be measuring seizure threshold—is that what they mean by “resilience”?

R: By resilience, we mean “the maximum perturbations that can be absorbed without transitioning to an alternative state”. This definition is now highlighted in the text (line 72). For fold bifurcations which have fixed threshold, this corresponds to the distance from the equilibrium point (bottom of the basin of attraction) to the seizure threshold. See our reply above for the correlation between line-length and the distance to the seizure threshold.

More importantly, Fig. 2 has 9 panels, but 8 of them are just examples. The only “data” are in C4. Is that the entire summary of the experiment? I cannot read that tiny figure, which has a mean difference to NaCl and bootstrapping.

R: For the sake of transparency and clarity, we deliberately opted to show each step of our quantification as example stimulations in in-silico, in mice and in humans. We agree with the reviewer that these panels take up some space, and we now increased the readability of panel C3 (previously C4). Following the Reviewer's advice we also added more data in this panel by adding the quantification of the effect of different doses of Diazepam on the ‘Time-to-seizure’.

Additional data going with this figure are the quantification of cumulative LL shown in Supp. Fig. 2 K. We put them in supplementary as we found they were redundant with ‘Time to seizure’, and only interesting for readers who want to make a parallel with the model's state-space.

We also measured seizure severity (based on a modified Racine scale) and duration, these data are in Supp. figure 6. We didn't include them in the main figure as they are not directly related to seizure resilience.

What is the difference from “normal”?

R: As time-to-seizure can vary across animals, data in C4 are expressed as percent change compared to the control session (NaCl) in the same animal and same week. Sessions were randomized. Raw data are shown in Supp. Fig. 6.

This figure should be focused on the summary results, not simply rehashing the concepts from prior figures.

R: Following the Reviewer's advice, we redesign our figures 1-3 to integrate the conceptual and methodological panels in the first figure 1, and focus figures 2 and 3 on the summary results

L204-222: This section and Fig 3 assess how the systems respond to single pulse stimuli, essentially measuring the evoked potential with line length. The results are that PTZ makes the LL longer, and Benzos make it shorter. These are straightforward and unsurprising results. I do not understand the use of AUC in 3C3 and 3D3. AUC is usually from an ROC curve—which curve? I also do not understand the scatterplot in 3C3.

R: Indeed our results were in-line with the theoretical prediction, and strengthen this framework. Here the AUC is used as a way of quantifying responses across different stimulation intensities. They are not calculated on ROC curves but on the input-output curves shown in 3C3 and 3D3. This is explained in lines 209-212. To avoid any confusion, we now rename this metric IOC ('input-output curve') instead of AUC. We used this measurement as it is more robust and less arbitrary than stimulating and measuring LL at a single intensity. Instead, the IOC captures the entire input-output curve.

The scatterplot in Fig. 3C2 shows the quantification of these IOC across all animals and sessions.

L 224-273 show that the PTZ/BZD changes the background activity and connectivity, which are not surprising.

R: Again, our results were in line with the theoretical predictions. These statistical markers of critical slowing were already shown by other groups (W. C. Chang et al. 2018; Maturana et al. 2020; Meisel and Kuehn 2012), but we now demonstrated that they correlate with the level of GABAergic modulation, in controlled pharmacological experiments.

L 274-292: This section now applies all the prior methods, which were not very surprising or informative, to make a classifier of dynamics. This is the first section that has intriguing data.

R: We thank the Reviewer for highlighting the interest of this section. We firmly believe that the prior sections find their importance in confirming each theoretical prediction that lead up to this machine-learning section. Had we not constructed our study this way, would we have ended up with another obscure machine-learning result.

As in most prior figures (which is a weakness), Fig 6 is mostly examples and the actual results are relegated to a corner and hard to read. Though it is hard to decipher, I believe what they have shown here is that these quantitative methods are able to discern between animals with BZD, PTZ, or nothing. That is really important and needs to be much clearer. And the results are extremely hard to interpret.

R: We agree with the Reviewer that this is an important figure that builds upon all prior figures. In figure 6, only two panels out of nine are examples (B and C). We now increased the size of panels G-I to improve their readability. The Reviewer's interpretation of the figure is correct: we used a machine learning algorithm to discern between pharmacological conditions, based on single trials of short iEEG segments. One detail 'nothing' is actually an injection of NaCl 0.9%, as a 'sham' pharmacological control.

Were there held out data or is this overtrained?

R: The classification was done on test data using standard cross-validation procedure. Information concerning the classifier design and testing can be found in the method section: "For each mouse, one classifier was trained to discriminate between three classes (NaCl,

subconvulsive PTZ and BZD) using a 5-folds cross-validation strategy”. In each fold, 80% of the data were used as a training set and 20% were kept as a test set. We now specify that the average accuracy was for test data.

Just how good is the classifier? What do these performance measures mean?

The F1-score is a well established metric of performance for machine learning classifiers (see (Subasi 2019)). It combines two important metrics, precision and recall, into a single value to provide a balanced measure of a model's accuracy. Its superiority to other metrics such as accuracy reside in the fact that it is less vulnerable to imbalance in the dataset. However, in our case, the datasets are strictly balanced as there is exactly the same number of observations in each class. To improve clarity and readability, we now opted to use accuracy as our main performance metric, which is likely more widely used than the F1-score and can be directly interpreted as the proportion of samples correctly classified. An accuracy of 0.83 means the classifier found the correct label for 83% of the observations. As it is a three-label classification problem (BZD, PTZ, or NaCl), the chance level would be at 33%.

Why is the multivariate “combined” worse than just active probing (6H)?

R: For the issue regarding the “combined classifier”, please see the answer to reviewer 1 above.

L 294-314: These are very interesting experiments, made to assess whether they can measure seizure threshold in a controlled fashion. These results are quite intriguing. However, once again the summary result (the most important) is relegated to small parts of the figure. I recommend making it much more clear how these measures are functioning as a biomarker of seizure risk.

R: We thank the Reviewer for this suggestion, we have now added this sentence to the discussion: *“The importance of such measurements is to be found in their ability to reflect the risk of upcoming seizures”*.

One important thing to discuss is how to account for if one of the SPES triggered the seizure (which is certainly likely in these PTZ animals). Would that SPES be removed from analysis? How would you know if it was the trigger?

R: As explained in the original result section, some seizures did start from a single pulse stimulation but this was not the majority (6 out of 18 seizures). SPES that triggered seizure were defined as SPES followed by the start of a seizure in the next second. These SPES were removed from the analysis.

Discussion:

The statement “fundamental mathematical predictions that are domain agnostic” is vague and should be rewritten.

R: We revised this sentence: *“In this study, we verified how fundamental mathematical predictions on fold bifurcations may apply to hippocampal seizures in mice and humans in vivo”* (lines 317-8).

L. 330: This study did not prove lack of resonance. It is also unclear why you need to prove it was an integrator.

R: See responses above, the discussion was changed accordingly.

L 338. This study did not confirm that increased excitability decreases [i.e. ‘causes’ a decrease in] resilience [which is never defined]. It confirmed that some types of increased

excitability are correlated with seizure risk in these limited models. This statement needs to be heavily qualified

R: We rephrased this paragraph (4th) of the discussion, and now have included a sentence on our seizure model in the limitation section of the discussion: “*Third, our optogenetic model of seizures-on-demand enabled circuit and cell-type specificity, but likely does not represent the sole neuronal mechanism of ictal transition*” (lines 387-9).

As discussed above, a proper definition of resilience was added in the introduction.

L. 345. The study did not show the measures had precision. Perhaps they did—but that was not demonstrated by the figures. Is there a dose response relationship between the measures and the drug? That would be tremendously important.

R: We did find a dose-response relationship with the dose of BZD for both our metrics of seizure resilience and neural excitability. These results are shown in the Ext. Figure 7 (previously Ext. Figure 5) and main Figure 6 (panels D-E-F). We now also integrated them to the main Figure 2 (panel C3) to support the claim on precision.

Reviewer #3 (Remarks to the Author):

Start of results. Is epileptor a neural mass model or a phenomenological model of the EEG?

R: The Epileptor is a phenomenological model. According to some taxonomy, it can also be called a neural mass model in the sense that it recapitulates the local field potential dynamics of a brain area, similarly to for instance the Wilson-Cowan neural mass model. To avoid any confusion, we remove this terminology from the results section.

Figure 1. It would be helpful to know what the control parameter/neural excitability represents in the context of the epileptor variables.

R: In the Epileptor, the control parameter is set by the excitability (also called epileptogenicity) parameter x_0 , this was explained in the method section but we now added this information to the Fig. 2 legend. The effect of the control parameter has been extensively characterized before (El Houssaini et al. 2015; Proix et al. 2014).

There seems to be a grammar error in this sentence: We transfected these each group of neurons using simple of intersectional transfections with adeno-associated viruses (AAV) carrying Channelrhodopsin 2

R: Thank you for pointing out this mistake, we changed the sentence.

What evidence do you have that epileptor is a good model of limbic seizures?

R: One of the goals of the study is to experimentally verify a number of predictions made by Epileptor, which consolidate the notion that Epileptor is relevant to hippocampal seizures. The Epileptor is a model that seems to replicate many invariant features of seizures (Jirsa et al. 2014) and has been shown to predict well seizure onset, propagation and termination in humans (Proix et al. 2017, 2018) including temporal lobe epilepsy. It also corresponds to one of the most commonly found dynamotypes (Saggio et al. 2020).

Figure 2, what is the link between neural excitability in the epileptor model and the duration of optogenetic/electrical stimulation? Is neural excitability a variable in epileptor? and is that

variable somehow related to duration of optogenetic/electrical stimulation? i.e. without considering every single detail what is the intuitive biophysical relationship between the two? Neural excitability is a property of the model. Optogenetic stimulation is not part of the model. Is the control parameter x_0 a measure of neural excitability? Why? The manuscript should make these points clearer.

R: Yes, indeed the control parameter x_0 is setting the level of neural excitability in the Epileptor. 3 different levels of neural excitability (x_0) correspond to 3 pharmacological conditions in vivo (PTZ / BZD / NaCl). In the Epileptor, the parameter x_0 is set independently from the train stimulations and is not affected by them. The “optogenetic” stimulation simply assesses the resilience-to-seizure for a given excitability level (x_0).

Figure 2, "Quantification of changes in resilience (time-to-seizure) across mice (a) and sessions (s)". Not sure what "(a)" and "(s)" is referring to in the figure.

R: (a) is the abbreviation for “animal” and (s) for “session” and are used in the figure to report the sample size. In this sentence, we changed to “mice” to avoid any confusions.

Supplementary figure 2. In the epileptor model, what does it mean to provide excitatory stimulation vs inhibitory stimulation? In B4 why stimulate epileptor for 0.75s while in C4 you stimulate for 3.5s? In the caption you say validate the epileptor prediction but hasn't epileptor been designed to produce/mimic this data? Normally a model would be fit to some data that it could explain and then it gets applied to something new where if it explains something new, that can be considered a prediction.

R: Excitatory or inhibitory stimulations in the Epileptor model means that a respectively positive or negative input current is added to the equations through the variables I1 and I2. The Epileptor is a previously developed model (Jirsa et al. 2014), designed to qualitatively reproduce a number of features of epileptic seizure recordings, which did not include electrical stimulations. The model has not been developed for “optogenetic” seizure induction and to our knowledge it actually has never been used to induce seizures with trains of stimulation, but rather by setting x_0 such that spontaneous seizures occur.

Except in Fig. 7, we used x_0 values where no spontaneous seizures occur. The behavior of the model when stimulating is a consequence of the onset fold bifurcation that was chosen to reproduce other features of seizure dynamics. The originality of our work resides in the fact that we explore the model dynamics when stimulated and directly compare it with *in vivo* observed dynamics.

The stimulation time necessary to induce a seizure (time-to-seizure) is dependent on the control parameter value (x_0) but also on some other parameters chosen to run the model that were kept consistent across all simulations. We didn't choose such values to exactly match quantitative *in vivo* findings but rather to reflect a general hierarchy of time scales (for more on this, please see response above to Reviewer 2).

Supplementary figure 3. The column titles '...in silico' and '...in vivo' seem to clash with the caption descriptions of A to K. For instance the caption description in C refers to in vivo but it is in the 'in silico' column and G seems to refer to the model but then is in the 'in vivo' column. The caption needs to better clarify what is in silico and what is not.

R: There was indeed a mistake in the legends, we apologize for this. For more details on this, please see the response to Reviewer 2.

9. typo...within on average a median [range] of 3.0 seconds

R: Thank you for pointing this out, the typo has now been corrected.

10. You say "From a dynamical standpoint, the limbic system is thus bistable in healthy mice and human subjects." This seems overly confident. What do you mean by this? Up to this point you mainly showed how time to seizure properties are similar in silico and in vivo. Why does this imply bistability? Why not multistability or something else?

R: By bistability, we meant that for a given level of neural excitability, the system has two different regimes (ictal and non-ictal) and that this can be revealed by forcing the system to transition through a sustained perturbation (i.e. electrical/optogenetic stimulation).

The reviewer is right in saying that technically we didn't exclude the possibility that there are even more possible regimes. We now remove the word *bistability* from the discussion and describe seizure as a "*latent regime*" (line xx).

Also, doesn't electrical/optogenetic stimulation drive fast neural activity rather than a change say in slowly varying physiological parameters like synaptic strength, so are we really moving from one fixed point to another via a parameter change?

R: As shown in Figure 1D or 2A1, the electrical/optogenetic stimulation does not change the control parameter (x-axis), just forces the system in the other alternative state (i.e the seizure) without a parameter change. The control parameter (x_0) is set independently of the stimulation, and corresponds *in vivo* to the GABAergic drugs.

11. Figure 3 B2 what does CE stand for? Cortical excitability or something else? The model and the data have similar properties, what can be done to better fit the model to the data? i.e. B3 vs C3. In Figure 2 the physiological 'control parameter' is time to seizure but then in figure 3 the physiological 'control parameter' is the drug being used, so what is the actual physiological control parameter and how does it relate to the model's control parameter?

R: Yes, CE stands for Cortical excitability which is always the control parameter in all our figures. We however changed it for neural excitability in all our figures to be consistent. The "actual physiological control parameter" is the level of neural excitability set by injection of the GABAergic drugs prior to the stimulations.

12. Figure 6, F, what is actually represented on the x-axis? What is 'network excitability'?

R: As explained in the main text (lines 251-3), 'network excitability' refers to network level responses to single-pulse perturbations, calculated across all the responding electrodes using Non-negative Matrix Factorization. The 'delta network excitability' in figure 6F is simply the differences in the network response to single pulse compared to the control condition (NaCl).

13. For decoding network dynamics, does time of day confound the ability to classify BZD, NaCl and PTZ states? i.e. was BZD, NaCl and PTZ each always delivered at the same time of day?

R: Time of day could indeed be a modulator of the network dynamics. To account for this, all experiments were done around the same time of the day (second half of the light phase). This is specified in the methods section, line 725. Additionally, the different sessions using BZD, NaCl or PTZ were carried out in a randomized order, to avoid the possibility of temporal confounder longer than circadian.

14. "we found that iEEG responses slightly increased up to 10 minutes before the ictal transition" increased in frequency, amplitude, duration or what?

R: All our measures of response to stimulation across the paper are made using the line-length of iEEG the signal on a 250ms window (see Supp. Fig. 2H and methods).

15. Figure 7, thinking of PTZ concentration as the physiological control parameter, is there are way to model the increase in concentration in the tissue as it is being absorbed? and plotting the profile of this modelled concentration in the lead up to the first seizure, for instance in E and F.

R: We thank the reviewer for this suggestion, which we had considered at the time of designing our experiments. However, the instrumentalization needed to measure such concentration directly is heavy, and would have compromised our chronic experiments. The peak in brain PTZ level is reached after 5-10 min after injections (see Yonekawa, Kupferberg, and Woodbury 1980), which corresponds roughly to the latency to seizure we observed. As we don't have precise data to base our modeling on and we feel it would be quite redundant with the increase in excitability modeled in the panel B-C, we would prefer not to add such a plot to our figure.

Reviewer #4 (Remarks to the Author):

“The Critical dynamics of limbic seizures” by Lepeu et al. detail a series of experiments in computer simulations, mice and humans that investigate if epileptic seizures can be predicted in a using passive observations or active probing. I enjoyed reading the paper and in my view, the noteworthy results are some level of consistency across computer simulations, mice and humans, particularly in the active probing paradigm. These sets of results contribute towards a growing literature on markers of increased excitability, proximity to bifurcation points, or impending transitions to seizures. The results presented largely agree with existing work [Meisel 2015, Chang 2018, Maturana 2020, Graham 2023 just to name a few], therefore providing further support to multiple strands of ongoing investigations.

The results also offer some potentially novel insights, particularly the comparison of passive signal markers vs active probing. It appears that active probing is superior or “more sensitive than passive signature” in humans. However, these results are also somewhat in disagreement with Meisel 2015 (PNAS) showing that passive markers of network synchronisation are at least correlated with active probing results with an $R^2 > 0.5$. The authors explain that “The brain operates over multiple spatial and temporal scales and statistical properties may differ at different scales not studied here”, which may serve as an explanation. However, as an additional analysis, I suggest the authors directly compare their passive and active signatures to enable comparison between their models, as well as to previous literature.

R: We thank the Reviewer for this comment. As suggested, we looked at the correlation between all the different metrics (see Supp. Fig. 9B). For each single pulse, we computed the different passive metrics on the 4-8s window after the single pulse. We found a significant but rather weak correlation between active and passive metrics, strengthening the idea that active probing adds important additional information. Interestingly, the line-length computed in the absence of stimulation was the passive metric that correlated the most with active probing.

The authors further highlight another potential novelty in the nature of the bifurcation (fold) in limbic seizures. Please can the authors clarify if these results are limited to the computer model and mouse model (and only partially verified in one human patient)? If so, the conclusion/discussion paragraph should highlight that we cannot draw this conclusion on humans yet. Indeed, multiple papers in the past have hypothesised that a variety of bifurcation types might underpin limbic seizures in humans [Saggio 2020].

R: Yes these results are limited to our animal model of provoked seizure in the hippocampus and we cannot indeed draw any conclusions on humans based on our data. Such limitation was already expressed in the dedicated paragraph (lines 384-7), we now added a direct reference to bifurcation characterization.

Overall, the level of evidence for human epilepsy is comparatively low and bordering on anecdotal. Nevertheless, I think the authors should be applauded for their effort in attempting to establish this link. Some more nuanced discussion on the translational perspective would better highlight this effort and do it justice in terms of extrapolating from the animal and computer model.

R: We agree with the Reviewers that our human data, especially regarding seizure inductions are limited. Such limitations are expressed in the limitation paragraph (lines 384-7).

To assess the statistical validity of this work, more details are needed in the description, and in the code, which should be made available for review.

R: We compile the code in a GitHub folder accessible here:
https://github.com/GregLepeu/Critical_dyn_Hpp_Sz

You will find all the simulation codes as well as the main functions used for the analysis of the animal and human data.

For example, in Fig 5 D, I would either expect to see 16, or 7 data points in each condition, but the figure appears to show 12? I would strongly suggest more transparency in this regard in the next revision.

R: In addition to n reported directly in the figures, we now added the sample size explicitly for all results presented in the text. The figure 5D shows 16 data points in each condition. Some points overlap, but if the Reviewer counts the lines, they will obtain 16 (12 unbroken lines and 4 dashed lines), as written in the legend.

It also appears that multilevel statistics might be appropriate in some instances, e.g. to account for multiple observations from individuals or conditions.

R: We thank the Reviewer for this suggestion. As several observations are made on the same mice, multilevel statistics are indeed a good addition. As the plot in the figure 6F summarizes our findings both regarding the resilience and excitability, we used a linear mixed effect model to redo the linear regression between time-to-seizure and the network excitability (fixed effect) taking this time into account the different animals (random effects, allowing for both random slopes and random intercepts).

These results are now displayed in the Supplementary Figure 9A. As can be seen in this plot, the relationship between the resilience and the excitability is quite similar across animals (colors dots and thin lines) and the regression based on the linear mixed effect model (thick black line) is really similar to one obtained with simple linear regression (thick red line).

Bibliography:

- Ang, Chyze Whee, Gregory C. Carlson, and Douglas A. Coulter. 2006. 'Massive and Specific Dysregulation of Direct Cortical Input to the Hippocampus in Temporal Lobe Epilepsy.' *The Journal of Neuroscience : The Official Journal of the Society for Neuroscience* 26(46):11850–56. doi: 10.1523/JNEUROSCI.2354-06.2006.
- Cela, Elvis, Amanda R. McFarlan, Andrew J. Chung, Taiji Wang, Sabrina Chierzi, Keith K. Murai, and P. Jesper Sjöström. 2019. 'An Optogenetic Kindling Model of Neocortical Epilepsy'. *Scientific Reports* 9(1):5236. doi: 10.1038/s41598-019-41533-2.
- Chang, Michael, Joshua A. Dian, Suzie Dufour, Lihua Wang, Homeira Moradi Chameh, Meera Ramani, Liang Zhang, Peter L. Carlen, Thilo Womelsdorf, and Taufik A. Valiante. 2018. 'Brief Activation of GABAergic Interneurons Initiates the Transition to Ictal Events through Post-Inhibitory Rebound Excitation'. *Neurobiology of Disease* 109:102–16. doi: 10.1016/j.nbd.2017.10.007.
- Chang, Wei Chih, Jan Kudlacek, Jaroslav Hlinka, Jan Chvojka, Michal Hadrava, Vojtech Kumpost, Andrew D. Powell, Radek Janca, Matias I. Maturana, Philippa J. Karoly, Dean R. Freestone, Mark J. Cook, Milan Palus, Jakub Otahal, John G. R. Jefferys, and Premysl Jiruska. 2018. 'Loss of Neuronal Network Resilience Precedes Seizures and Determines the Ictogenic Nature of Interictal Synaptic Perturbations'. *Nature Neuroscience* 21(12):1742–52. doi: 10.1038/s41593-018-0278-y.
- Cuello Oderiz, Carolina, Nicolás Von Ellenrieder, François Dubeau, Ariella Eisenberg, Jean Gotman, Jeffery Hall, Ana Sofía Hincapié, Dominique Hoffmann, Anne Sophie Job, Hui Ming Khoo, Lorella Minotti, André Olivier, Phillippe Kahane, and Birgit Frauscher. 2019. 'Association of Cortical Stimulation-Induced Seizure with Surgical Outcome in Patients with Focal Drug-Resistant Epilepsy'. *JAMA Neurology* 76(9):1070–78. doi: 10.1001/jamaneurol.2019.1464.
- Franklin, Keith B. J., and George Paxinos. 2008. *The Mouse Brain in Stereotaxic Coordinates*. 3rd Edition. Amsterdam: Academic Press, an imprint of Elsevier.
- Gunaydin, Lisa A., Ofer Yizhar, André Berndt, Vikaas S. Sohal, Karl Deisseroth, and Peter Hegemann. 2010. 'Ultrafast Optogenetic Control'. *Nature Neuroscience* 13(3):387–92. doi: 10.1038/nn.2495.
- Holling, C. S. 1973. 'Resilience and Stability of Ecological Systems'. *Annu.Rev.Ecol.Syst.* 4:1–23.
- El Houssaini, Kenza, Anton I. Ivanov, Christophe Bernard, and Viktor K. Jirsa. 2015. 'Seizures, Refractory Status Epilepticus, and Depolarization Block as Endogenous Brain Activities'. *Physical Review E - Statistical, Nonlinear, and Soft Matter Physics* 91(1):2–6. doi: 10.1103/PhysRevE.91.010701.
- Izhikevich, Eugene M. 2006. *Dynamical Systems in Neuroscience: The Geometry of Excitability and Bursting*. The MIT Press.
- Jirsa, Viktor K., William C. Stacey, Pascale P. Quilichini, Anton I. Ivanov, and Christophe Bernard. 2014. 'On the Nature of Seizure Dynamics'. *Brain* 137(8):2210–30. doi: 10.1093/brain/awu133.
- Lu, Yi, Cheng Zhong, Lulu Wang, Pengfei Wei, Wei He, Kang Huang, Yi Zhang, Yang Zhan, Guoping Feng, and Liping Wang. 2016. 'Optogenetic Dissection of Ictal Propagation in the Hippocampal-Entorhinal Cortex Structures.' *Nature Communications* 7:10962.
- Maturana, Matias I., Christian Meisel, Katrina Dell, Philippa J. Karoly, Wendyl D'Souza, David B. Grayden, Anthony N. Burkitt, Premysl Jiruska, Jan Kudlacek, Jaroslav Hlinka, Mark J. Cook, Levin Kuhlmann, and Dean R. Freestone. 2020. 'Critical Slowing down as a Biomarker for Seizure Susceptibility'. *Nature Communications* 11(1):2172. doi: 10.1038/s41467-020-15908-3.

- McCormick, D. A., and D. Contreras. 2001. 'On the Cellular and Network Bases of Epileptic Seizures'. *Annual Review of Physiology* 63:815–46. doi: 10.1146/annurev.physiol.63.1.815.
- Meisel, Christian, and Christian Kuehn. 2012. 'Scaling Effects and Spatio-Temporal Multilevel Dynamics in Epileptic Seizures'. *PLoS ONE* 7(2). doi: 10.1371/journal.pone.0030371.
- Proix, Timothée, Fabrice Bartolomei, Patrick Chauvel, Christophe Bernard, and Viktor K. Jirsa. 2014. 'Permittivity Coupling across Brain Regions Determines Seizure Recruitment in Partial Epilepsy'. *Journal of Neuroscience* 34(45):15009–21. doi: 10.1523/JNEUROSCI.1570-14.2014.
- Proix, Timothée, Fabrice Bartolomei, Maxime Guye, and Viktor K. Jirsa. 2017. 'Individual Brain Structure and Modelling Predict Seizure Propagation'. *Brain* 140(3):641–54. doi: 10.1093/brain/awx004.
- Proix, Timothée, Viktor K. Jirsa, Fabrice Bartolomei, Maxime Guye, and Wilson Truccolo. 2018. 'Predicting the Spatiotemporal Diversity of Seizure Propagation and Termination in Human Focal Epilepsy'. *Nature Communications* 9(1):1088. doi: 10.1038/s41467-018-02973-y.
- Saggio, Maria Luisa, Dakota Crisp, Jared Scott, Phillippa J. Karoly, Levin Kuhlmann, Mitsuyoshi Nakatani, Tomohiko Murai, Matthias Dümpelmann, Andreas Schulze-Bonhage, Akio Ikeda, Mark Cook, Stephen V Gliske, Jack Lin, Christophe Bernard, Viktor Jirsa, and William Stacey. 2020. 'A Taxonomy of Seizure Dynamotypes'. *Elife* 1–56. doi: <https://doi.org/10.7554/eLife.55632>.
- Scheffer, Marten. 2009. *Critical Transitions in Nature and Society*. Princeton University Press.
- Scheffer, Marten, Jordi Bascompte, William A. Brock, Victor Brovkin, Stephen R. Carpenter, Vasilis Dakos, Hermann Held, Egbert H. Van Nes, Max Rietkerk, and George Sugihara. 2009. 'Early-Warning Signals for Critical Transitions'. *Nature* 461(7260):53–59.
- Scheffer, Marten, Stephen R. Carpenter, Timothy M. Lenton, Jordi Bascompte, William Brock, Vasilis Dakos, Johan Van De Koppel, Ingrid A. Van De Leemput, Simon A. Levin, Egbert H. Van Nes, Mercedes Pascual, and John Vandermeer. 2012. 'Anticipating Critical Transitions'. *Science* 338(6105):344–48. doi: 10.1126/science.1225244.
- Sessolo, Michele, Iacopo Marcon, Serena Bovetti, Gabriele Losi, Mario Cammarota, Gian Michele Ratto, Tommaso Fellin, and Giorgio Carmignoto. 2015. 'Parvalbumin-Positive Inhibitory Interneurons Oppose Propagation but Favor Generation of Focal Epileptiform Activity'. *Journal of Neuroscience* 35(26):9544–57. doi: 10.1523/JNEUROSCI.5117-14.2015.
- Shiri, Zahra, Frédéric Manseau, Maxime Lévesque, Williams Sylvain, and Massimo Avoli. 2014. 'Interneuron Activity Leads to Initiation of Low-Voltage Fast-Onset Seizures'. *American Neurological Association* 1–6.
- Subasi, Abdulhamit. 2019. *Practical Guide for Biomedical Signals Analysis Using Machine Learning Techniques*. Elsevier.
- Wozny, C., S. Gabriel, K. Jandova, K. Schulze, U. Heinemann, and J. Behr. 2005. 'Entorhinal Cortex Entraines Epileptiform Activity in CA1 in Pilocarpine-Treated Rats'. *Neurobiology of Disease* 19(3):451–60. doi: 10.1016/j.nbd.2005.01.016.
- Yekhhlef, Latefa, Gian Luca Breschi, Laura Lagostena, Giovanni Russo, and Stefano Taverna. 2015. 'Selective Activation of Parvalbumin- or Somatostatin-Expressing Interneurons Triggers Epileptic Seizurelike Activity in Mouse Medial Entorhinal Cortex.' *Journal of Neurophysiology* 113(5):1616–30. doi: 10.1152/jn.00841.2014.-GABAergic.

Yonekawa, Wayne D., H. J. Kupferberg, and Dixon M. Woodbury. 1980. 'Relationship between Pentylenetetrazol-Induced Seizures and Brain Pentylenetetrazol Levels in Mice'. *Journal of Pharmacology and Experimental Therapeutics* 214(3):589–93.

REVIEWER COMMENTS

Reviewer #1 (Remarks to the Author):

The authors have made some modifications to the manuscript, and have improved it, but it remains a difficult paper to read (and therefore to review). Part of the problem is that they do not always follow convention here (see my comments about the abstract and SI units).

The best element is the discussion, which does a decent job at summarizing the findings, but the other sections remain rather messy. Given that the discussion does present a reasonably clear synopsis, I was disappointed that the authors pushed back on my comments about the abstract, which is almost devoid of substance. I complained previously about the ridiculous opening statement, that mentions volcanoes and markets, neither of which are referred to again anywhere else in the study (because the study is not about these things). Much of the rest of the abstract is empty padding; the only actual results mentioned of the paper are compressed into a single sentence of the abstract, starting with the words “the boundary is ... quantifiable” and constituting about 15 words (<10% of the abstract). And in that sentence, the fact that hippocampal excitability is modulated by GABA is hardly news. Anyone reading the abstract would be very hard put to work out what is novel in this paper. At the very least, I would request that the abstract be rewritten so that it actually conveys some of the substance of the paper.

The most interesting result concerns how ‘active probing’ may help understand seizure transitions, even in the clinic. Unfortunately, the writing was at times ambiguous or confusing – for instance, the section headed “Passive signatures of neural excitability” starts by saying “Next we asked whether active probing...”, and then didn’t cover active probing at all in this section. Also, the Methods descriptions relating to the analyses should be improved. For starters, if people wanted to try to replicate elements of this study, it would be helpful to get some indication of what prior work informed the analyses. I struggled to understand the statistical analyses relating to the section entitled “Decoding network dynamics”. I could not work out how the jargon term ‘empirical chance accuracy’ was derived, and the descriptions of the statistical comparisons was lacking in this section. Also in this section, in what sense are the pharmacological conditions “balanced”?

On a related note, the authors seem peculiarly resistant to drawing parallels with two previous papers that show, to my eyes, virtually the same transition shown here in Figure 7. I don’t know if the authors feel that by doing so, it undermines the impact of their own work. I would disagree, because this seems to me to be a really important observation, that has now been replicated in at least three groups independently – by Klorig, Graham and now this study. The similarity of the results presented in Fig 7 with these prior papers should be made much more explicitly (that is, within the results section and not rather cryptically in the discussion) – please do so in the results

section where it is reported. I did ask for this previously and I do not feel this has been flagged sufficiently clearly. The authors do refer to these other two studies but rather elliptically and only in the discussion. Many readers will not make this connection, and these papers should be linked far more explicitly. It is only by doing so that the field starts to take note of these important observations.

Some of the figures were overly complex with too many panels, so that the key point is not always obvious. The figure legends should perhaps flag what the take home message should be. At times there are notable features in the figures that are largely ignored. For instance, in figure 3, the in silico, mouse and human data look qualitatively rather different, but that is not what the authors focus upon. In fact, they make a very trivial point, that the direction of change is consistent when they added GABA antagonists or blockers. But one is left wondering about other aspects of that figure – why for instance have they shown three barely different simulations in 3B – is it not possible to get the epileptor model to show a larger range of line-length modulation? Are the differences between C and D related to the apparent difference in the duration over which line-length was calculated. The color choices of brown and grey are poor, and at the scale that most of the figures are presented, it is very difficult to distinguish the different colors of the dots, particularly in the insets.

The figures are not helped by poor labelling, where conventional SI units ('s' and 'm') are used to mean multiple different things. For instance, in Figure 4, 's' sometimes means 'sessions', and sometimes means 'signals' and also sometimes means 'seconds' (as it should), while elsewhere in the figure, 'seconds' are referred to as 'secs'. The SI units were introduced for a reason – please stick to these, and do not use reserved terms to mean other things. The misuse of conventional SI units occurs in multiple places throughout the text (e.g. 'N = 8m' – please just write 'mice').

There are fewer syntactical errors, although a good number remain.

Line 69: 'At such tipping or critical point' – this should be 'points'

Line 187 – vice versa – should this be italicized, like all other Latinate phrases?

Line 242: 'dynamical signatures' appears to be both the subject and object of the sentence.

Line 248: 'receiving a BZD' – add 'injection', or omit the 'a'

Line 633: 'bur holes' should be 'burr holes'

In the methods there are repeated references to 'mice or participants'. Please say these are 'human' participants (or my preference would be 'human patients').

Line 799: 'condition' – should this be plural?

Line 873 – TN is not listed. Also, the equation could be simplified to

$$\text{Accuracy} = \text{TP} + \text{TN}$$

if these are proportions (which is how they are reported in the results), or $(\text{TP} + \text{TN})/100$ if these are percentages. This however is not how such analyses are usually done; the more conventional approach is to present the ‘sensitivity’ and ‘specificity’, which distinguishes whether the analyses do better in terms of true positives or true negatives.

Reviewer #2 (Remarks to the Author):

The authors have made several changes to the article, which on a revised reading removes most of my concerns. There is one response that I felt was technically OK, but really could have been done better. And one other minor comment. Otherwise, I find this article very interesting. Being able to manipulate and test seizure threshold in rodents and humans, tied to modeling, is a major achievement.

My question about provoked seizures being similar to spontaneous seizures:

The authors provide the following: additional panel to Fig 1, a description that they have “all the classical EEG hallmarks of seizures”, some subjective descriptions, and Supp Fig. 4. That supp fig shows a rudimentary spectral analysis with a pretty wide stdev. To be fair, this type of cursory spectral analysis is what we typically see in publications, and it technically does answer the question. But to be honest it’s a pretty weak case. Given the high level of quantitative expertise in this group, this was disappointing. Other options could be to analyze the dynamical features (e.g. fitting bifurcation parameters), to compare several standard signal features with a classifier to see if they are similar, etc. What we have are one human and one rat subjective example, which is probably cherry picked, and the only summary is Supp Fig 4 D,E, which have a stdev spanning 2 orders of magnitude in a log plot. Not a strong case. Note that the descriptions of semiology are not good evidence either—that really only proves the seizures are in the same location and spread the same way—not that they have the same dynamical character. I still don’t really know if the provoked and unprovoked seizures are REALLY the same.

Supplement, L 180-184. The sentence about ChETA does not point to any figure, then the next sentence about irregular stimulations points to C4, F. Does that mean the irregular stim are with the

ChETA? I think not. Where are the CHETA data? The main text (L 174) says Suppl fig 2G-I, but that does not seem right.

Reviewer #3 (Remarks to the Author):

The reviewers have addressed most of my concerns other than this one " In the caption you say validate the epileptor prediction but hasn't epileptor been designed to produce/mimic this data? Normally a model would be fit to some data that it could explain and then it gets applied to something new where if it explains something new, that can be considered a prediction." so I'll expand on it.

Although epileptor phenomenologically describes normal and epileptic EEG behaviour as done in previous studies and it can be shown to produce similar behaviours to the data shown here, the model has not actually been fit to your data so you are just showing a model with similar waveforms to the data but not actually providing a model of the data. For instance in the simulations it looks like you place the model at a fixed point driven with out noise and inject a perturbation current to get the different responses. A fixed point for a set of parameters not derived from the data but probably chosen to produce similar waveforms to the data. How do we know the data are operating near a fixed point? More specifically the fixed point you are using in the model? These limitations need to be discussed.

Reviewer #4 (Remarks to the Author):

I thank the authors for addressing my comments and performing additional analyses.

Overall, I enjoyed reading the rebuttal and I am satisfied with most of the response to my comments and believe that this work is sound in methodology and conclusions.

I am, however, still unclear regarding the novelty/overall message of the results presented. In my eyes, the key novelty rests on one result that is currently in supplementary and actually contradicts previous findings.

This is one of the first publications to directly compare active probing vs passive markers of epileptic seizures and I see this as a key novelty. Passive markers alone "based on a sound

understanding of their dynamics” alone is a very old concept and repeated attempts at verifying such an understanding has resulted in mixed conclusions.

I suggest the authors

1. Clarify in their abstract what exactly their key novelties/advances are. For example “we demonstrate that the boundary between physiological hippocampal excitability and seizures is quantifiable” is not necessarily new; “modulated by GABAergic inhibition” is to be expected; “and can be inferred from dynamical signatures detectable in brain recordings” has also been claimed by various paper in animal models and humans.
2. Possibly restructure their results according to each point of novelty/advance. The current changes still tell a very long-winded story about what was done in what order and less about the key novel questions that need to be answered, and what the answers are.
3. Dedicate some space in Discussion on active probing vs passive markers, particularly as their results in Supp Fig 9B are **contradicting** the only other study I am aware of that compared active vs. passive markers (Meisel 2015 PNAS) in vivo – they found a good agreement between active and passive markers!

I also suggest the authors make their code more readable and organise it in a similar way as their simulation code, such that figures can be directly reproduced with obvious instructions. Currently, none of the underlying data is public either, meaning that apart from reading through the dense code, no one can verify the analysis. Some processed data that can be inspected should be publishable without concerns for privacy and identifiability. This is the minimum for publication in my opinion.

Personally, I would still support publication, but only with a clear new structure highlighting novelty/advance, discussing conflicting results with previous work explicitly, and a commitment to transparent data and code sharing.

2nd round of review for Lepeu et al

Reviewer #1 (Remarks to the Author):

The authors have made some modifications to the manuscript, and have improved it, but it remains a difficult paper to read (and therefore to review). Part of the problem is that they do not always follow convention here (see my comments about the abstract and SI units).

Reply: We believe that part of the difficulty and the strength of our study is precisely that it builds bridges between theoretical, computational, experimental, and clinical neuroscience. This made the writing particularly challenging and enjoyable. We opted to always give the theoretical context and the intuition first, to then provide an experimental and clinical illustration, followed by a quantification with statistical testing. This explains that the structure of our paper departs somewhat from purely experimental papers in neuroscience. Our structure is however well in line with the series of papers that have dealt with a dynamical systems approach to epilepsy, such as Jirsa et al., *Brain*, 2014, Chang et al., *Nature Neuroscience*, 2018, and Maturana et al., *Nature Communications* 2020. All of these papers introduce the theory first and then illustrate the concepts with experiments, before providing a final quantification with statistical testing. We believe that the systematic structure we have implemented in the text adds clarity, which we view as a strength of the paper. We kept the overall structure of the paper, taking into account the feedback from Reviewers with presumably different backgrounds, but also addressed the Reviewer's concerns as detailed below.

The best element is the discussion, which does a decent job at summarising the findings, but the other sections remain rather messy. Given that the discussion does present a reasonably clear synopsis, I was disappointed that the authors pushed back on my comments about the abstract, which is almost devoid of substance. I complained previously about the ridiculous opening statement, that mentions volcanoes and markets, neither of which are referred to again anywhere else in the study (because the study is not about these things). Much of the rest of the abstract is empty padding; the only actual results mentioned of the paper are compressed into a single sentence of the abstract, starting with the words "the boundary is ... quantifiable" and constituting about 15 words (<10% of the abstract). And in that sentence, the fact that hippocampal excitability is modulated by GABA is hardly news. Anyone reading the abstract would be very hard put to work out what is novel in this paper. At the very least, I would request that the abstract be rewritten so that it actually conveys some of the substance of the paper.

Reply: We would like to first highlight the limitations imposed by the journal on the abstract, which serves a slightly different purpose than in most journals, where longer abstracts are accepted. From Nature communications guidelines: "The abstract — which should be no more than 200 words long and contain no references — should serve both as a general introduction to the topic and as a brief, non-

technical summary of the main results and their implications.” However, we have now rewrote parts of the abstract based on feedback from Reviewers 1 & 4. We removed the opening statement and emphasized specific and novel findings, including the fact that active probing surpasses passive recordings to decode underlying excitability.

Original Abstract	Large and complex systems sometimes undergo abrupt and potentially devastating regime shifts: volcanos erupt, markets crash, and brains seize. Such critical transitions occur upon small perturbations in highly interconnected systems presenting positive feedback loops and can be modeled as mathematical bifurcations between alternative regimes. The predictability of critical transitions represents a major challenge for modern science, but dynamical systems theory predicts the appearance of subtle dynamical signatures on the verge of instability. Whether such dynamical signatures can be measured before impending seizures remains uncertain. Using a combination of mathematical modelling, optogenetics in mice and intracranial electrical stimulations in patients with epilepsy, we demonstrate that the boundary between physiological hippocampal excitability and seizures is quantifiable, modulated by GABAergic inhibition, and can be inferred from dynamical signatures detectable in brain recordings. Our findings provide a promising approach for predicting and preventing seizures, based on a sound understanding of their dynamics.
Revised Abstract	Epilepsy is defined by the abrupt emergence of harmful seizures, but the nature of these regime shifts remains enigmatic. From the perspective of dynamical systems theory, such critical transitions occur upon inconspicuous perturbations in highly interconnected systems and can be modeled as mathematical bifurcations between alternative regimes. The predictability of critical transitions represents a major challenge, but the theory predicts the appearance of subtle dynamical signatures on the verge of instability. Whether such dynamical signatures can be measured before impending seizures remains uncertain. Here, we verified that predictions on bifurcations applied to the onset of hippocampal seizures, providing concordant results from in silico modeling, optogenetics experiments in mice and intracranial EEG recordings in human patients with epilepsy. Using pharmacological control over neural excitability, we showed that the boundary between physiological excitability and seizures can be inferred from dynamical signatures passively recorded or actively probed in hippocampal circuits. Of importance for the design of future neurotechnologies, active probing surpassed passive recording to decode underlying levels of neural excitability, notably when assessed from a network of propagating neural responses. Our findings provide a promising approach for predicting and preventing seizures, based on a sound understanding of their dynamics.

The most interesting result concerns how ‘active probing’ may help understand seizure transitions, even in the clinic. Unfortunately, the writing was at times ambiguous or confusing – for instance, the section headed “Passive signatures of neural excitability” starts by saying “Next we asked whether active probing...”, and then didn’t cover active probing at all in this section.

Reply: we agree with the Reviewer that phrasing here could have been misleading and have now rewritten this transition:

Original	Next, we asked whether active probing was necessary to uncover underlying excitability.
--

Revised	We further found that changes in the system's recovery rates were also reflected in simple passive statistics of longitudinal iEEG recordings including variance, skewness, line-length and auto-correlation of the signal, in line with prior publications.
----------------	--

Also, the Methods descriptions relating to the analyses should be improved. For starters, if people wanted to try to replicate elements of this study, it would be helpful to get some indication of what prior work informed the analyses.

Reply: In addition to the 23 references already cited in the result section, we now added more direct phrasing to refer to methods already used by others. For example:

- 'Like others (REFs), we probed resilience to ictal transitions using the time-to-seizure, that is the number of pulses or total duration of stimulation necessary to force an ictal transition, (Fig. 1E).'
- 'We further found that the system's recovery rates were also reflected in simple passive statistics of longitudinal iEEG recordings including variance, skewness, line-length and auto-correlation of the signal, expanding on findings in experiments in vitro (REFs)'
- 'To capture iEEG responses shared in a subnetwork connected to an electrode undergoing single-pulse probing over a range of intensities, we used Non-negative Matrix Factorization (NMF, Fig. 5C-D, Supp. Fig. 8)(REFs).'

I struggled to understand the statistical analyses relating to the section entitled "Decoding network dynamics". I could not work out how the jargon term 'empirical chance accuracy' was derived, and the descriptions of the statistical comparisons was lacking in this section.

Reply: we have now clearly defined the chance-level in the result section, in addition to the previously provided explanation in the methods.

Original	We trained a multilabel logistic regression model to classify states of low, normal, and high excitability corresponding to the three balanced pharmacological conditions (BZD, NaCl and PTZ, empirical chance accuracy ~0.33)
Revised	We trained a multilabel logistic regression model to classify states of low, normal, and high excitability corresponding to the three balanced pharmacological conditions (one third each: BZD vs. NaCl vs. PTZ, chance-level accuracy ~0.33, estimated for each mouse by training on shuffled labels)

Also in this section, in what sense are the pharmacological conditions "balanced"?

Reply: in classification problems, the balance between the number of occurrences of an element in a given class is of utmost importance in determining chance-level. For example, if the classes are unbalanced with 99% control condition, it would be very easy to be right 99% of the time by claiming that every single trial comes from the control condition. The fact that we had 1/3 of observations in each pharmacological category ensured that the classification problem was balanced in our case. We have now clarified this point in the result section as shown above.

On a related note, the authors seem peculiarly resistant to drawing parallels with two previous papers that show, to my eyes, virtually the same transition shown here in

Figure 7. I don't know if the authors feel that by doing so, it undermines the impact of their own work. I would disagree, because this seems to me to be a really important observation, that has now been replicated in at least three groups independently – by Klorig, Graham and now this study. The similarity of the results presented in Fig 7 with these prior papers should be made much more explicitly (that is, within the results section and not rather cryptically in the discussion) – please do so in the results section where it is reported. I did ask for this previously and I do not feel this has been flagged sufficiently clearly. The authors do refer to these other two studies but rather elliptically and only in the discussion. Many readers will not make this connection, and these papers should be linked far more explicitly. It is only by doing so that the field starts to take note of these important observations.

Reply: We fully agree with the Reviewer that highlighting commonalities between studies is important. We now also refer to these two studies in the results, at the Reviewer request. We now open the paragraph with: *'In a final mouse experiment, we showed how active probing could anticipate a PTZ-induced ictal transition, as done by Graham et al. for 4-aminopyridine-induced seizures'* (lines 307-308). We draw a direct parallel on the finding, as follows: *'In the last four minutes before the ictal transition though, single-pulses now triggered large epileptiform responses across the network (+54% [+36,+70], Fig. 7H3), mimicking spontaneous epileptic spikes that also appeared in the recording (Fig. 7H4, Supp. Fig. 10B6) and reminiscent of results in Graham et al'* (lines 323-326). It is important to note that there was no probing of an ictal transition in Klorig et al., as done by us and Graham et al. (Klorig et al. used subconvulsive doses of PTZ). We now added a sentence in the results to highlight the difference between Klorig's and our measurements: *"As opposed to the method developed by Klorig et al., we did not rely on the response probability to estimate excitability, which drastically reduced the number of probing pulses needed"* (lines 224-6). We will defer to the Editor on the final decision of including this type of comparison to other studies directly in the result section. Regarding the discussion, we added one sentence to explain Graham's findings in even more detail and make the link to our findings even clearer (lines 370-9).

Before	'Such pre-ictal 'step-change' in recovery was also recorded by others in vitro in brain slices exposed to penicillin ⁵¹ , high potassium ¹³ , low magnesium ²⁹ or 4-aminopyridine ²⁹ as well as in vivo , upon cortical injection of 4-aminopyridine ²⁹ . In the latter experiment, Graham et al. observed pre-ictal prolonged plateau potentials in dendrites, reflecting a calcium entry upon optogenetic probing ²⁹ . Thus, different seizure-inducing mechanisms share the same pre-ictal dynamical signatures - critical slowing - a possibly universal phenomenon ^{11,12,17-19} , that may relate to specific neuronal (or dendritic) mechanisms in epilepsy.'
Revised	'Such pre-ictal 'step-change' in recovery was also recorded by others in vitro in brain slices exposed to penicillin ⁵¹ , high potassium ¹³ , low magnesium ²⁹ or 4-aminopyridine ²⁹ as well as in vivo , upon cortical injection of 4-aminopyridine ²⁹ . In the minutes preceding the onset of seizures in slices and in vivo , Graham et al. observed the appearance of prolonged potentials in dendrites, reflecting increased

	calcium entry upon optogenetic probing ²⁹ . These dendritic ‘plateau potentials’ were associated with increased neuronal firing rates and prolonged recovery in the evoked cortical response ²⁹ . Thus, different modes of inducing seizures share the same pre-ictal dynamical signatures - critical slowing - a possibly universal phenomenon ^{11,12,17-19} , that may relate to specific neuronal (or dendritic) mechanisms in epilepsy.
--	--

Some of the figures were overly complex with too many panels, so that the key point is not always obvious. The figure legends should perhaps flag what the take home message should be.

Reply: We now highlight in the figure legend what the key novelty is. To that aim, we added one sentence after the figure’s title in each figure legend. To improve the figure 2 clarity, we removed the quantification insets from inside the panel.

At times there are notable features in the figures that are largely ignored. For instance, in figure 3, the in silico, mouse and human data look qualitatively rather different, but that is not what the authors focus upon. In fact, they make a very trivial point, that the direction of change is consistent when they added GABA antagonists or blockers. But one is left wondering about other aspects of that figure – why for instance have they shown three barely different simulations in 3B – is it not possible to get the epileptor model to show a larger range of line-length modulation?

Reply: Depending on the exact parameters, these curves can have more marked difference. We now provide an updated version of the in silico simulations (figure 3B2 in the main text) where we used additional stimulation values, as well as an example with added noise in the supplementary data, which show more marked differences across the three conditions. We now observe a clear jump in line-length at different stimulation values in silico, which is similar to the jumps observed in mice (C2) and humans (D2). The main difference between those three subplots is that the differences in line-length across conditions remain marked for higher stimulation values in mice, while it disappeared for both humans and in silico. One possible interpretation is that the intensity of optogenetic stimuli is inherently restricted by the numbers of neurons transfected with the opsin. Consequently, the stimulation never reaches the “saturation point” where the cortical response becomes the same regardless of the underlying excitability. We modified the describing text in the Result section to highlight these differences: *“The area under this curve (thereafter simply IOC) captures in one value (from zero: no response at any intensity, to one: maximal response at minimal intensity) a given excitability level, irrespective of changes in slope (e.g. Fig. 3D2) or height (e.g. Fig. 3C2)”* lines (214-7).

Are the differences between C and D related to the apparent difference in the duration over which line-length was calculated.

Reply: No, line-lengths were calculated in a 250ms window across all modalities (in silico, mice and human experiment). We now plotted all three examples of response to single pulse in a similar time window to avoid any confusion.

The color choices of brown and grey are poor, and at the scale that most of the figures are presented, it is very difficult to distinguish the different colors of the dots, particularly in the insets.

Reply: We have now taken the insets out of the panel and made another panel for better readability.

The figures are not helped by poor labelling, where conventional SI units ('s' and 'm') are used to mean multiple different things. For instance, in Figure 4, 's' sometimes means 'sessions', and sometimes means 'signals' and also sometimes means 'seconds' (as it should), while elsewhere in the figure, 'seconds' are referred to as 'secs'. The SI units were introduced for a reason – please stick to these, and do not use reserved terms to mean other things. The misuse of conventional SI units occurs in multiple places throughout the text (e.g. 'N = 8m' – please just write 'mice').

Reply: We thank the Reviewer for highlighting this important point. We agree that we should have used the SI system, we systematically verified our use of units and avoided using acronyms where not needed.

There are fewer syntactical errors, although a good number remain.

Reply: before each submission, we had our manuscript checked for misspelling and syntactical errors by two independent native English speakers. We thank the Reviewer for spotting additional inconsistencies that they missed.

Line 69: 'At such tipping or critical point' – this should be 'points'

Reply: Yes, we changed to 'points'

Line 187 – vice versa – should this be italicized, like all other Latinate phrases?

Reply: Our dictionary indicates that vice versa is typically not italicized, as it is part of common language. We only italicized latinate phrases that refer to scientific concepts.

Line 242: 'dynamical signatures' appears to be both the subject and object of the sentence.

Reply: indeed, we now removed the second use of the term 'dynamical signatures'.

Line 248: 'receiving a BZD' – add 'injection', or omit the 'a'

Reply: we now added the term 'injection' in 'participants receiving a BZD injection'.

Line 633: 'bur holes' should be 'burr holes'

Reply: we now write 'burr holes'

In the methods there are repeated references to 'mice or participants'. Please say these are 'human' participants (or my preference would be 'human patients').

Reply: we indeed very consistently used 'participants' to refer to human patients. We have now specified in the method section that these are 'human participants' according to the Reviewer's suggestion.

Line 799: 'condition' – should this be plural?

Reply: yes, this should be plural, we now wrote 'conditions'.

Line 873 – TN is not listed.

Reply: we now list TN as 'true negatives'.

Also, the equation could be simplified to

$Accuracy = TP + TN$

if these are proportions (which is how they are reported in the results), or $(TP + TN)/100$ if these are percentages.

Reply: We did not express true and false negatives and positives as proportions. Following the convention, true and false negatives and positives are defined as absolute numbers, not proportions. In our original methods, we specified: 'Where TP, TN, FP and FN are respectively the numbers of true positives, true negatives, false positives and false negatives.' We did not find the report of true positives or negatives in the result section, where only the accuracy is reported. The proportion in the accuracy metric comes from the ratio to all other numbers.

This however is not how such analyses are usually done; the more conventional approach is to present the 'sensitivity' and 'specificity', which distinguishes whether the analyses do better in terms of true positives or true negatives.

Accuracy is a widely used metric, which is both influenced by specificity and sensitivity. We used accuracy as a global assessment, to avoid the complexity of defining specificity and sensitivity for three classes, which would result in a total of six metrics, depending on the comparison at hand. Indeed, we have a three-class problem, and one would have to report the sensitivity and specificity of classifying 'high excitability' versus all other categories ('low' and 'normal' excitability) and so on. We believe the result is more digestible and intuitive in the form of the Accuracy which can directly be interpreted as the percentage of trials that were correctly attributed to their respective category. We now emphasize this latter point on the interpretation in the result section in the following sentence: *'To evaluate and compare the performance of each classifier, we calculated the Accuracy of this three-label classification (see methods), which can be directly interpreted as the*

percentage of single trials that were correctly classified at one of the three levels of cortical excitability' (lines 287-290).

Reviewer #2 (Remarks to the Author):

The authors have made several changes to the article, which on a revised reading removes most of my concerns. There is one response that I felt was technically OK, but really could have been done better. And one other minor comment. Otherwise, I find this article very interesting. Being able to manipulate and test seizure threshold in rodents and humans, tied to modeling, is a major achievement.

Reply: we thank the Reviewer for their very high appreciation of our study.

My question about provoked seizures being similar to spontaneous seizures: The authors provide the following: additional panel to Fig 1, a description that they have “all the classical EEG hallmarks of seizures”, some subjective descriptions, and Supp Fig. 4. That supp fig shows a rudimentary spectral analysis with a pretty wide stdev.

Reply: the Std deviation comes mostly from averaging across animals. We now normalized all PSD in order to decrease the effect of inter-animal variability (Supp Fig 4D, E).

To be fair, this type of cursory spectral analysis is what we typically see in publications, and it technically does answer the question. But to be honest it's a pretty weak case. Given the high level of quantitative expertise in this group, this was disappointing. Other options could be to analyze the dynamical features (e.g. fitting bifurcation parameters), to compare several standard signal features with a classifier to see if they are similar, etc. What we have are one human and one rat subjective example, which is probably cherry picked, and the only summary is Supp Fig 4 D,E, which have a stdev spanning 2 orders of magnitude in a log plot. Not a strong case. Note that the descriptions of semiology are not good evidence either—that really only proves the seizures are in the same location and spread the same way—not that they have the same dynamical character. I still don't really know if the provoked and unprovoked seizures are REALLY the same.

Reply: In addition to the spectral analysis, we now added a discussion on both provoked and unprovoked seizure dynamotypes in the supplementary materials, following the methodology proposed by Saggio et al. (eLife, 2020). After extracting

interspike intervals and spikes magnitudes, we visually reviewed each seizure to define its dynamotype (Supplementary Table 2). As previously explained in Saggio et al. (eLife, 2020), such classification is ambiguous in presence of high-pass filters, which precludes observing DC shifts, and therefore distinguishing between saddle-node and subcritical Hopf bifurcations. We observed both the saddle-node/subcritical Hopf and saddle-node on invariant cycle types of bifurcation, but not the subcritical Hopf type. We did not observe statistically significant differences in the type of onset ($p=0.18$, Chi-square test) and offset bifurcations ($p=0.42$, Chi-square test) between provoked and unprovoked seizures.

Of importance for our study, both saddle-node and saddle-node on invariant cycles are fold bifurcations, and therefore compatible with an integrator type bifurcation. To further rule out the subcritical Hopf bifurcation as an onset type, we experimentally tested the integrator properties of the recorded signal for provoked seizure by showing that no resonant frequency could be found (Sup. Fig 5). Rather perturbations systematically accumulate faster for higher stimulation rates, a behavior that is typical of integrator type of bifurcations.

Supplement, L 180-184. The sentence about ChETA does not point to any figure, then the next sentence about irregular stimulations points to C4, F. Does that mean the irregular stim are with the ChETA? I think not. Where are the CHETA data? The main text (L 174) says Suppl fig 2G-I, but that does not seem right.

Reply: Indeed, there was a mistake in our figure referencing, we thank the Reviewer for spotting it. ChETA ospins were used only in the PV interneurons experiment. Histological results can be found in Supp. Fig. 3G-H and an example trace is shown Supp. 5 D3. We now clarify this both in the main text (lines 180-2) and in the methodology section (lines 685-687).

Reviewer #3 (Remarks to the Author):

The reviewers have addressed most of my concerns other than this one " In the caption you say validate the epileptor prediction but hasn't epileptor been designed to produce/mimic this data? Normally a model would be fit to some data that it could explain and then it gets applied to something new where if it explains something new, that can be considered a prediction." so I'll expand on it.

Although epileptor phenomenologically describes normal and epileptic EEG behaviour as done in previous studies and it can be shown to produce similar behaviours to the data shown here, the model has not actually been fit to your data so you are just showing a model with similar waveforms to the data but not actually providing a model of the data. For instance in the simulations it looks like you place the model at a fixed point driven with out noise and inject a perturbation current to get the different responses. A fixed point for a set of parameters not derived from the data but probably chosen to produce similar waveforms to the data.

Reply: The reviewer is correct that the Epileptor is a phenomenological model which has been built to reproduce several qualitative features of ictal and interictal EEG. These qualitative features are essential because they allow identifying the nature of the bifurcations that drive the dynamics. It is important to identify the nature of the bifurcations because the system, close to a bifurcation, behaves quantitatively according to the (low-order) normal form of the bifurcation, independently of higher-order terms that may appear in the equations fitting the system's behavior. It is also a modeling level which was deliberately chosen because it allows extracting invariants across species and seizures that would likely not be found for models quantitatively fitted to the data. Indeed, it is clear that there are large quantitative differences between seizures of different subjects/animals.

Of interest for our study, the seizure onset bifurcation in the Epileptor model was chosen to be a fold bifurcation in the original study (Jirsa et al., Brain, 2014). In the original study, this choice was validated by a series of in vitro experiments that confirmed this choice, such as the interspike intervals between pre-ictal spikes and the presence of a DC shift at seizure onset. This was later extended to an in vivo human study attempting to identify the seizure onset bifurcations in patients with epilepsy. A large prevalence of fold bifurcations was found (Saggio et al., eLife, 2020).

What has not been shown experimentally is whether electrical stimulation in vivo, which can induce epileptic seizures, can also demonstrate the folded nature of the seizure onset bifurcation. We thus used the Epileptor model to do just that: to specify what qualitative feature we would expect to see by stimulating the system. Some of these features could have been derived directly from the fold normal form equation, such as the integrator nature of the system. But the Epileptor also makes more specific assumptions, especially about the dynamics of the interictal spikes observed after stimulation. These assumptions have qualitative consequences on the dynamics predicted by the Epileptor model. In particular, we did not change the

equations or qualitative behavior of the Epileptor to fit the data. We only changed the time scale of one parameter to facilitate comparison with experimental data and the excitability parameter to reflect our three experimental conditions. Despite the strong priors established by the original Epileptor's equations, the experimental recordings mostly confirmed qualitatively the predicted dynamics obtained by electrical stimulation. In particular, we showed both *in silico* and *in vivo* that sustained stimulations lead to cumulative perturbations that can reach the seizure threshold (Supp. Fig 2 A-C and H-J). We showed that resilience to ictal transition, calculated as either time-to-seizure or cumulative line-length, is directly proportional to the distance to the critical point (Supp. Fig 2 E-G). Finally, we used a series of experiments (frequency effects, irregular stimulations and interneuron stimulations) to confirm that we have integrator rather than resonator dynamics in the mouse hippocampus, similar to those implemented in the Epileptor (Supp. Fig. 5).

How do we know the data are operating near a fixed point? More specifically the fixed point you are using in the model? These limitations need to be discussed.

That the non-epileptic regime operates at a fixed point is indeed an assumption that derives from the modeling choices made in the Epileptor model. Non-epileptic brain dynamics are obviously not settled at a single fixed point, which clearly cannot explain the full range of brain dynamics from resting states to cognition. But we think that it is an approximation that makes sense in terms of non-epileptic versus epileptic dynamics: compared to epileptic dynamics, which are characterized by highly synchronized and high-amplitude oscillations, non-epileptic states are relatively quiet and stable, hence modeled as a fixed point. We could confirm this experimentally, with *in vivo* experiments indicating that the optogenetic stimulations showed qualitative features (integrator dynamics, Sup. Fig. 5) corresponding to the qualitative behaviors of a fixed point located near a fold bifurcation.

Reviewer #4 (Remarks to the Author):

I thank the authors for addressing my comments and performing additional analyses. Overall, I enjoyed reading the rebuttal and I am satisfied with most of the response to my comments and believe that this work is sound in methodology and conclusions.

Reply: we thank the Reviewer for the very positive overall appreciation of our work, and believe we were able to address the remaining concerns as detailed below.

I am, however, still unclear regarding the novelty/overall message of the results presented. In my eyes, the key novelty rests on one result that is currently in supplementary and actually contradicts previous findings. This is one of the first publications to directly compare active probing vs passive markers of epileptic seizures and I see this as a key novelty. Passive markers alone “based on a sound understanding of their dynamics” alone is a very old concept and repeated attempts at verifying such an understanding has resulted in mixed conclusions.

Reply: we have now emphasized the direct comparison between passive markers and active probing in the:

- Abstract (see below)

- Introduction (last sentence, lines 103-104: ‘Our findings highlight the superiority of actively probing the cortex as opposed to relying on passive dynamic signatures to assess underlying levels of cortical excitability.’),
- Results (lines 281-283: ‘Next, we formally assess the superiority of a probing strategy in decoding momentary states of excitability over the poorer predictive value of partially correlated passive dynamical signatures (R^2 ranging from 0.09 to 0.29, Supp. Fig. 9B)’ and lines 302-304 ‘These machine-learning results highlight the decoding value of active probing over that of relying on partially correlated passive dynamical signatures’)
- Discussion (lines 332-340):

Original Discussion	In this study, we verified how fundamental mathematical predictions on fold bifurcations may apply to hippocampal seizures in mice and humans in vivo . By actively probing the hippocampus, we found dynamical signatures of critical slowing that can serve as warnings about imminent ictal transitions. Expanding previous work on passive dynamical signatures ^{13,22-25,48} , our work provides a robust framework to gauge excitability and seizure thresholds, adding mathematical formalism ^{14,19} to terms that are sometimes ambiguous in epileptology. The importance of such measurements is to be found in their ability to reflect the risk of upcoming seizures. The key contributions of this study are further specified below.
Revised Discussion	In this study, we verified how fundamental mathematical predictions on fold bifurcations apply to hippocampal seizures in mice and humans in vivo , powerfully expanding prior experimental evidence. Beyond previous work on passive dynamical signatures ^{13,22-25,48} , our study also provides a robust experimental and clinical framework to actively gauge excitability and seizure thresholds , adding mathematical formalism ^{14,19} to terms that are sometimes ambiguous in epileptology. The importance of such measurements is to be found in their ability to reflect the risk of upcoming seizures. By actively probing the hippocampus, we uncovered dynamical signatures of critical slowing that were ambiguous in passive iEEG recordings but can serve as warnings about imminent ictal transitions. The key contributions of this study are further specified below.

- We added a direction comparison with the previous study by Meisel and colleagues in our discussion (lines 388-390, see below).

I suggest the authors

1. Clarify in their abstract what exactly their key novelties/advances are. For example “we demonstrate that the boundary between physiological hippocampal excitability and seizures is quantifiable” is not necessarily new; “modulated by GABAergic inhibition” is to be expected; “and can be inferred from dynamical signatures detectable in brain recordings” has also been claimed by various paper in animal models and humans.

Reply: we have now rewritten the abstract based on feedback from Reviewer 1 & 4, and include more specificities on our findings.

Original Abstract	Large and complex systems sometimes undergo abrupt and potentially devastating regime shifts: volcanos erupt, markets crash, and brains seize. Such critical transitions occur upon small perturbations in highly interconnected systems presenting positive feedback loops and can be modeled as
---

	mathematical bifurcations between alternative regimes. The predictability of critical transitions represents a major challenge for modern science, but dynamical systems theory predicts the appearance of subtle dynamical signatures on the verge of instability. Whether such dynamical signatures can be measured before impending seizures remains uncertain. Using a combination of mathematical modelling, optogenetics in mice and intracranial electrical stimulations in patients with epilepsy, we demonstrate that the boundary between physiological hippocampal excitability and seizures is quantifiable, modulated by GABAergic inhibition, and can be inferred from dynamical signatures detectable in brain recordings. Our findings provide a promising approach for predicting and preventing seizures, based on a sound understanding of their dynamics.
Revised Abstract	Epilepsy is defined by the abrupt emergence of harmful seizures, but the nature of these regime shifts remains enigmatic. From the perspective of dynamical systems theory, such critical transitions occur upon inconspicuous perturbations in highly interconnected systems and can be modeled as mathematical bifurcations between alternative regimes. The predictability of critical transitions represents a major challenge, but the theory predicts the appearance of subtle dynamical signatures on the verge of instability. Whether such dynamical signatures can be measured before impending seizures remains uncertain. Here, we verified that predictions on bifurcations applied to the onset of hippocampal seizures, providing concordant results from in silico modeling, optogenetics experiments in mice and intracranial EEG recordings in human patients with epilepsy. Using pharmacological control over neural excitability, we showed that the boundary between physiological excitability and seizures can be inferred from dynamical signatures passively recorded or actively probed in hippocampal circuits. Of importance for the design of future neurotechnologies, active probing surpassed passive recording to decode underlying levels of neural excitability, notably when assessed from a network of propagating neural responses. Our findings provide a promising approach for predicting and preventing seizures, based on a sound understanding of their dynamics.

2. Possibly restructure their results according to each point of novelty/advance. The current changes still tell a very long-winded story about what was done in what order and less about the key novel questions that need to be answered, and what the answers are.

Reply: We thank the Reviewer for this suggestion. We have taken the following actions based on the Reviewer's and other's suggestions: 1) we now include a lay out of the result section in the first paragraph of the results (lines 109-121). We believe this will help readers understand the logical flow of the results: "To generate testable hypotheses, we implemented the previously published Epileptor model⁵ (Fig. 1C-D) that reproduces invariant features of seizures *in silico* (Fig. 1E1) and provided detailed predictions on the dynamics of seizure onsets (i.e. at the critical point) beyond those previously tested. First, we verified the nature of the bifurcation *in vivo*, probing resilience and recovery with neurostimulation in non-epileptic freely-moving wildtype mice (N=17, Fig. 1E2) and in patients with epilepsy undergoing a presurgical evaluation via iEEG for clinical reasons (N=10, Fig. 1E3, Supp. Table 1). This allowed us to establish robust, interpretable and translational means of measuring resilience (Fig. 1E) and recovery (Fig. 1F-G) and to verify their dynamical meaning in the model. Second, we showed the correlation between resilience, recovery and passive dynamical signatures in pharmacologically-

controlled conditions of excitability and compared their ability to predict underlying neural excitability. Finally, we verified whether active and passive dynamical signatures heralded ictal transitions in mice. “

2) we now highlight the finding at the beginning of each result paragraph. Instead of highlighting the question at hand, we state the finding, and then justify it with data.

3) We now also highlight the finding at the beginning of each figure legend, after the short figure title.

4) We have contracted some of the result descriptions into simpler sentences. We understand what the Reviewer means with a ‘very long-winded story’ and agree that the order of the results and figures is incremental. Given the complexity of the concepts, and the fact that most readers will not be familiar with either the theoretical or the experimental side, we nevertheless opt to keep this approach. Indeed, we chose Nature communications for our publication, in part because the word limit allowed for a more in-depth gradual explanation of our work, a dissection of the theoretical concepts and a step-by-step explanation of the experiments. We believe that this article will be very didactic to help readers familiarize themselves with the application of dynamical system theory to epilepsy. We see this didactic aspect as a strength of our article.

3. Dedicate some space in Discussion on active probing vs passive markers, particularly as their results in Supp Fig 9B are *contradicting* the only other study I am aware of that compared active vs. passive markers (Meisel 2015 PNAS) in vivo – they found a good agreement between active and passive markers!

Reply: The correlation shown in Figure 1 in Meisel et al., PNAS 2015 is between the magnitude of responses to stimulation-evoked potentials (2 channels) and a synchrony index (across channels) in two human patients over a full day of recording (~24h). The synchrony index is measured for various broad-bands at high-frequency, of 50-100Hz, 100-200Hz and 200-400Hz. Thus, many aspects differ between our and these measurements, and we do not believe that a direct comparison is warranted, nor to conclude that there is a contradiction between the two.

- First, we did not measure the synchrony index between channels between 50 and 400Hz in our EEG traces. Of our five passive signatures, the one measurement that may reflect a synchrony index between brain regions is the spatial correlation (Fig. 5-6). In our case, spatial correlation may be mostly reflecting lower-frequency oscillations, as we filtered more broadly than in the study by Meisel et al.
- Second, we correlated passive signatures and active probing at the single-trial level, by comparing directly the 250ms after probing to 4s recordings between pulses. The weak correlations we found between active probing and passive signatures was always positive, thus in the same direction as found by Meisel et al. based on the comparison between average values. As highlighted in the paper by Meisel et al. responses to probing stimuli can be quite variable from trial to trial. The fact that the correlation was weaker, could in part be explained by the difference between comparing single trials versus comparing

averages in 10 minute segments. This is why our single-trial machine-learning decoding result is remarkable. Despite the variability, active probing shows consistent enough changes in single-trial responses that can be captured by a simple logistic regression.

- Third and most importantly, we did not compare the changes in passive and active markers over a full day of recording (including wake and sleep) but only in awake subjects undergoing controlled pharmacological interventions. As highlighted in the paper by Meisel et al. in their Fig. 1 and 4, the sleep-wake cycle has large influences on measurements of synchrony and excitability. It is thus possible that the strong correlation they found was mostly driven by changes in vigilance stages that were not included in our recordings.

A full investigation of all passive signatures and active probing and the many factors that influence these measurements should be the focus of a follow-up study. We now highlight this point in our discussion, citing Meisel et al., PNAS 2015 with the following statement about passive and active dynamical signatures (lines 388-390): “Fourth, our machine-learning results showed that snap-shots of actively probed signals (250ms) more reliably uncovered underlying levels of excitability with ~80% accuracy, over passive signatures of critical slowing (increased line-length, variance, skewness, autocorrelation and spatial correlation) which were here ambiguous, and have yielded conflicting results in the literature^{13,22–25,48}. Unlike others using signal averaging over longer 10-min recordings in sleep and wake, we found weak correlations between passive and active dynamical signatures at a shorter timescale in the awake brain, a practical result in line with theoretical predictions^{27,28,30,52}. “

I also suggest the authors make their code more readable and organise it in a similar way as their simulation code, such that figures can be directly reproduced with obvious instructions. Currently, none of the underlying data is public either, meaning that apart from reading through the dense code, no one can verify the analysis. Some processed data that can be inspected should be publishable without concerns for privacy and identifiability. This is the minimum for publication in my opinion.

Reply: We agree with the Reviewer that code and data to reproduce the main results should always be made available. We now share the mouse and human data. The code for in-vivo analysis has now been cleaned and restructured. It is now provided in Jupyter notebooks along with the simulation code. Both example and processed data are now provided with the code in order to reproduce all the panels shown in the main figures.

Personally, I would still support publication, but only with a clear new structure highlighting novelty/advance, discussing conflicting results with previous work explicitly, and a commitment to transparent data and code sharing.

Reply: We believe we have addressed these three points above.

REVIEWERS' COMMENTS

Reviewer #2 (Remarks to the Author):

The authors have answered all my questions.

Reviewer #3 (Remarks to the Author):

The authors have adequately addressed my concerns. The combination of the modelling, animal models and human data makes this a high quality paper supporting the ideas underlying simple bifurcation models of epilepsy.

Reviewer #3 (Remarks on code availability):

In the manuscript it says

The code and some sample or processed data necessary for running the code and reproducing the 921 analysis and results presented in the paper are available at: (see figshare DOI).

but I cannot click anything to take me to the code to evaluate it. Ideally someone will do this before acceptance. I am happy to check it.

Reviewer #4 (Remarks to the Author):

I thank the authors for their detailed response to my questions, and corresponding changes in the paper. Reading this revised version, my opinion is that the clarity is greatly improved, and the context of the work is a lot clearer.

I have no further comments, but to congratulate the authors on an important piece of work that I believe will be appreciated by many in the community.

Reviewer #4 (Remarks on code availability):

I have also reviewed the code base briefly to ensure that the underlying data is provided, and the code is usable for reproduction.

But I did not have time for more in-depth review.